# Intensity fluctuations in Hurricane Irma (2017) during a period of rapid intensification

William Torgerson[1], Juliane Schwendike[1], Andrew Ross[1], and Chris J. Short[2]

[1]School of Earth and Environment, University of Leeds, LS2 9JT, Leeds, UK
[2]Met Office, FitzRoy Road, Exeter, EX1 3PB, UK

**Correspondence:** William Torgerson (ee16wst@leeds.ac.uk)

**Abstract.** This study aims to understand fluctuations observed in Hurricane Irma (2017), which change the tangential wind speed and the size of the radius of maximum surface wind and therefore affect short–term destructive potential. Intensity fluctuations observed during a period of rapid intensification of Hurricane Irma between 04 September and 06 September 2017 are investigated in a detailed modelling study using an ensemble of Met Office Unified Model (MetUM) convection permitting forecasts. Although weakening and strengthening phases were defined using 10–m wind, structural changes in the storm were observed through the lower troposphere, with the most substantial changes just above the boundary layer (at around 1500–m). Isolated regions of rotating deep convection, coupled to outward propagating vortex Rossby waves, develop during the strengthening phases. Although these isolated convective structures initially contribute to the increase in azimuthally averaged tangential wind through positive radial eddy vorticity fluxes, the continued outward expansion of convection eventually leads to a negative radial eddy vorticity flux which halts the strengthening of the tangential wind above the boundary layer at the start of the weakening phase. The outward expansion of the azimuthally averaged convection also enhances the outflow above the boundary layer in the eyewall region as the convection is no longer strong enough to ventilate the mass inflow from the boundary layer in a process similar to one described in a recent idealized study.

## 1   Introduction

One of the biggest challenges in weather forecasting is predicting when a tropical cyclone (TC) will rapidly intensify. Rapid intensification is defined as a rate of surface wind increase of at least $15.4 \, \mathrm{m \, s^{-1}}$ per 24 hours (Kaplan et al., 2010). Most strong TCs undergo a period of rapid intensification (Kaplan and DeMaria, 2003). Although convection–permitting numerical weather prediction models are capable of producing rapidly intensifying TCs, models still perform poorly when it comes to the timing of rapid intensification events (e.g. Short and Petch, 2018; DeMaria et al., 2021), indicating that the current understanding and representation of intensification processes prior to and during rapid intensification is likely incomplete. Being able to accurately predict rapid intensification events can influence mitigation strategies as the wind speed strongly influences the potential damage the tropical cyclone may cause.

The simplest paradigm for tropical cyclone intensification can be understood by considering the case of a stationary vortex in gradient wind balance. Eliassen (1951) derived the Sawyer–Eliassen equation, that describes the response of the secondary

circulation to angular momentum and heat sources. A point heating source located just within the height–dependent radius of maximum windspeed (RMW) will result in an axisymmetrical response of the secondary circulation, in accordance with the dipolar solutions of the Sawyer–Eliassen equation, with most of the streamlines outside the RMW aligning in the radial direction and most of the streamlines inside the RMW in the vertical direction. The result is a drawing in of absolute angular momentum (AAM) surfaces which, in turn, causes an increase in the tangential velocity, and forms a more intense TC (Vigh and Schubert, 2009).

The boundary layer spin–up mechanism, as described by Montgomery and Smith (2018), has extended the understanding of intensification mechanisms by examining the role of the highly unbalanced boundary layer. If air parcels spiral inwards towards a TC centre fast enough to compensate for frictional AAM loss, then an initially subgradient tangential wind in the boundary layer inflow may become supergradient, allowing the tangential wind within the boundary layer to be higher than the tangential wind above it. The unbalanced mechanism can also spin up the free vortex above the boundary layer through vertical transport of the high AAM air at the top of the boundary layer.

The axisymmetric theory does not fully explain the development of a TC, particularly during rapid intensification, due to the presence of asymmetric processes. These include the role of isolated regions of deep rotating convection, which are local small, regions of high relative vorticity and high vertical velocity regions within the eyewall. Isolated regions of deep rotating convection and their associated downdrafts can act to transport heat and angular momentum inwards to the eye prior to rapid intensification (Guimond et al., 2010) causing the storm to intensify by warming the eye and increasing the relative vorticity in the region of the isolated regions of deep rotating convection. One other phenomenon not accounted for in the balanced, symmetric paradigm is vortex Rossby waves (VRWs) which are waves that propagate on the radial potential vorticity (PV) gradients in tropical cyclones in a similar way to Rossby waves on planetary scale meridional PV gradients (Montgomery and Kallenbach, 1997). Vortex Rossby waves are capable of inducing barotropic instability within the eyewall which can affect the annular heating distribution and therefore impact on the intensity of the storm (Schubert et al., 1999b).

Many of these unbalanced and asymmetric processes have been examined in studies of intensity fluctuations that occur during the intensification of TCs, which are not easily explained by an axisymmetric balanced dynamical theory. One example is vacillation cycles, a form of intensity fluctuations that sometimes occurs during rapid intensification. Nguyen et al. (2011) showed that, during rapid intensification, Hurricane Katrina (2005) exhibited structural changes that caused it to 'vacillate' between monopolar and ring–like radial vorticity distributions, which also led to short–term intensity changes with the more monopolar states associated with acceleration of the tangential wind well inside the RMW and little intensification near the eyewall. The monopolar and the ring–like states were termed 'symmetric' and 'asymmetric' respectively because the former was associated with a smaller azimuthal standard deviation of PV and the latter a higher azimuthal standard deviation of PV. It should be noted that monopolar vs. ring–like and symmetric vs. asymmetric are independent metrics but are, in this case, correlated. Nguyen et al. (2011) showed that the asymmetric states were associated with radially inward moving isolated PV anomalies and related asymmetric periods to barotropic instabilities cooperating with a background convective instability. Hardy et al. (2021) showed similar processes occurring during the rapid intensification of Typhoon Nepartak (2016) with monopolar states associated with near stagnant tangential wind tendency and weaker eyewall updrafts than in the ring–like

phase. Similar changes in structure have been identified in observational data, notably in Kossin and Eastin (2001) who identified two regimes with a monopolar and ring–like angular velocity distribution, which also have concomitant monopolar and ring–like equivalent potential temperature distributions.

Another form of intensity fluctuation was identified recently in Smith et al. (2021) where a vortex, having undergone a period of rapid intensification underwent a relatively brief decay period linked to the inability of the deep convection within the eyewall to ventilate strong boundary layer inflow. A well known form of intensity fluctuation that can occur in strong TCs are eyewall replacement cycles, where convection associated with outer rainbands develop into a second outer eyewall that gradually moves inwards and replaces the original inner eyewall (Willoughby et al., 1982). Eyewall replacement cycles are known to cause large intensity changes in TCs; however, rapid intensification does not typically resume immediately after the formation of the secondary eyewall, although they are often the cause of cessation of a rapid intensification period, for instance in Hurricane Earl (2010) (Montgomery et al., 2014). Diurnal cycles have also been known to induce intensity fluctuations in the TC structure during rapid intensification (Lee et al., 2020; Dunion et al., 2014) although these fluctuations can be explicitly linked to the external environment and have an imposed period of 24 hours.

Hurricane Irma (2017) underwent rapid intensification twice (Fig. 1b). During the latter rapid intensification event intensity fluctuations have been observed by Fischer et al. (2020) who used observational data to identify two periods of weakening during rapid intensification where the RMW suddenly increased. The two periods of weakening were hypothesised to have different causes but were both linked to lower tropospheric convergence and VRW activity. The intensity fluctuations in Fischer et al. (2020) were subtle (relatively small intensity changes compared to most eyewall replacement cycles), but did involve an expansion of the RMW which, as in the case of a full eyewall replacement cycle, can increase the radius of gale force winds and increase the probability of storm surge, hence motivating a need to understand and be able to predict these forms of fluctuations.

In this paper we analyse the intensity fluctuations of Hurricane Irma using both observations and convection–permitting ensemble simulations to help to understand whether or not the inner core intensity fluctuations are a previously unknown phenomenon or exist on a spectrum that may include vacillation cycles, eyewall replacement cycles or other structural changes that occur during rapid intensification. The analysis will involve investigating the cause of the intensity fluctuations and understanding the structural and dynamical changes of the TC in the transition between a strengthening and weakening phase.

The paper will be organised in the following way: Section 2 will describe the evolution of Hurricane Irma during the relevant rapid intensification period and highlight the structural and intensity changes as well as the track. Section 3 will describe the data used in the analysis, including observations, and the setup of the model simulations. The results are presented in section 4 with discussion in section 4.5. Section 5 generalizes the results across more ensemble forecasts and concluding remarks are given in Section 6.

## 2  Synoptic overview of Hurricane Irma (2017)

Hurricane Irma was the first major hurricane of the 2017 North Atlantic hurricane season. Irma peaked at an intensity of $80 \, \mathrm{m \, s^{-1}}$ (1-minute sustained surface wind speeds) with a central surface pressure estimate of $914 \, \mathrm{hPa}$ early on 06 September

before making landfall in Barbuda. A summary of the track of Irma is shown in Fig. 1 along with the best track surface wind speed.

Irma formed out of an African easterly wave off the west coast of Africa at around 30°W, 17°N on 30 August. On 31 August Irma began to rapidly intensify, reaching hurricane strength with a cloud free eye structure and moving in a northwesterly direction. This first period of rapid intensification terminated early on the 1 September with an intensity of 50 $\mathrm{m\,s^{-1}}$ at 03 UTC. Irma remained at an intensity of around 50 $\mathrm{m\,s^{-1}}$ during the period from the 01 September to 02 September and did not intensify further due to sea surface temperatures (26–27°C) and a dry Saharan airmass to the northwest of the storm centre. Irma's track also became more southwestward.

The second period of rapid intensification began on 04 September with Irma intensifying from a Category 3 storm (945 hPa, 50 $\mathrm{m\,s^{-1}}$) at 00 UTC on 04 September to a Category 5 storm (929 hPa, 75 $\mathrm{m\,s^{-1}}$) at 12 UTC on 05 September. At this time, Irma was in a low wind shear environment with sufficient mid–level tropospheric moisture (with the 500–700 hPa relative humidity around 55%) for intensification and high sea surface temperatures of 28–28.5°C . The influence of the subtropical anticyclone to the north of Irma pushed the storm in a westward direction with a translational velocity of about 5 $\mathrm{m\,s^{-1}}$. A peak intensity of 80 $\mathrm{m\,s^{-1}}$ was reached on 06 September at 06 UTC. Irma made landfall in Barbuda at near peak intensity at 0536 UTC with a minimum recorded sea level pressure of 915.9 hPa. During the course of 06 September Irma maintained its intensity and landfall occurred later that day at St. Martin at 1115 UTC and Virgin Gorda at 1630 UTC.

Despite favourable environmental conditions, with a low vertical wind shear, high sea surface temperatures and adequate mid–level moisture, Irma weakened to Category 4 during 07 September due to the start of an eyewall replacement cycle. Irma passed over Little Inagua at 05 UTC on the same day.

Thereafter, apart from a brief period of intensification that occurred around 03 UTC on the 09 September, Irma gradually weakened due to increasing vertical wind shear and eventually land interaction after making landfall in Florida on 11 September. Irma finally dissipated inland on 13 September. Further details on the synoptic overview of Hurricane Irma (2017) are available in Cangialosi and Berg (2018).

## 3  Data and Methods

### 3.1  Observational data

A key source of observational data were aircraft flyovers. Multiple flights were made through Hurricane Irma operated by the National Oceanic and Atmospheric Administration (NOAA). The flyovers were conducted with aircraft from the NOAA aircraft operations centre and the 53rd Weather Reconnaissance Squadron. Observations used from these flights include in-situ wind speed and pressure measurements, dropsondes and airborne radar. Satellite visible, infra-red (IR) and morphed integrated microwave imagery (MIMIC; Wimmers and Velden, 2007) provide additional information. Intensity estimates from the Satellite Consensus (SATCON) algorithm using blended data (Velden and Herndon, 2020) are used in conjunction with those from the lower temporal resolution best track data (HURDAT2; Landsea and Franklin, 2013).

The SATCON intensity estimates are derived from the structure of the TC with heavy usage of microwave and satellite IR imagery, so relating structural changes to intensity changes would be a circular argument. Where possible, therefore, mean sea level pressure (MSLP) data from flights and dropsondes is also used for short periods where there are a large number of flyovers such as in the afternoon of 05 September. MSLP data is preferable to tangential wind data as an intensity proxy, because the latter is strongly dependent on the direction of the flight into the eyewall and the height of the aircraft.

The dropsonde data is available in a quality–controlled post processed format (in some cases raw data was used instead due to lack of availability). In addition some of the NOAA aircrafts are equipped with C–band and doppler radars on the nose, lower fuselage and tail. The processed lower fuselage and tail radar data is used in the analysis and shows precipitation in dBZ reflectivity. All the processed dropsonde and flight–level data used in this analysis is available from the Hurricane Research Division [1].

## 3.2   Intensity fluctuations in observations

The focus of the analysis is on the second period of rapid intensification which starts on 04 September at around 00 UTC and finishes around 00 UTC on 06 September (Fig.1b, Fig.2). During the period of rapid intensification the MSLP decreases from around 970 hPa to its minimum value of 914 hPa. This rapid deepening is interrupted by two periods of stagnation or slight weakening where the MSLP does not continue to decrease. These periods of weakening are marked by blue bands in Fig.2. The 140 first weakening period starts around 13 UTC on 04 September and lasts for about 12 hours and is followed by a strengthening period from 01 UTC on 05 September until 11 UTC on 05 September. The second weakening period starts around 11 UTC on 05 September and lasts for about 4 hours.

    Figure 3 shows observations, from in–flight radar and satellite imagery, of the structural changes just before and after the start of the second weakening period. The convection during the weakening period appears more azimuthally symmetric and 145 continuous as shown in Fig. 3b compared to Fig. 3a where two regions in the north–west and south–east eyewall have relatively high rainrates. The convection is shallower in the weakening period as indicated by warming cloud tops shown in Fig. 3d compared to Fig. 3c. The shallower nature of the convection is also evident in the microwave imagery in Fig. 3e and Fig. 3f. A similar structural change occurs during the first weakening period (not shown) with banded features within the eyewall giving way to broader but shallower convection compared to prior to the weakening period.

## 150  3.3   Numerical model

An 18-member ensemble of convection-permitting forecasts for Hurricane Irma has been produced using a limited-area configuration of the Met Office Unified Model (MetUM; Cullen, 1993), coupled to the Joint UK Land Environment Simulator (JULES) model for the land surface (Best, 2011; Clark et al., 2011). The ensemble forecast was initialised at 00 UTC 03 September 2017 and run out to four days.

The MetUM solves the fully compressible, deep-atmosphere, non–hydrostatic equations of motion using a semi-implicit, semi-Lagrangian scheme (see Wood et al. (2014) for details). Model prognostic variables are defined on a grid with Arakawa–

---

[1]URL: https://www.aoml.noaa.gov/hrd/Storm_pages/irma2017/

C grid staggering (Arakawa and Lamb, 1977) in the horizontal and Charney–Phillips grid staggering (Charney and Phillips, 1953) in the vertical, with a terrain-following vertical coordinate.

Both the MetUM and JULES include a comprehensive set of parametrisation schemes for key physical processes, and the way in which these are configured defines a model science configuration. Here we use the tropical version of the Regional Atmosphere and Land – Version 1 (RAL1-T) configuration presented in Bush et al. (2020), designed for use in km-scale regional models in the tropics. We have made one change to the RAL1-T configuration, which is to reduce the air–sea drag at high wind speeds, as motivated by observational data (Powell et al., 2003; Black et al., 2007). This improves the match to the observed wind–pressure relation of TCs and has since been included in RAL2-T.

The extent of the regional model domain is shown in Fig. 1, and has been chosen so that Hurricane Irma is located well away from the boundaries at the forecast initialisation time. The horizontal grid spacing is 0.04 deg (approximately 4.4km) in both directions, and there are 80 vertical levels with a horizontal lid at 38.5 km above sea level (ASL). The model time step is 75 s.

Each member of the convection-permitting ensemble is one-way nested inside a corresponding member of the Met Office global ensemble prediction system, MOGREPS-G (Bowler et al., 2008). The science configuration used in MOGREPS-G is Global Atmosphere 6.1 (GA6.1; Walters et al. (2017)), which was used operationally at the Met Office for global deterministic and ensemble weather forecasting at the time the research was undertaken. The most important difference between GA6.1 and RAL1-T is that convection is parametrised in the former but explicit in the latter. The global model grid spacings are $0.28125°$ and $0.1875°$ in the zonal and meridional directions (about 31 km $\times$ 21 km in the tropics), respectively. In the vertical there are 70 levels up to a fixed model lid 80 km ASL. The model time step is 450 s. Initial conditions for each MOGREPS-G member are formed by adding perturbations to the Met Office global analysis, where the perturbations are generated using an ensemble transform Kalman filter (Bishop et al., 2001). The initial state of each MOGREPS-G member is then interpolated to the finer regional grid to provide initial conditions for the nested convection-permitting ensemble member. There is no data assimilation or vortex specification scheme in the regional model itself, but central pressure estimates from tropical cyclone warning centres are assimilated as part of the global model data assimilation cycle (Heming, 2016). Lateral boundary conditions for each convection-permitting ensemble member are provided by the driving MOGREPS-G member at an hourly frequency. The initial SSTs, which differ between perturbed members, are held fixed throughout each forecast.

MOGREPS-G includes two stochastic physics schemes to represent the effects of structural and subgrid-scale model uncertainties: the random parameters scheme (Bowler et al., 2008) and the stochastic kinetic energy backscatter scheme (Bowler et al., 2009). These are not included in the convection-permitting ensemble, so that ensemble spread is generated only by differences in the initial and boundary conditions inherited from the driving model.

### 3.4 Diabatic tracers

Incorporated into the MetUM are two sets of tracers (PV and potential temperature, $\theta$) capable of diagnosing diabatic contributions from various parametrisations within the model (Saffin et al., 2016). Examples of this being done previously in extratropical cyclones include, for example, Chagnon et al. (2013). The PV is diagnosed in a semi–Lagrangian way by the tracer such that,

$$\frac{D(PV)}{Dt} = \sum_{phy} \frac{D(PV)}{Dt} + \sum_{dyn} \frac{D(PV)}{Dt} + \varepsilon. \tag{1}$$

The change in PV is given by the sum of increments from all physical processes in the first term represented by the subscript $phy$ (namely radiation, microphysics, gravity wave drag, boundary layer diabatic heating and friction and cloud pressure rebalancing). There are also dynamical processes in the second term represented by the subscript $dyn$ which include the dynamical solver and mass conservation tracers. Ideally these would be zero and preserve the material conservation of PV. However, approximations in the dynamical core mean that such processes may be non zero. The $\varepsilon$ term represents residuals in the PV budget which may come from truncation errors or non linear interaction effects between the physical tracers. It was found that the budget balanced almost perfectly, with the value of $\varepsilon$ at least an order of magnitude lower than the other terms in equation 1. The tracer used most in this analysis is the "initial PV advected" tracer, $PV_{adv}$, which can be used to work out what portion of the change in PV at a particular grid point is due to advection only (i.e. ignoring any change in PV generated by diabatic processes). Every hour the $PV_{adv}$ tracer is reset to the diagnosed PV. The change in PV due to advection at a grid point (x,y,z) over the course of an hour is then given by:

$$PV_{adv}(x,y,z,t+1) - PV(x,y,z,t). \tag{2}$$

### 3.5 TC centre finding method

Much of the analysis is done from an axisymmetric perspective in storm relative cylindrical coordinates. Calculations such as this can be highly sensitive to the location of the storm centre. The simplest way to find the TC centre in the model simulation is to find the coordinates that minimize the surface level pressure field. However, meso–vortices within the eyewall often lead to the minimum surface level pressure being displaced from the geometric centre of the eye into the inner eyewall which can cause the tangential flow within the eye to be overestimated and the tangential flow outside the eye to be underestimated. Several more robust methods have been proposed, each with their own advantages and disadvantages. These include finding PV centroids (e.g. Riemer et al., 2010), geopotential height minima (e.g. Stern and Zhang, 2013) or finding the point that maximises tangential wind speed in cylindrical coordinates at its RMW (e.g. Ryglicki and Hart, 2015).

The method used in this analysis balances the need for a consistent and reliable method for finding the location of the TC centre to an appropriate degree of precision, while considering the computational cost of doing so for 18 ensemble members over a 4 day simulation period. The method used here is similar to the one used by Reasor et al. (2013) for flight–level radar data which can also be applied to model fields and uses a simplex algorithm to find the point that maximises the tangential wind within an annulus with a radius equal to the surface RMW. The simplex algorithm applies geometric transformations to triangles consisting of three points (the simplex) to find the next set of three points. Each point in each simplex is a prospective TC centre where the tangential wind within the surface RMW annulus can be evaluated. For each iteration in the simplex algorithm the three points will, progressively, increase the tangential wind within the surface RMW annulus until it is maximised.

The convergence criteria for the algorithm are: no more than 50 function evaluations, an absolute error between iterations of no more than $0.5\,\mathrm{m\,s^{-1}}$ for the function evaluation, and an absolute error of no more than $0.5\,\mathrm{km}$ between points inside a simplex (well under the grid spacing of the model at $4.4\,\mathrm{km}$). Some studies (e.g. Bell and Lee, 2012) average an ensemble of solutions based on different initial simplexes; however, it was found that changing the location of the initial simplex did not
result in a significantly different TC centre and so a single solution method was used.

## 4    Results

The fluctuations modelled during rapid intensification in Hurricane Irma have similarities to both vacillation cycles and eyewall replacement cycles but with important differences that will be discussed in detail.

### 4.1    Model simulation of intensity fluctuations

The second period of rapid intensification in Hurricane Irma is broadly captured by the convection-permitting ensemble forecasts (Fig. 1). One of the ensemble members (ensemble member 15) was analysed in detail as it was judged to be most representative in terms of the size of the surface RMW, the surface wind speed, MSLP and track, in comparison to the observations. Fig. 4 shows how the MSLP and 10–m total wind speed change in this ensemble member in addition to the surface RMW. The modelled MSLP is slightly higher than the NOAA best track values but the rate of deepening is captured well with the rapid
intensification occurring at the correct time. Even with the reduced drag at high wind speeds the wind–pressure relation in the model is too steep (wind speeds too slow for a given central pressure) and consequently the wind–speed is underestimated compared to observations once rapid intensification occurs. However, the timing of the rapid intensification and its cessation is accurate. The track of this forecast and the other ensemble members are shown in Fig. 1 and all agree reasonably well with the best track.
By examining the change in the 10–m total wind speed, MSLP and 10 m RMW over time (Fig. 4) the development of the TC has been split into distinct phases defined from these quantities. The pre–fluctuation rapid intensification phase covers the first 45 hours of the simulation. During this time, after an initial model spin–up period, the storm intensifies nearly monotonically; the wind speed increases rapidly at all levels (within the lower and mid troposphere), the MSLP decreases and the RMW (at all heights in the lower and mid troposphere) contracts. During weakening phases (blue bands in Fig. 4) the MSLP stagnates
or increases, the maximum unaveraged 10–m total wind speed decreases and the RMW (at both heights) expands.The opposite occurs in the strengthening phases (red bands in Fig. 4).

The maximum tangential wind, particularly near the top or just above the boundary layer (e.g. at 1532 m) also exhibits these fluctuations but does lag behind compared to higher levels (e.g. at 3002 m) where the maximum tangential wind follows a similar pattern to the 10–m total wind speed. The lag is also present in the expansion of the RMW, with the increase in the
RMW happening at 1532 m (dark green line) prior to the increase in the 10–m RMW (aqua line). At the surface, the signal in the tangential wind speed is weaker compared to at higher levels. The role the radial flow plays in modifying the total surface

windspeed during the fluctuations, and the reason for the tangential wind spin–down preceding a weakening phase is explored in detail in Section 4.4.

The simulation shows four weakening periods and three strengthening periods which are defined in terms of 10–m total wind speed, 10–m RMW and MSLP. There is also an uninterrupted period of intensification prior to these fluctuations. During the period of intensity fluctuations from 45 hours to 84 hours Irma is still rapidly intensifying overall, so the brief interruptions in intensification do not stop rapid intensification from happening. The main aim of the analysis is to understand better why these intensity fluctuations happen during this period of rapid intensification,and to gain insight into elements of the mechanism behind the fluctuations and any structural changes with which they are associated. Although the intensity fluctuations have been defined with respect to the surface, for the purposes of easy comparison with observational data, the most dramatic changes occur just above the boundary layer so the subsequent analysis will focus on the 1500–m level and how structural changes at this level impact the boundary layer below it.

It should be noted that during the analysed rapid intensification period Hurricane Irma was a fairly symmetric storm under low vertical wind shear with environmental factors playing a minimal role in these fluctuations. Changes in vertical shear, translation speed, sea surface temperature, maximum potential intensity and the diurnal cycle of convection are not correlated with the intensity fluctuations (not shown).

## 4.2   Barotropic structural changes

### 4.2.1   PV symmetry and structure

Previous studies on vacillation cycles have used PV as a metric to show the structural changes of the vortex during the weakening and strengthening phases. Van Sang et al. (2008) described how, a barotropically unstable ring–like PV state would break down into isolated inward moving PV anomalies. To determine whether the intensity fluctuations are similar to these vacillation cycles it is helpful to examine this PV structure. Fig.5 and Fig.6 show the PV field from a horizontal (just above the boundary layer where the change is most visible) and azimuthally–averaged perspective with times selected to best illustrate the evolution of the PV from just prior to the start of a weakening phase to the end of the weakening phase and start of the next strengthening phase. The evolution during the strengthening phases is less dramatic and is not shown. Prior to each weakening phase the PV field is ring–like and elliptical (Fig. 5a, f, k, p). This elliptical PV field becomes more circular at the start of each weakening phase (Fig. 5b,g,l,q). The PV field also becomes less ring–like during a weakening phase with higher PV in the centre of the storm and lower PV in the eyewall. A comparison of Fig. 6a,f,k,p with Fig. 6b,g,l,q shows that the weakening of the ring–like PV structure at the start of the weakening phase occurs primarily just above the boundary layer especially between 1 km and 2 km height. The trend towards a less ring–like distribution continues to the middle of the weakening phases where a 'C' shaped ring of high PV (Fig. 5c,h,m,r) develops near the TC centre above the boundary layer (Fig. 6c,h,m,r). The PV within the boundary layer also declines but maintains a more ring–like structure. The end of the weakening phase is characterised by the upward movement of the high PV zone at around 2 km height in the eye (Fig. 6d,i,n,s), and re–formation of a weak, circular, PV ring above the boundary layer (Fig. 5d,i,n,s). The start of the strengthening phase roughly coincides

with the strengthening of this new PV ring (Fig. 5e,j,o,t) which becomes increasingly elliptical during the strengthening phase. The elliptical to circular transitions are particularly prominent in W1 and W4 which are more pronounced weakening phases than W2 and W3.

Figure 7a summarises these PV structure changes throughout the simulation with an index that describes how ring–like the PV distribution is above the boundary layer (Hardy et al., 2021). Higher values of the ratio $PV_0/PV_{max}$, where $PV_0$ is the layer averaged PV at the centre of the storm and $PV_{max}$ is the maximum layer averaged PV, imply the vorticity structure is less ring–like with a weaker radial PV gradient while lower values imply the structure is more ring–like. A value of 1 for this index would imply the PV structure was monopolar with the maximum value achieved in the centre of the storm.

During the weakening phases there is a trend for the PV structure to become less ring–like. At the end of each weakening phase the trend suddenly reverses and the vorticity structure becomes more ring–like. The change in the tendency of the vorticity structure is very sudden and coincides exactly with the start and end of each phase. However, as indicated by Fig. 6 the PV distribution does not change uniformly at all heights. At lower levels closer to the boundary layer the PV field is less ring–like at the beginning of the weakening phase, while at higher levels it lags behind and is less ring–like at the start of the next strengthening phase. Note how the storm is continually transitioning away from or towards a ring-like structure. This behaviour is different to intensity fluctuations associated with vacillation cycles where the storm can remain in a fully monopolar state for 10 hours or more (Hardy et al., 2021). It should be noted that the more dramatic weakening phases, W1 and W4 shown in Fig. 5a–e,p–t and Fig. 6a–e,p–t are associated with a more pronounced realignment of PV both in terms of the ring becoming less ring–like and an overall decrease in PV between Fig. 5 c,r and Fig. 5 d,s. Fig. 7a shows a much bigger increase in $PV_0/PV_{max}$ for W1 and W4 compared to W2 and W3. This is also seen in Hardy et al. (2021) with a greater change in $PV_0/PV_{max}$ associated with a more dramatic intensity fluctuation. Other metrics that describe the barotropic structure (Fig. 7b–d) also show a more pronounced change during W1 and W4 compared to W2 and W3. Annular vorticity rings, without the presence of diabatic forcing, are unstable and breakdown via the formation of mesovortices into a monopole like structure (e.g. Prieto et al., 2001; Schubert et al., 1999a; Kossin and Schubert, 2001). A similar mixing process between the eyewall and the eye may be present here. To test whether PV transport between the eye and eyewall is occuring, Fig. 8 shows the PV tendency due to radial and vertical advection only over the previous hour. [2] The start of the weakening phase shows PV transported to the eye at T+45 h (Fig. 8a). At T+48 h (Fig. 8b) the PV transport occurs above the boundary layer including at the 1532– m level shown in Fig. 5. At T+45 h the transport of PV into the eye at this level is weak with different azimuthal starting points in the trajectories leading to rather different end points. Therefore, the gain of PV within the eye is due to eddies transporting more PV inwards than outwards. By T+48 h there is a more distinct vertical transport of PV in the eye from the boundary layer. So, the change to a less ring–like structure can be explained by an initial inward asymmetric radial transport of PV within the eye followed by the development of a very weak (on the order of 0.02 m s$^{-1}$), deep ascent layer, transporting PV slowly upward. PV is also transported radially inward in the eye although the radial transport is weak (trajectories in Fig. 8b). The weak ascent

---

[2]The PV tendency due to the physical processes has also been calculated, with the cloud rebalancing term (PV change due to cloud formation) dominating. Overall the PV change due to physical processes is large and positive on the inner side of the eyewall and responsible for the maintenance of the PV ring structure to counterbalance the PV loss due to vertical upward transport.

that develops within the eye originates within the eyewall and gradually extends inwards into the eye (not shown). The upward vertical motion is weak and inconsistent, only becoming apparent when 10–minute data is averaged over an hour. The PV contribution from diabatic processes other than large scale transport, during the weakening phase, is negative indicating the entire positive PV tendency is linked to movement of PV into the eye. The negative PV tendency regions in Fig. 8 are caused by the loss of PV through the updraft in the eyewall. There is also a gain of PV advected near the surface particularly at T+48 h (Fig. 8b) which can be linked to an increase in the inflow within the eye region and transport of frictionally generated PV from greater radii.

In addition to the radial PV structure the PV also varies azimuthally with the intensity fluctuations. One way of describing the azimuthal PV symmetry is the method of Nguyen et al. (2011) and Reif et al. (2014), where the azimuthal standard deviation of PV is calculated at each radius and the maximum value is taken. A high standard deviation of PV implies a less azimuthally symmetrical storm. It should be emphasised that this is a separate metric not related to the radial distribution of PV (i.e. monopolar and ring–like distributions). In the case of Nguyen et al. (2011) for example, the radial and azimuthal measures of PV were used interchangeably to describe 'symmetric' or 'asymmetric' states (the ring–like PV distribution in Nguyen et al. (2011) was correlated to an azimuthally symmetric state which is not the case here). In this study, references to symmetry only refer explicitly to variations in the azimuthal distribution of PV.

Figure 7b shows how this metric varies throughout the simulation. The red curve shows that the change in the variation of azimuthal PV at the RMW follows a similar pattern to the maximum azimuthal PV (black line). At the start of a weakening phase the maximum azimuthal standard deviation of PV decreases rapidly or becomes more azimuthally 'symmetrical' with the inverse happening during strengthening phases. The weakening phases are, therefore, characterised by more azimuthally symmetric, less ring–like PV fields while the strengthening phases are characterised by a less azimuthally symmetric, more ring–like PV distribution. The azimuthal symmetrisation of the PV field occurs at approximately the same time that the field becomes less ring–like. This contrasts with prior work on vacillation cycles (e.g. Nguyen et al., 2011) where a more azimuthally symmetric PV field in Hurricane Katrina (2005) was associated with a ring–like distribution of PV. The change in the azimuthal symmetry is also described in Fig. 7c which shows that during the strengthening phases the initially circular PV rings become increasingly more elliptical (higher eccentricity) confirming that the start of a weakening phase is associated with a rapid change from an elliptical PV ring to a more circular one (also seen in Fig. 5).

To attempt to explain the causes of the change in PV structure the barotropic conversion rate was computed (as in Van Sang et al., 2008), their equation (1):

$$BARO = \overline{u'u'}\frac{\partial \overline{u}}{\partial r} + r\overline{uv'}\frac{\partial}{\partial r}\left(\frac{\overline{v}}{r}\right) + \overline{u'\omega'}\frac{\partial \overline{u}}{\partial p} + \overline{v'\omega'}\frac{\partial \overline{v}}{\partial p} + \frac{\overline{u}}{r}\overline{v'v'},$$

(3)

where $BARO$ is the barotropic conversion rate, $u$ is the radial wind, $v$ is the tangential wind, $\omega$ is the vertical velocity, $p$ is the pressure, primes represent the perturbation from the azimuthal mean of these quantities, and the overbar represents the azimuthal average.

The barotropic conversion rate describes how kinetic energy is transferred between eddies and the mean flow. Hankinson et al. (2014) showed that the conversion rate, in their simulation, is always negative which implies a conversion of kinetic energy between the mean state and the eddy state. It is worth noting that despite kinetic energy always flowing from the mean to eddy state the storm does not necessarily spin down due to other terms in the kinetic energy budget (given in Appendix 2 of Hankinson et al. (2014)) in particular the radial and vertical mean kinetic energy fluxes.

Figure 7d shows the barotropic conversion rate as a function of time. The beginning of the weakening phase is accompanied by a distinct rise in the barotropic conversion rate (it becomes less negative) while the start of the strengthening phase is associated with a more negative conversion rate. As the strengthening phases are associated with a less symmetric PV structure more kinetic energy is transferred from the mean state to the eddy state. The start of a weakening phase is therefore associated with a rapid reduction in the amount of kinetic energy transferred away from the mean state to the eddy state. The magnitude of the barotropic conversion rate is typically at its lowest at the end of a strengthening phase which is also when the isolated regions of deep rotating convection are at their strongest and implies that barotropic instability may be at its greatest.

### 4.2.2 Isolated regions of deep rotating convection

In order to understand the role of these isolated regions of deep rotating convection in the intensity fluctuations, their strength and prevalence prior to and during the weakening phases will be examined, particularly in their appearance as a manifestation of cooperative barotropic and convective instability. The involvement of the isolated regions of deep rotating convection as a potential trigger for the weakening will also be investigated.

During the strengthening phases, isolated regions of deep rotating convection are apparent as small–scale local regions of high vorticity and vertical velocity within the eyewall. These features resemble vortical hot towers (VHTs), formally defined in Smith and Eastin (2010) as regions with maximum perturbation vertical velocities greater than $5\,\mathrm{m\,s^{-1}}$ over a depth of $6\,\mathrm{km}$ and perturbation relative vorticity greater than $10^{-3}\,\mathrm{s^{-1}}$ over at least half of the updraught and with the perturbation vorticity maximum below the vertical velocity maximum. The structures here do not meet these strict requirements, however, it is common to see updraughts, several kilometres deep, with $3$–$5\mathrm{m\,s^{-1}}$ perturbation vertical velocities and maximum pertubation relative vorticities above $10^{-3}\,\mathrm{s^{-1}}$. These structures appear frequently and may play a significant role in the development of the cyclone. Since they look like VHTs but are not strong or deep enough to meet the criteria for a VHT they will simply be described as isolated regions of deep rotating convection.

Figure 9 shows perturbation vertical velocity and relative vorticity at different heights at the same times as in Fig. 5. The isolated regions of deep rotating convection are more likely to be present during strengthening phases (particularly towards the end of the strengthening phases) and rarely form during weakening phases although an already existing isolated region of deep rotating convection may persist for a couple of hours into the weakening phase. These structures typically last on the order of an hour which is a little shorter than the lifespan of similar convective structures found by Yeung (2013) during the rapid intensification of Typhoon Vicente. The isolated regions of deep rotating convection move anticlockwise but slower than the tangential flow. Filaments of high pertubation vertical velocity and cyclonic pertubation relative vorticity emanate outward from these isolated regions of deep rotating convection (see, for example Fig. 9p north of the RMW) as convectively coupled

vortex Rossby waves. The Fourier decomposed PV anomalies (not shown) associated with the outward propagating filaments of cyclonic vorticity and ascent were largely wavenumber–2 and moved radially, azimuthally and vertically as predicted by the vortex Rossby wave dispersion relation (Montgomery and Kallenbach, 1997) giving strong evidence that the anomalies were vortex Rossby waves. It is therefore fairly common, within the strengthening phases (when the wavenumber–2 anomalies are strongest), to see two isolated regions of deep rotating convection at once which typically are 180 degrees from each other. In this case one isolated region of deep rotating convection tends to be much stronger than the other. An example of this is shown in Fig. 9a with the isolated region of deep rotating convection in the southwest quadrant being more intense and deeper than the one in the northeast quadrant.

During the weakening phases isolated regions of deep rotating convection rarely form such that in the middle of a weakening phase it is unusual to see one of these structures. The T+72.2 h panel (Fig. 9 m) does show a weak, shallow, isolated region of deep rotating convection in the northwest quadrant though it should be noted that W3 is the weakest weakening phase. Towards the end of a weakening phase isolated regions of deep rotating convection may redevelop and often form outside of the RMW. The T+50.7 h panel (Fig. 9d) shows signs of an isolated region of deep rotating convection on the eastern side of the TC outside of the RMW that forms before moving inwards. If Fig. 9 is compared to Fig. 5 it can be seen that the isolated regions of deep rotating convection are typically located at the two points on the elliptical PV rings furthest away from the centre (i.e. along the semi–major axis of the PV elliptical ring). Away from the two isolated regions of deep convection there is often weak ascent in the eyewall region but also downdraughts associated with the isolated regions of rotating deep convection. The strongest isolated regions of deep rotating convection tend to form just prior to a weakening phase and may last for the first few hours of the weakening phase. The convective structures in Fig. 9a,p are examples of particularly strong isolated regions of deep rotating convection that occur just prior to the W1 and W4 phases respectively but are shown to very quickly dissipate during the start of W1 and W4 respectively (Fig. 9b,q). The regions of locally high vertical velocity and relative vorticity associated with the isolated regions of deep rotating convection becomes increasingly de–localized and distributed over the entire eye–wall region resulting in a more axi–symmetric structure. Any regions of high pertubation vorticity or vertical velocity that form during the weakening phases are much weaker and shallower than the isolated regions of deep rotating convection that form during the strengthening phases (such as the low–level region of high relative vorticity north–west of centre in Fig. 9m) or occur well outside of the RMW (such as the updraught south–east of centre in Fig. 9r).

### 4.2.3 Tangential wind budget

The spin–up of a TC can be examined in terms of the tangential wind budget which describes contributions to the mean tangential wind tendency from radial and vertical advection, which can be further split up into mean and eddy contributions. A form of the tangential wind budget based on Persing et al. (2013) is:

$$\frac{\partial \overline{v}}{\partial t} = -\overline{u}\,\overline{(f + \zeta)} - \overline{w}\frac{\partial \overline{v}}{\partial z} - \overline{(u'\zeta')} - \overline{\left(w'\frac{\partial v'}{\partial z}\right)} + F, \tag{4}$$

where $v$ is the tangential wind, $u$ is the radial wind, $w$ is the vertical velocity, $f$ is the Coriolis parameter, and $\zeta$ is the relative vorticity. Overbars represent azimuthal averages of these terms while primes represent perturbations from the azimuthal average. The terms on the right hand side of the equation from left to right are: mean radial vorticity flux, mean vertical advection of absolute angular momentum, eddy radial vorticity flux and vertical eddy advection of absolute angular momentum. The final term, $F$, represents sub–grid frictional contributions to the budget which are negligible outside of the boundary layer.

In order to understand the contribution of the isolated regions of deep rotating convection to the spin–up or spin–down of the TC, the eddy and mean contributions to the tangential wind budget were examined. Fig. 10 shows the contributions to the tangential wind budget through mean and eddy radial vorticity fluxes and vertical advection of AAM. Near the eyewall, the mean term has a positive contribution to the tangential wind in the boundary layer due to the radial inflow and a negative contribution above the boundary layer where the boundary layer outflow jet is (Fig. 10a,c). The larger positive contribution to the tangential wind in the boundary layer, and larger negative contribution above the boundary layer in S1 compared to W1 is attributed to a stronger inflow and outflow in and above the boundary layer respectively.

Just above the boundary layer the eddy term has a positive contribution to the tangential wind budget in both S1 and W1 (Fig. 10 b,d). However, in S1 the magnitude of the positive eddy contribution above the boundary layer (around 1500 m) is larger. The positive eddy contribution is mostly associated with the positive radial eddy contribution to the tangential wind budget (not shown). This finding is robust across all strengthening and weakening phases and extends generally to other ensembles that show these intensity fluctuations (see Section 5). The greater positive contribution, to the tangential wind, of the eddies just above the boundary layer during the strengthening phases is associated with isolated regions of deep rotating convection. These results are illustrated in Fig. 11 which shows during the 45.5 hour to 57.5 hour period (comprising both W1 and S1 periods) a composite of all times where there are either no isolated regions of deep rotating convection (Fig. 11 a,b) or many strong isolated regions of deep rotating convection (Fig. 11 c,d). In total there were 12 times where many strong isolated regions of deep rotating convection were present and 10 times where no isolated regions of deep rotating convection were present during this period. Compositing times where the isolated regions of deep rotating convection were present or not present allows the effect of the isolated regions of deep rotating convection to be analysed more directly. As can be seen by comparing Fig. 11 b and d isolated regions of deep rotating convection are associated with an increased positive tangential wind tendency from the eddy terms just above the boundary layer compared to times without isolated regions of deep rotating convection. This increased positive tangential wind tendency is despite the increase in the negative contribution from the mean flow (Fig. 11 a,c). It is harder to say if the association between isolated regions of deep rotating convection and an increased eddy positive wind tendency above the boundary layer is causal and may instead be related to the relative frequency of isolated regions of deep rotating convection during weakening phases compared to strengthening phases. Times during S1 with isolated regions of deep rotating convection (not shown) were associated with greater eddy tangential wind tendency compared to times during S1 without isolated regions of deep rotating convection but the effect was small.

However, the radial location of the isolated regions of deep rotating convection seems to be important, the isolated region of deep rotating convection inside the RMW in Fig. 9p is concurrent with an eddy effect that spins down the eyewall (negative

contribution to the tangential wind budget) and spins–up the eye (not shown). Likewise the isolated region of deep rotating convection in Fig. 9t is associated with a positive eddy tangential tendency outside the eyewall and a spin down within the eyewall. During the strengthening phases isolated regions of deep rotating convection become more prevalent due to the presumed increase in barotropic instability. The convection from the isolated regions of deep rotating convection, in turn, may have the ability to change the PV structure of the storm by enhacing the growth of the barotropically unstable modes (Nguyen et al., 2011) particularly wavenumber 2 (not shown). Stirring in higher PV from the eyewall into the eye can spin up the eye (e.g. Hankinson et al., 2014) and induce a weakening of the ring–like vorticity structure.

## 4.3 Convective structural changes

To understand how the convective structures change with the intensity fluctuations the diabatic heating profiles are investigated, in particular, how the heating profiles change from strengthening phases transitioning to weakening phases. Understanding the distribution of the diabatic heating and its vertical and radial gradients can allow links to be made with the barotropic structure, through the spatial gradient of diabatic heating term in the PV generation equation. The diabatic heating (Fig. 12 and 13) is calculated using Eularian potential temperature increments directly output from the MetUM. The main contribution to the potential temperature budget, below the freezing level, is from the latent heating associated with cloud formation. The boundary layer scheme has a small contribution to the diabatic heating but this contribution does not change between the strengthening and weakening phases.

During both weakening and strengthening phases there are some similarities, notably two separate heating maxima, one in the inflow boundary layer at around 1 km and the other in the mid–troposphere associated with the latent heat release above the freezing level in the free vortex at around 7 km. The majority of the heating occurs around the RMW in the eyewall, although small amounts of heating also occur out to 150 km associated with outer rainbands.

One of the biggest differences between the weakening and strengthening phases are the radial extent of their respective azimuthally averaged heating distributions. All of the weakening phases have a heating distribution with a greater radial extent compared to all of the strengthening phases (not shown). This can also be seen in the observations in Fig. 3 a,b which shows the convection in the eyewall appearing to thicken with the moderately high precipitation rates occupying a greater radial extent during a weakening period than just prior to it. The overall heating rates are substantially weaker during the middle of the weakening phases compared to the strengthening phases (e.g. a maximum of around 30 $K h^{-1}$ in the middle of W1 compared to around 45 $K h^{-1}$ at the start of S1) with substantial heating occurring outside the RMW. In the strengthening phases the heating is concentrated in a narrow band (of around 10 km width) just inside the RMW, while in the weakening phases the heating maximum is shifted outside of the RMW. Just above the boundary layer there is a heating maximum in both the strengthening and weakening phases, the heating here is stronger in the strengthening phases but is located inside the RMW during both the weakening and strengthening phases. The dominant component of diabatic heating, just above the boundary layer is from the latent heating due to cloud formation at the top of the boundary layer. The change in heating distribution during the course of the strengthening phases (not shown) is much less significant with no secondary heating

maxima appearing, although there is a tendency for the diabatic heating within the eyewall to become a bit stronger during the course of a strengthening phase.

The effect of eddy diabatic heating was also investigated. These results are not shown since the azimuthally averaged eddy heating was small, typically an order of magnitude smaller than the mean heating terms which is similar to the results of, for instance, Montgomery and Smith (2018). The eddy terms had the largest contribution just below the freezing level and had a dipole–like structure with heating below and cooling above. No significant differences in the azimuthally averaged eddy heating distribution were detected between the strengthening and weakening phases with eddy momentum effects from the isolated regions of deep rotating convection playing a much more prominent role in causing the intensity fluctuations than their effect on azimuthally averaged eddy diabatic heating.

In terms of how the heating distribution changes just prior to a weakening phase Fig. 12b,c shows a secondary heating maxima at around 55 km radius and 5 km height associated with the inner rainbands. Along these rainbands near their intersection with the eyewall there are regions of enhanced convection which can be seen in Fig. 13a T+44.5 h in the northwest and southeast associated with isolated regions of deep rotating convection which are responsible for most of the heating. The secondary heating maxima associated with the inner rainbands becomes more distinct by T+45.5 h (Fig. 12b) which develops into a secondary updraft by T+46.5 h (Fig. 12c). A single isolated region of deep rotating convection is still visible at T+46.5 h in the southeast quadrant (Fig. 13c). However, by T+47.5 h (Fig. 13d) an azimuthal symmetrisation has taken place with the inner-rainband convection visible as a second ring outside the eyewall. The heating from isolated regions of deep rotating convection that occur in the inner rainbands near where they intersect with the eyewall becomes less significant between T+44.5 h and T+47.5 h (Fig. 12a–d), but the secondary heating maximum from the inner rainbands becomes more distinct (Fig. 13a–d).

Over the next few hours the secondary convective ring becomes more symmetrical and the isolated regions of deep rotating convection continue to become less visible. Eventually by T+50.5 h the secondary convective ring has replaced the first (Fig. 13g). In the remaining hour of W1 the RMW expands out to coincide with the diabatic heating maximum. Note, the inner rainband activity and the associated isolated regions of deep rotating convection may be necessary conditions for a weakening phase to begin; however, they are not sufficient. For example, prior to W1 a VRW event at T+38 h led to the development of a secondary convective ring, which subsequently weakened and did not replace the primary ring. At around T+35 h there were many strong isolated regions of deep rotating convection, in the eyewall region, but they did not lead to an intensity fluctuation.

It was found that weakening phases were associated with weaker heating outside of the RMW compared to strengthening phases associated with stronger narrower columns of diabatic heating just inside the RMW which is consistent with a simple balanced dynamical interpretation (e.g. Smith and Montgomery, 2016) whereby convection occurring outside the RMW acts to spin–up the primary circulation outside the RMW and spin–down the primary circulation inside the RMW. The increase in convection outside of the RMW is linked to the ascent associated with the isolated regions of deep rotating convection spreading out azimuthally and evolving from isolated regions of convection to a ring of ascent outside of the eyewall. The convection then becomes increasingly dominant at this slightly greater radius over a period of a few hours and the RMW increases.

## 4.4 Unbalanced dynamics and the boundary layer

If the boundary layer plays a significant role in the cause of the intensity fluctuations then it may be necessary to attempt to understand the fluctuations in terms of the boundary layer spin–up mechanism as described by Montgomery and Smith (2018). This requires air parcels within the boundary layer to gain enough AAM through rapid reduction of radial distance that it counteracts the reduction in AAM caused by friction so that the tangential wind speed is able to increase. A consequence of this is the initially subgradient tangential wind within the boundary layer becoming supergradient. Examining the agradient wind in and above the boundary layer allows the importance of the unbalanced spin–up mechanism in the intensity fluctuations to be determined.

### 4.4.1 Primary and secondary circulation in or just above the boundary layer

The agradient wind is the deviation of the tangential wind from gradient wind balance (as in, for example, Miyamoto et al., 2014). The gradient wind is not output directly from the MetUM but calculated from other diagnostic variables. Details of the form of the agradient wind are available in the Appendix.

Figure 14 shows how the agradient wind, the tangential and radial wind vary throughout the simulation both at the radius of 35 km and at the RMW (such that the agradient wind can be examined both at the eyewall and at a fixed radius as during a weakening phase the RMW increases). A negative agradient wind corresponds to a subgradient flow while a positive agradient wind corresponds to a supergradient flow. The blue curve near the surface is chosen to show the subgradient boundary layer flow. The green curve shows the agradient flow a little higher up but still within the boundary layer (Fig.14a) this is at a height where during the weakening phases the subgradient flow becomes supergradient indicated by the crossing of the zero line. The yellow curve is at a height that roughly corresponds to the middle of the outflow jet and the red curve represents a level near the top of the outflow jet where the flow has returned to near gradient wind balance.

Throughout the storm's lifetime the tangential wind is supergradient near the eyewall within the boundary layer, with the highest agradient wind being around 670 m. The supergradient wind is advected vertically upwards; above the boundary layer the radial outflow removes more absolute angular momentum than is gained by the vertical advection so the wind is near gradient wind balance. Above the boundary layer, the storm can intensify in two ways described in Schmidt and Smith (2016); either though the classical spin–up mechanism where a balanced inflow radially advects AAM inwards or through the unbalanced spin–up mechanism where AAM from the boundary layer is transfered upwards into the free vortex. In order for the tangential wind, above the boundary layer, to increase by the unbalanced spinup mechanism the contribution from the vertical advection of high AAM from the boundary layer must exceed the AAM lost through the outflow jet advecting low AAM from the eye. In the weakening phases, the unbalanced spin-up mechanism is unable to increase the tangential wind above the boundary layer but it is in the strengthening phases. Throughout the simulation the classical intensification paradigm is still able to spin up the TC outside the eyewall, and within the inner rainband region.

Just prior to the weakening phase the inflow in the boundary layer at a radius of 35 km decreases (Fig. 14d) while the inflow at larger radii (e.g. 100 km) may increase (not shown). This decrease in inflow at small radii is followed by a marked increase

in the agradient wind at all levels (Fig. 14a,c). The increase in the agradient wind is not accompanied by an increase in the tangential wind (Fig. 14b) at any level which implies the increase in the agradient wind is caused by a decrease in the pressure gradient force per unit mass (PGF) which is also shown in Fig. 14 a and c. The decreased PGF, above the boundary layer is accompanied, by a decrease in the tangential wind (Fig. 14 b, yellow and red lines) and therefore the centrifugal and coriolis force such that approximate gradient wind balance is maintained. Any weakening in the tangential wind above the boundary layer (Fig. 14b, yellow and red lines) would result in a decrease in the PGF (assuming approximate gradient wind balance is maintained), this reduction in the PGF would then be instantaneously transmitted down within the boundary layer (Fig. 14a, dotted blue and green lines). The reduction in the PGF within the highly unbalanced boundary layer is not accompanied by the same immediate reduction in the centrifugal and coriolis force leading to an increase in the agradient wind and a modest decline in the frictionally induced inflow (Fig. 14d, green and blue lines).

The reduction in the boundary layer inflow from the decrease in the PGF is not enough to spin–down the boundary layer and the frictionally induced inflow remains strong. Therefore, at the surface, the reduction in maximum total winds (black line in Fig. 4) during the weakening phases are not due to a tangential wind decrease in the boundary layer but rather a combination of a decrease in the radial inflow and an azimuthal symmetrisation of the wind field (i.e. the maximum 10–m total wind speed decreases faster than the mean (azimuthally averaged) 10–m wind speed).

During the weakening phase an increase in the agradient wind is seen within the boundary layer (Fig. 14 a and c) which contributes in part to a stronger outflow jet just above the boundary layer (Fig. 14d). This enhanced outflow jet continues to increase throughout the weakening phase and reaches a maximum at the start of the next strengthening phase.

The start of a strengthening phase is characterised by a strong outflow jet and a slightly subgradient 'overshoot' (red line in Fig. 14a slightly below zero near the start of the strengthening phases) i.e. as the ascending air within the super–gradient layer decelerates it overshoots to a value lower than the gradient wind, a centrifugal wave effect described in Persing et al. (2013).

### 4.4.2 Mass ventilation

A key feature that appears during the weakening phases is a thin layer of outflow above the boundary layer which has been noted to occur in order to return the unbalanced supergradient tangential flow to gradient wind balance above the boundary layer. Another contributing factor to this outflow layer is a mismatch in the mass flux expelled from the boundary layer and ventilated by the deep convection. The residual mass that cannot be evacuated through the main system scale tropospheric outflow channel must leave through the outflow jet at the top of the boundary layer. In order to better understand the changes in the strength of the outflow jet and its importance in causing weakening phases the ventilation diagnostic as developed in Smith et al. (2021) will be examined. Their equation 1 for the ventilation diagnostic is given as:

$$\Delta M_{flux}(R_{int}, t) = 2\pi \int_0^{R_{int}} [<\rho w>_{z=Uppertrop} - <\rho w>_{z=BL}] r dr, \tag{5}$$

where $\Delta M_{flux}$ represents the ventilation diagnostic and triangular brackets indicate azimuthally averaged quantities as a function of the integration radius and time. This ventilation diagnostic describes the ability for deep convection within the TC,

at a given radius, to evacuate mass through flowing inwards in the boundary layer (z=BL) outwards in the upper troposphere (z=Uppertrop), the levels used are 5955 m for the upper troposphere and 1052 m for the boundary layer . If the convection is not strong enough to ventilate the converging mass within the boundary layer then there will be an outflow jet at the top of the boundary layer in addition to the main upper tropospheric outflow. A positive value of $\Delta M_{flux}$ indicates that the convection, at that radius, is more than capable of ventilating mass inflow, while a negative value of $\Delta M_{flux}$ indicates the convection is unable to ventilate the mass inflow at that radius.

Figure 16 shows the ventilation diagnostic over time as well as the radial inflow at the surface and outflow above the boundary layer. Throughout the tropical cyclone development the ventilation index is negative which at least partially explains the ubiquitous presence of the boundary layer outflow jet throughout the simulation. In Fig. 16 a,c it can be seen that prior to a weakening phase the 60–80 km radial region where inner rainbands and isolated regions of deep rotating convection are active has near neutral or a slightly positive ventilation indicating that convection is strong enough in this region to evacuate mass from the boundary layer. During the strengthening phase as deep tropospheric convection increases the ventilation index becomes more positive. However, this enhances boundary layer convergence through an increased near surface inflow (Fig. 16 b,d) which eventually leads to the ventilation index in this inner–rainband region becoming negative and in turn leads to a positive outflow above the boundary layer. During the weakening phase the inflow continues to increase outside of the eyewall while the boundary layer outflow advects low absolute angular momentum air outwards and decelerates the wind inside the eyewall. This, in turn, weakens the eyewall convection and enhances the outflow within the eyewall itself as even more mass is unable to be ventilated.

### 4.4.3 Tangential wind budgets

To understand how the boundary layer and outflow jet change and lead to a spin–down above the boundary layer Fig. 15 shows how the primary and secondary circulation change and what drives these changes by using the tangential wind budget decomposition. The times shown correspond to the times in Fig. 5a–c.

The increase of the agradient wind at the start of the weakening phase leading to an intensification of the outflow jet can be seen by comparing Fig. 15a with Fig. 15c. The main result of this comparison is a radial advection of low angular momentum (Fig. 15d) which acts to cause a spin–down of the eyewall above the boundary layer (Fig. 15c). The spin–down of the tangential wind just above the boundary layer pushes the RMW outwards and results in the 'kink'–like appearance of the RMW. Above the kink the tangential wind is in approximate gradient balance and the flow runs nearly parallel to the AAM surfaces. Eventually the expansion of the RMW above the boundary layer in combination with the weakening inflow within the boundary layer leads to the vertical advection of angular momentum into the low angular momentum region above the boundary layer which can be seen in the pink area near the RMW (in the highlighted yellow ellipse) in Fig. 15f compared to Fig. 15d where the same region is blue. At the increased radius, the coherent eyewall structure reforms with a spin–up as a result of the vertical advection of absolute angular momentum. The outflow jet, which previously reduced the tangential wind in the eyewall now does so within the eye which brings the TC into a strengthening phase. The PGF increases, the supergradient wind in the boundary layer becomes less supergradient, and the outflow jet weakens.

In summary the intensity fluctuations in Hurricane Irma can be understood in terms of unbalanced boundary layer dynamics and the interplay between the boundary layer and the free vortex above. Firstly the agradient wind in the boundary layer increases as a result of a decline of the PGF which is, itself, caused by an initial decrease in the azimuthally tangential wind above the boundary layer. The rapid increase in the supergradient wind within the boundary layer is associated in part with an intensification of the outflow jet just above the boundary layer which acts to spin down the primary circulation above the

boundary layer by advecting in low angular momentum air from the eye, as well as expanding the RMW above the boundary layer. An increased super–gradient wind also implies a stronger agradient force, promoting ascent out of the boundary layer at larger radii which can be seen explicitly by looking at Fig.12h; the eye–wall forms at approximately the same radius as the updraft located further from the centre of the storm in Fig.12d.

### 4.5    Discussion

During the weakening phases the RMW expanded, the azimuthally averaged tangential wind speed (at all height levels in the lower and mid troposphere) decreased and the MSLP stagnated or rose, whereas during the strengthening phases the opposite occurred.

    The fluctuations observed in Hurricane Irma are proposed to be the result of changes in the barotropic structure (namely the proposed onset of barotropic instability during the strengthening phases) cooperating with convective instability to reduce

the e–folding time of disturbances from barotropic instabilities similar to the arguments presented in Hankinson et al. (2014) where both convective and barotropic effects caused vacillation cycles in Hurricane Katrina (2005). Barotropic instabilities are initially proposed to grow during the strengthening phases and are represented as wavenumber–2 PV (or relative vorticity) anomalies constrained to outward propagating vortex Rossby waves. [3] The proposed cooperation between barotropic and convective instability can be explained by an Ekman pumping–like effect. Under the assumption that the rotating convection

has a large enough scale for the ekman balance to be valid, the ascent depends on both relative vorticity and radial gradients of vorticity (Smith and Montgomery, 2021) such that isolated regions of rotating deep convection encourage ascent out of the boundary layer which helps convective inhibition to be overcome and convective instability to arise. Throughout the course of the strengthening phase mean kinetic energy is converted into eddy kinetic energy (Fig. 7d) in tandem with the strengthening of the isolated regions of rotating deep convection.

The start of a weakening phase during a rapid intensification period is associated with the presence of isolated regions of deep rotating convection built up over the course of the preceding strengthening phase, that produce significant heating and are visible in an azimuthally averaged perspective as a secondary updraft outside of the RMW (Fig. 19a,b). We showed that PV is transported into the eye during the weakening phase. We have been able to extend the work of Hankinson et al. (2014), by showing that PV increases within the centre of the eye were caused by apparent upward and inward advection of PV from the

outer eye region using a Lagrangian tracer method. In contrast to Hankinson et al. (2014) this increase of PV in the eye was

---

[3]The rise in barotropic instability is inferred by the growing wave-2 PV anomalies during the strengthening phase and the satisfaction of the Rayleigh-Kuo criterion where a sign change is evident in the radial gradient of vorticity. However in order to fully verify the existence and increase in barotropic instability a linear stability analysis would be a useful extension to this paper.

associated with a small increase in tangential windspeed within the eye but surprisingly a rise in MSLP rather than a fall. The reason for this deviation is uncertain. Nevertheless, a plausible explanation is that the non–local system wide weakening of the TC induced a degradation of the warm core manifest as a cooling, above the boundary layer, that became less significant with height. The degradation of the warm core is also partially responsible for the transient PV bridge structure[4] seen in Fig 6c,h,m,r

just below 2 km height. By Stokes theorem (Haynes and McIntyre, 1987) the decrease in tangential wind within the eyewall should be associated with a concomitant outward transport of PV which is attributed to the outward motion of the VRWs having a deleterious effect on the azimuthally averaged tangential wind without the presence of sufficient diabatic forcing to maintain the strong narrow vorticity annulus structure (e.g. Kuo et al., 1999; Williams, 2017).

This process seems to be similar to that described in the observational study of Kossin and Eastin (2001) where intensifying
TCs could transition from a state with high vorticity and angular velocity in the eyewall (compared to the eye) to a state where the vorticity and angular velocity was similar in the eyewall and the eye. This second regime was similar to our weakening phase and associated with weakening tangential velocity within the eyewall. It is also noteworthy that prior to this second regime the eyewall was observed to become more elliptical which we have also observed in this model study. An important difference compared to Kossin and Eastin (2001) is that in our case the vorticity structure never becomes truly monopolar.
During the weakening phase the isolated regions of deep rotating convection are less favoured to form, as the TC is likely in a barotropically stable state, which results in a more azimuthally symmetric structure.

The dynamical effect of the isolated regions of rotating deep convection may be an important element in the transition between the strengthening and weakening phases. During the strengthening phases these isolated regions of deep rotating convection are not harmful to the storm's intensification and can, through the eddy radial vorticity flux (term 3 on the right
hand side of equation 4), contribute to the intensification of the azimuthally averaged tangential wind above the boundary layer. However, isolated regions of deep rotating convection that move too far inwards or outwards can have a disruptive effect and trigger a weakening phase by decelerating the tangential wind within the eyewall. These results seem surprising given that Kilroy and Smith (2016) suggest that an updraft in a vortex results in an increased contribution to the tangential wind budget radially outwards of the updraft and a negative contribution radially inwards of the updraft (with more pronounced effects
from updraughts further from the storm centre). Isolated regions of deep rotating convection often appear in a TC just prior to its rapid intensification phase, such as in Guimond et al. (2010), where their appearance precedes the rapid strengthening and increased azimuthal symmetry of the storm. Although the isolated regions of deep rotating convection in Hurricane Irma do precede a more azimuthally symmetric state of the storm, this is typically during a weakening phase. This difference, on the storm's intensification, between the impact of isolated regions of deep rotating convection in this study and prior to rapid
intensification such as in Guimond et al. (2010) suggests that isolated regions of deep rotating convection may have different

[4]Plotting the relative vorticity rather than the PV shows a similar tendency as in Fig.6 or Fig.7a with the radial gradient of vorticity between the eyewall rapidly decreasing during the middle of the weakening phases but not to the extent that a 'bridge' or monopole structure forms. Hence, the change in the PV distribution between the strengthening phase and the weakening phase is linked to thermodynamic structural changes as well as proposed barotropic stability changes.

impacts on a mature storm undergoing rapid intensification compared to a much weaker storm that has not yet undergone rapid intensification.

The weakening that occurs in the tangential wind above the boundary layer is accompanied by a decrease in the PGF which is also transmitted through to the boundary layer. This decrease in the PGF (Fig. 4 a,c) is responsible for the increased agradient flow in the boundary layer. The emergent supergradient wind (Fig. 4 a green), erupting from the boundary layer then returns to gradient wind balance through an increased boundary layer outflow jet Fig. 4d yellow). It is likely that the unbalanced spin–up mechanism as described by Smith et al. (2009) prevents the tangential wind from weakening significantly within the boundary layer while frictionally induced inflow remains strong. The weakening that does occur happens 1-2 hours later than the weakening above the boundary layer suggesting that the instant decrease in the PGF is not wholly or directly responsible for this later weakening.

A potentially similar kind of intensity fluctuation explored in Smith et al. (2021) may highlight another reason for the strengthening outflow jet above the boundary layer. In their simulation a brief decay phase is brought on by a temporary disruption to the eyewall by a rainband complex structure. The eyewall is then unable to fully ventilate the mass flux in the boundary layer and, as such, flow above the boundary layer increases and enhances the rain–band structure. It is likely a similar phenomenon is occurring here. The azimuthally averaged diabatic heating within the initial eyewall region has weakened as a result of the reorganisation of the convection by the isolated regions of rotating deep convection which manifests, in the azimuthally averaged sense, of a greater radial spread of the heating and a comparative weakening of the heating within the eyewall region. Another possible reason for the weakening eyewall convection is entrainment from downdraughts caused by surviving isolated regions of rotating deep convection at higher radii. Either way, as a result the ventilation index becomes more negative and more mass originating from the boundary layer is evacuated by the strengthening outflow jet, further weakening the tangential wind above the boundary layer through outward advection of low AAM air. Examining the structure of Fig. 4b shows that the weakening above the boundary layer is composed of an initial modest weakening 1-2 hours prior to the surface defined weakening phase followed by a more dramatic weakening which could be attributed to spin–down from unventilated air originating in the boundary layer. Examination of a boundary layer slab model (not shown) driven by the model tangential wind field above the boundary layer was able to capture features of the intensity fluctuations including the increasing agradient wind indicating that the boundary layer control mechanism is at least partially responsible for the strengthening outflow jet. Nevertheless the enhanced outflow jet was also replicated (qualitatively) in a balanced model that solved the Sawyer Eliassen equations indicating that 'suction', above and independent of the boundary layer could also explain the strengthening outflow. These three effects; the boundary layer control, the supergradient wind returning to gradient balance and the effects of the convection itself cannot easily be separated.

The fluctuations presented here in Hurricane Irma do show similarities to vacillation cycles, particularly with the simulation conducted in Reif et al. (2014) which exhibits transitions from ring–like to monopolar PV distributions but with a more ring–like state than Nguyen et al. (2011). One significant difference compared to the vacillation cycles in Hardy et al. (2021) is that, in Hurricane Irma, the more monopolar state during the weakening phases were transient with $\mathrm{PV_0/PV_{max}}$ peaking at the end of the weakening phase before dropping rapidly. The role of barotropic and convective instability does also seem to play a

role. However, the azimuthally asymmetric periods dominated by isolated regions of deep rotating convection (for example in Nguyen et al., 2011) are not explicitly linked to strengthening phases as they are in this study. Fischer et al. (2020) did identify these fluctuations in the observational data of Hurricane Irma and described them as two separate eyewall replacement cycles triggered by lower–tropospheric convergence associated with a rainband and lower–tropospheric convergence associated with a super–gradient flow respectively. The fluctuations modelled here have some similarities with the second mechanism proposed in Fischer et al. (2020) with the secondary eyewall merging with the primary eyewall before dissipating.

The intensity fluctuations in Irma also have some similarities to a 'partial eyewall replacement cycle' described in Zhang et al. (2017) where the boundary layer updraft is unable to properly couple with a potential secondary updraft above. It is proposed that the fluctuations here are the result of the eyewall being temporarily disrupted by isolated regions of deep rotating convection in the inner rainbands and the resultant disruption of the coupling between the boundary layer and the free troposphere and the eventual reformation of the coherent eyewall structure. Unlike an eyewall replacement cycle there is no clear secondary eyewall formation event. In summary, the fluctuations we have seen bear many similarities to other phenomena analysed in previous work but no single study provides the complete picture; the fluctuations are likely due to a sequence of events linked to dynamics occurring at the interface of the boundary layer within the eyewall.

## 5   Composites over multiple ensemble forecasts

The prior analysis has been carried out for one ensemble forecast. To demonstrate the robustness of the analysis composites of selected key results will be presented across multiple ensemble members. Five out of 18 ensemble members (including ensemble member 15), initialized on 03 September 00 UTC, showed the intensity fluctuations previously discussed. A further six ensembles also showed similar but weaker fluctuations. An additional model simulation, initialized on 02 September 12 UTC, found seven out of 18 ensemble members with the same kind of fluctuations. The following composites are based on the five ensemble members initialised on the 03 September at 00 UTC that show the strongest fluctuations. The composites are over all weakening and strengthening phases in all of these five ensemble forecasts. These weakening and strengthening phases vary in length from one hour to 10 hours, with 4–5 hours being typical. There are a total of 45 weakening and strengthening phases averaged over.

One of the key aspects of the analysis is the transition during weakening phases from a ring–like PV distribution at the start of the weakening phase towards a more bridge like PV distribution towards the end of the weakening phase. Figure 17 shows a PV tendency composite plot for all weakening and strengthening phases for the five ensemble members with the strongest intensity fluctuations. During the weakening phases there is a positive PV tendency within the inner eye and a negative tendency within the high PV annulus confirming the results from Section 4.2 for Irma's PV structure to become more monopolar in the weakening phases. The opposite is shown in the strengthening phases with PV decreasing in the inner eye and rising in the high PV annulus. Near the RMW outside the PV ring there are positive PV tendencies at the end of the weakening phases which can also be seen in Fig. 5e,j,o,t to Fig. 5d,i,n,s which show, from left to right, the structural PV changes that occur from the start of

the weakening phases to the start of the strengthening phases in ensemble member 15. Near the RMW (dashed black line), PV starts to increase at the end of the weakening phases and into the start of the next strengthening phase.

Figure 18 shows the contributions to the tangential wind budget through mean and eddy advection of angular momentum of the strengthening composite relative to the weakening composite (strengthening phases minus weakening phases). Above the boundary layer at a radial distance of 20 km to 35 km the eddy term plays a beneficial role in both the strengthening and weakening phases; however, in the strengthening phases the effect is distinctly greater. This comparison confirms some of the findings shown in Fig. 10 that the eddy momentum flux acts to cause intensification above the boundary layer particularly during strengthening phases. The effect of the mean momentum fluxes are also similar with greater tangential wind spin–up in the boundary layer in strengthening phases compared to weakening phases but also with greater spin–down above the boundary layer in the outflow jet during the strengthening phases.

The composites demonstrate that similar processes are likely occurring in the other ensemble members. The fluctuations in intensity that occurred during rapid intensification are not just limited to a single ensemble member. This study focuses on a single case, Hurricane Irma (2017), so it is unclear how common this type of intensity fluctuation is in TCs. The ensemble forecasts showed no link between the likelihood of the intensity fluctuations and the environmental conditions so the causes of the fluctuations are likely stochastic in nature (in particular with respect to the radial location of isolated regions of deep rotating convection that develop). The fluctuations are shown to occur in around a third of the ensemble forecasts suggesting they may be a common feature in rapid intensification and motivating analysis of more cases.

## 6 Summary and Conclusions

The main aim of this study was to determine the cause of the observed intensity fluctuations in Hurricane Irma (2017), during rapid intensification, and to identify the processes responsible. Understanding these fluctuations is important as they can affect both the intensity and size of the RMW in the short–term and therefore the destructive potential of the TC. Although the intensity fluctuations have been observed at the surface (see Fig. 2) the biggest structural changes seen in the model simulations occurred just above the boundary layer at around 1500 m height and preceded the changes at the surface by a couple of hours. Hence, most of the results and discussion focus on the height just above the boundary layer and the interface between the boundary layer and the free vortex above. Further details about these definitions can be found in section 4.1.

A summary of the key findings and interpretations is:

– In Hurricane Irma, during the second period of rapid intensification, the focus of this study, intensity fluctuations occurred, defined as short–term intensification and weakening periods at 10 m height. During the weakening phases, MSLP increased, 10–m total wind speed decreased or remained constant and the 10 m RMW increased. In contrast during strengthening phases, MSLP decreased, 10–m total wind speed increased and the 10 m RMW decreased.

– Isolated regions of rotating deep convection form stochastically during the strengthening phases (Fig. 19a,b ). During the course of the strengthening phases the reorganisation of the initially ring–like eyewall convection into patches of

isolated rotating deep convection occurs. The isolated regions of deep rotating convection are stochastic in nature (some ensembles do not produce them) but the arrangement of convection into these isolated regions is more likely during the strengthening phase compared to the weakening phase plausibly due to the onset of barotropic instability (which is implied by the satisfaction of Rayleigh–Kuo criterion early in the strengthening phase at the 1500 m level (Fig. 6c)). The growth of wavenumber–2 PV anomalies, associated with the barotropic instability, is plausibly capable of enhancing convection through an Ekman pumping from the boundary layer where the pertubation vorticity is high by providing a favoured location for reducing convective inhabitation. The addition of convective instabilities greatly enhances the rate of growth than would otherwise be possible in a purely barotropic framework.

- The effect of the isolated regions of rotating deep convection is to initially spin–up the azimuthally averaged tangential wind in the eyewall region, above the boundary layer, with eddy vorticity flux and eddy vertical advection (terms 3 and 4 on the right hand side of equation 4 ) more than compensating for the spin down effect from mean radial vorticity flux and mean vertical advection of absolute angular momentum (see Fig. 10 and 11).

- During the strengthening phases, especially at 1500 m height, the radial PV distribution is an elongated ring (i.e. more eccentric, see Fig. 7c ). In addition, the diabatic heating distribution had a small radial spread (Fig. 12 ) and a strong heating maximum located within the RMW.

- At the start of the weakening phase (Fig. 19c,d), the deep rotating convective structures have become stronger (in terms of diagnosed local vertical velocity and vorticity) and more radially widespread. These structures move outwards, and retrograde to the tangential flow with outward propagating vortex Rossby waves. The isolated regions of deep convection appear to be constrained by the dispersion relation of vortex Rossby waves with convection strongly coupled to the PV anomalies. The isolated regions of deep convection are present at regions outside the eyewall at the start of the strengthening phase where vortex Rossby wave activity is largest. The radial location of the convection is important and influences where spin–down of the azimuthally averaged tangential wind above the boundary layer occurs. The radial eddy vorticity flux becomes negative near the maximum tangential wind radius at this time largely on account of positive perturbation radial velocity associated with the outward evolving regions of deep convection, which act to draw moist air into the attending updraughts.

- By the middle of the weakening phase (Fig. 19e,f), a change in heating structure is apparent, with isolated deep rotating convective structures forming outside the eyewall showing as a secondary heating column when azimuthally averaged (Fig. 12). The largest (positive) change in the strength of the outflow jet above the boundary layer occurs about 2 hours after the start of the weakening phase (Fig. 16). The increase in strength of the boundary layer occurs at the same time as a decrease in the ventilation index. This juxtaposition of flow features suggests that the particularly rapid decrease in tangential wind, above the boundary layer may be linked to the inability of the deep convection within the eyewall to ventilate the mass inflow from the boundary layer. We recognize it is difficult to separate this inadequate ventilation

effect from the suction of the outer convection or the balanced outflow at higher radii (which has been shown to increase in a balanced model).

The decrease in the azimuthally averaged tangential wind, above the boundary layer, can be linked to the strengthening outflow jet through the radial vorticity flux from the eye (region highlighted by the yellow circle in Fig. 15b,d).

The outflow at this time is also enhanced by a supergradient wind within the boundary layer leading to a positive agradient force near the RMW. Radially outside the eyewall, frictionally induced inflow is still strong; the continued mass influx from larger radii minimized the weakening in the boundary layer due to the unbalanced spin–up mechanism.

- During the entire weakening phase, the storm centre MSLP rises nearly concurrently with the weakening of the maximum 10– m total wind. This finding is the opposite of Nguyen et al. (2011) where the weakening of the wind is accompanied by a MSLP drop. In Nguyen et al. (2011) the fall in pressure in their asymmetric phases (comparable to our weakening phase) is linked to PV imported into the eye.There is an import of PV into the eye in Hurricane Irma (Fig. 6) during the weakening phases but as an apparent upward moving PV anomaly. Within the eye there is a transient increase in tangential wind speed associated with the inward mixing of cyclonic vorticity. The increased tangential wind speed within the eye not being associated with a MSLP drop could be linked to a system wide weakening degradation of the TC warm core during the weakening phases which is also likely partially responsible for the PV bridge structure that forms.

- The tangential wind above the boundary layer eventually starts to increase again (Fig. 19g,h ) as convection grows in a region radially outwards from the original eyewall aided by convergence from the suction effect of the convection and a possible balanced inflow away from the original eyewall. The restoration of the eyewall structure and strengthening of the diabatic heating gradients allow the wavenumber–2 barotropic instability to begin to grow again.

In conclusion, the findings from this analysis, as summarized in Fig. 19, show the proposed mechanism for the intensity fluctuations observed in Hurricane Irma, and highlight the importance of both the isolated regions of deep convection that develop on the intersection of inner rainbands with the eye–wall and of the development of the supergradient wind within the boundary layer. It was found that these intensity fluctuations appear in about 1/3 of the ensemble simulations. No link was found between the environment of the storms and the presence of these intensity fluctuations indicating stochastic processes are involved. In addition, the intensity of the storms at the end of the simulations with intensity fluctuations were similar to those without, indicating that the increased intensification rates during strengthening phases compensated for the weakening phases. This study gives potentially further insight into intensity fluctuations during rapid intensification, such as the vacillation cycles in Nguyen et al. (2011), and emphasises the role of the inner rainbands in causing weakening periods. The study also offers an explanation for the observed intensity fluctuations in Hurricane Irma shown in Fischer et al. (2020). A future direction of this work would be to investigate the similarities between these fluctuations in rapid intensification and eyewall replacement cycles and to determine whether they are caused by similar processes and to analyse more cases to assess to what extent these results can be generalized.

*Data availability.* Observational data used in this paper is made available online by the Hurricane Research Division and is available at https://www.aoml.noaa.gov/hrd/Storm_pages/irma2017/. The microwave data is made available online by CIMSS at http://tropic.ssec.wisc.edu/real-time/mimtc/2017_11L/web/mainpage.html. The model fields in a 200km box around the storm which are used for the analysis in this paper have been stored and can be made available on request.

## Appendix A: Calculation of agradient wind

The agradient wind is determined by taking the gradient wind balance, where the pressure force is balanced by the sum of the Coriolis and centrifugal forces: $\frac{1}{\rho}\frac{\partial p}{\partial r} = \frac{v_g^2}{r} + fr$, where $\rho$ is the dry density, $p$ the pressure, $v_g$ is the gradient wind, $f$ the Coriolis parameter and $r$ the radial distance from the centre. Substituting in the ideal gas law: $p = \rho RT$, where $R$ the ideal gas constant and $T$ the temperature, and then noting that the agradient wind is given by the deviation of the tangential wind from the balanced tangential wind: $v_{ag} = v - v_g$ where $v_{ag}$ is the agradient wind we arrive at,

$$850 \quad v_{ag} = v - \frac{1}{2}\left(\sqrt{\frac{4RrT}{p}\frac{\partial p}{\partial r} + f^2 r^2} - fr\right). \tag{A1}$$

Physically the agradient wind represents the deviation of the primary circulation from gradient wind balance. A subgradient wind means the wind speed is lower than the gradient wind, while a supergradient wind is higher than the gradient wind. In the boundary layer, both subgradient and supergradient winds are often found. At the surface friction reduces the tangential wind and causes it to be subgradient but the frictionally induced inflow can also lead to tangential acceleration at higher levels and

855 smaller radii which sometimes results in a supergradient layer.

*Author contributions.* **William Torgerson**: Conceptualization, Formal Analysis, Investigation, Methodology, Software, Writing – Original Draft Preparation. **Juliane Schwendike**: Conceptualization, Supervision, Writing – Review and Editing. **Andrew Ross**: Conceptualization, Supervision, Writing – Review and Editing. **Chris Short**: Data Curation, Methodology, Supervision, Writing – Review and Editing.

*Competing interests.* There are no competing interests

*Acknowledgements.* We would like to especially thank Roger Smith and Michael Montgomery for devoting large amounts of their time and support in helping to improve rigour and accuracy. We would also like to thank the anonymous reviewer for their helpful suggestions and comments. We thank the Hurricane Research Division for providing dropsonde and flight–level data as well as the images contained in Fig. 3a-d (available online at https://www.aoml.noaa.gov/hrd/). We thank the Cooperative Institute for Meterological Satellite Studies for making the microwave imagery available contained in Fig. 3e-f (available online at http://tropic.ssec.wisc.edu/real-time/mimtc/2017_865 11L/web/mainpage.html). This work used Monsoon2 (the Met Office and NERC supercomputing node) a collaborative high-performance

computing facility funded by the Met Office and the Natural Environment Research Council. Torgerson was funded by a PhD scholarship from the NERC SPHERES DTP (grant NE/L002574/1) and CASE support from the Met Office. This research was also partially funded by the Met Office Weather and Climate Science for Service Partnership (WCSSP) Southeast Asia, as part of the Newton Fund.

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

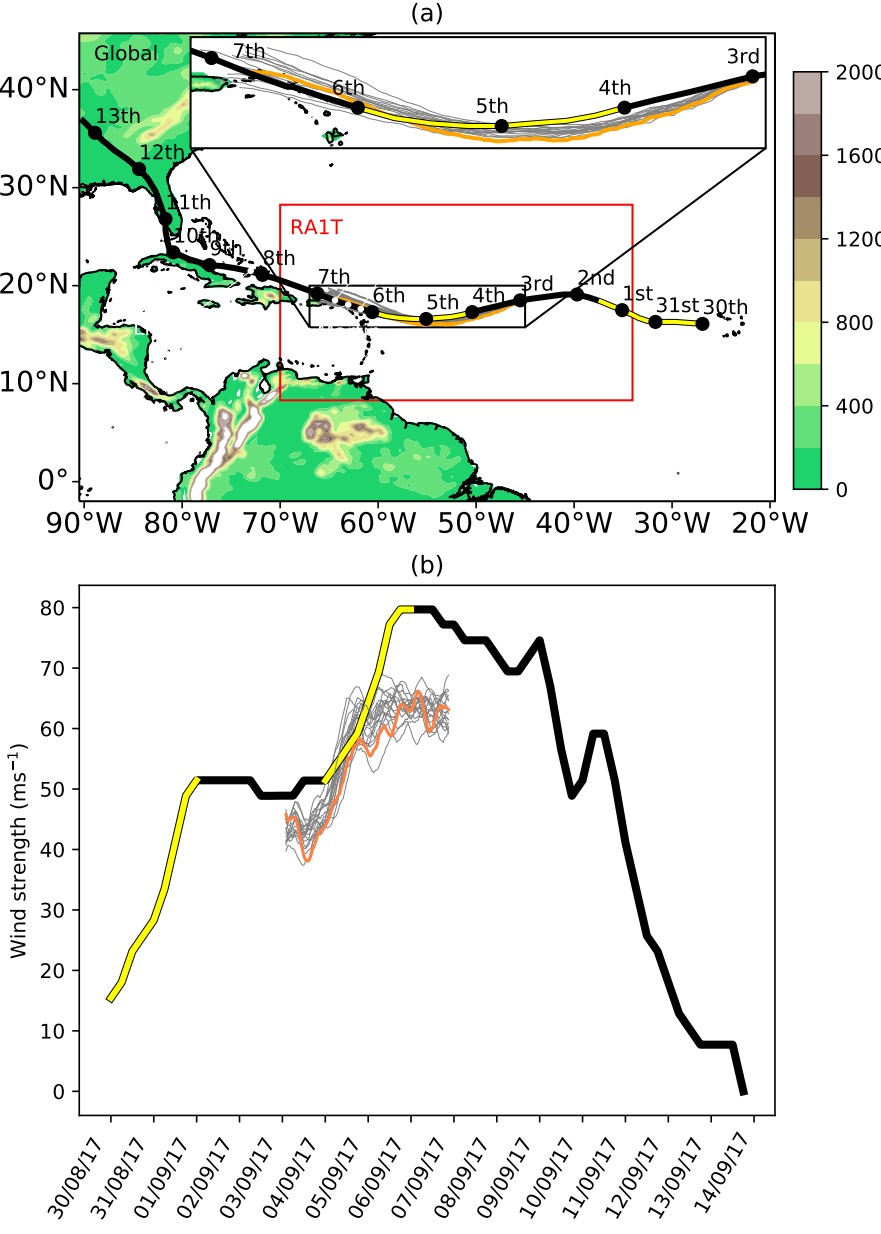

**Figure 1.** (a) Best track of Hurricane Irma (black line) with points corresponding to the position of Irma on each date from 30 August 2017 to 13 September 2017. Orography (m) is shown in shading. The domain of the regional model used in this study is shown by the red rectangle. The 18–ensemble member tracks are displayed in grey with ensemble member 15 shown in orange. Islands where landfall occurred are indicated by white dots and labels. (b) The best track wind speed (black), the maximun surface wind speed of the ensemble members initialised on 03 September 00 UTC (grey contours) with ensemble member 15 highlighted in orange. In both panels periods of rapid intensification are highlighted in yellow.

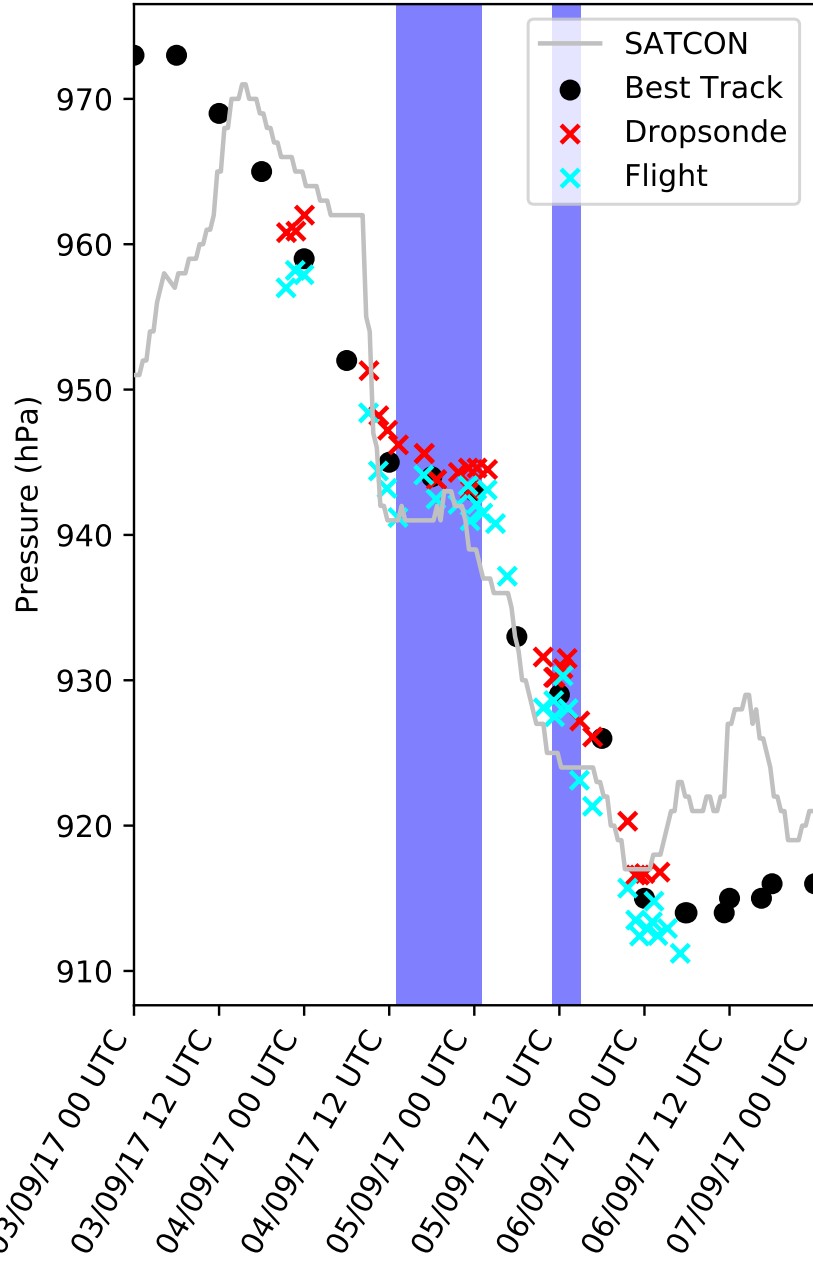

**Figure 2.** Observed minimum sea level pressure as a function of time based on SATCON and NHC forecaster assessed Best Track estimates as well as direct dropsonde and flight measurements. The 96–hour period shown is the same as the simulation initialized on 03 September 00 UTC. Two notable weakening/stagnation periods during the period of rapid intensification are highlighted by the blue bands.

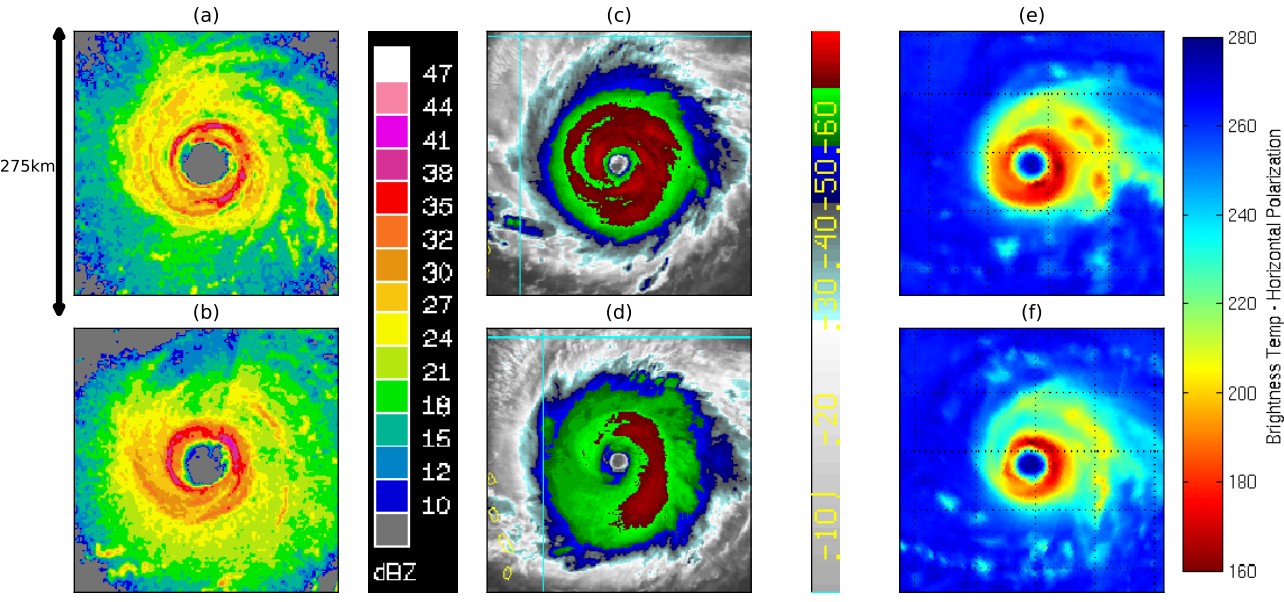

**Figure 3.** NOAA P3 flight–level radar (in dBZ) on (a) 05 September 0943 UTC and (b) 05 September 1232 UTC, colour enhanced infrared (IR) imagery (in °C) on (c) 05 September 0945 UTC and (d) 05 September 1245 UTC, and MIMIC microwave imagery (brightness temperature in K) for (e) 05 September 0945 UTC, (f) 05 September 1245 UTC. The upper and lower rows correspond to times just before and after the start of the period indicated by the second blue bar in Fig 2.

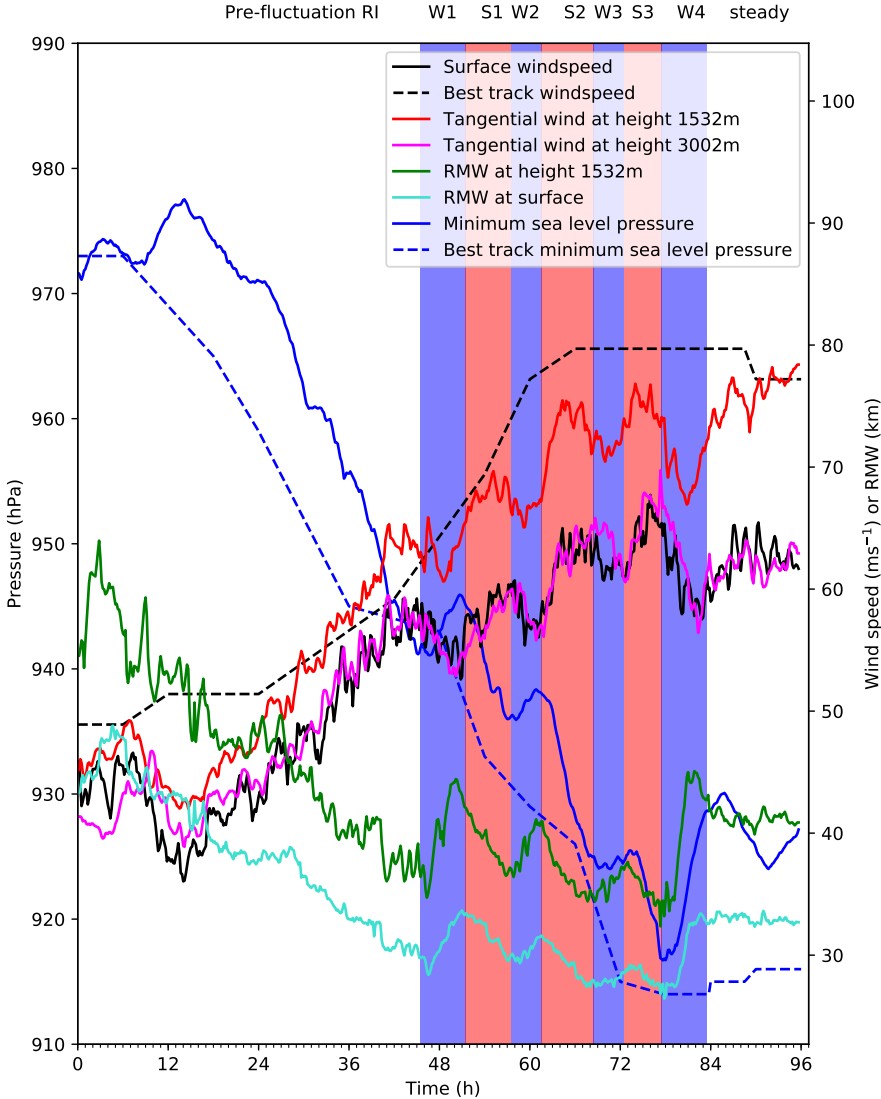

**Figure 4.** Various model diagnostics (solid lines) and corresponding observations (dotted lines, where available) as a function of time. Details are given in the legend. Blue bands indicate weakening phases, and red bands indicate strengthening phases during the rapid intensification period. The individual strengthening and weakening phases have been labelled (see top of plot). W stands for 'weakening', S stands for 'strengthening'. Phases have been subjectively identified. The RMW refers to the surface or 1532 m radius of maximum azimuthally–averaged tangential wind speed. The tangential wind is the azimuthally-averaged tangential wind at the RMW.

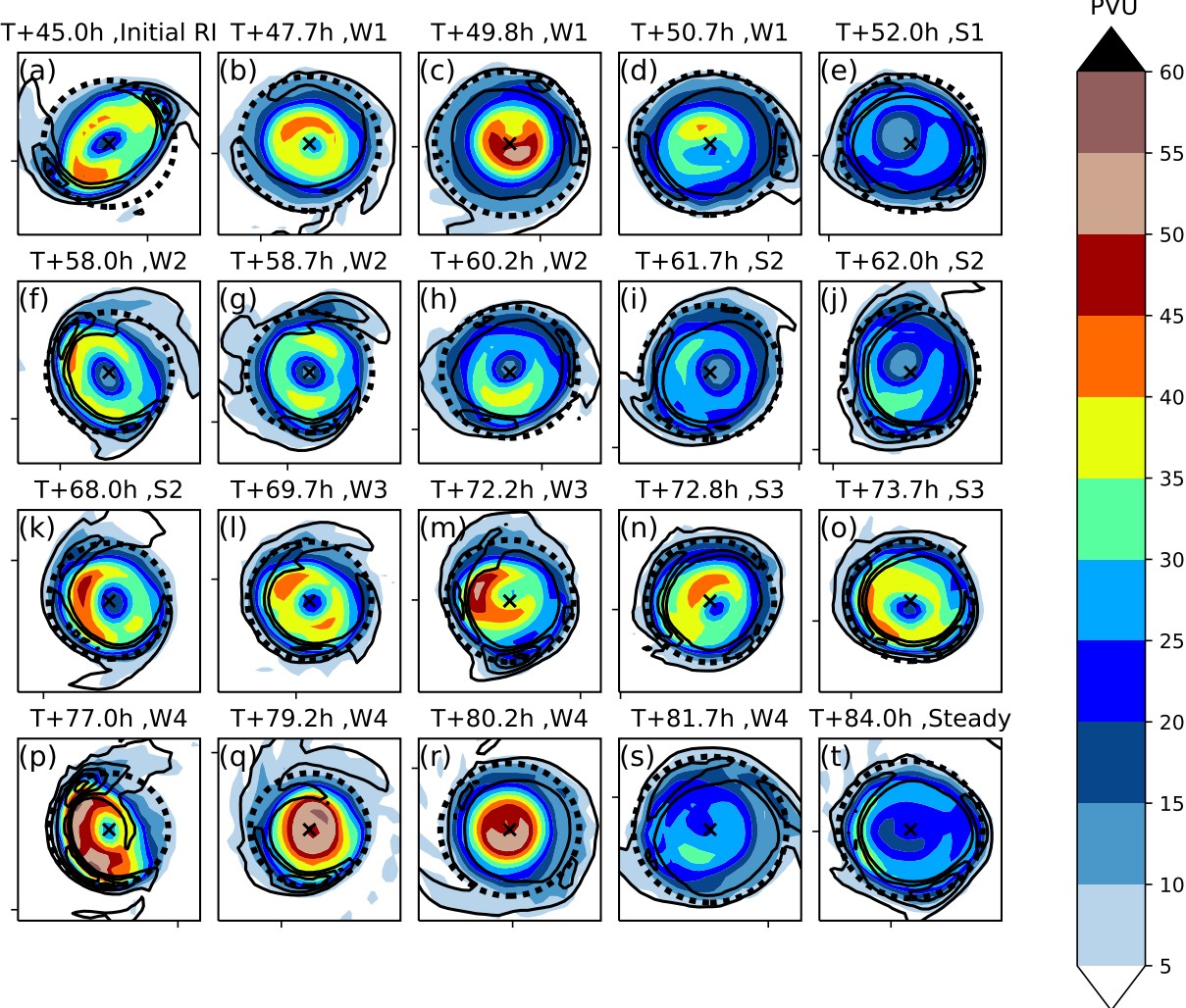

**Figure 5.** PV (PVU, shaded) at 1532 m height for selected times and vertical velocity (1 m s$^{-1}$, black contour). The 1532– m height RMW is indicated by the dashed black line. A cross marks the centre of the TC. The data is output in 10–minute intervals, times are given to the nearest 0.1 hours. The data is from ensemble member 15 which was initialised at 03 September 2017 at 00 UTC.

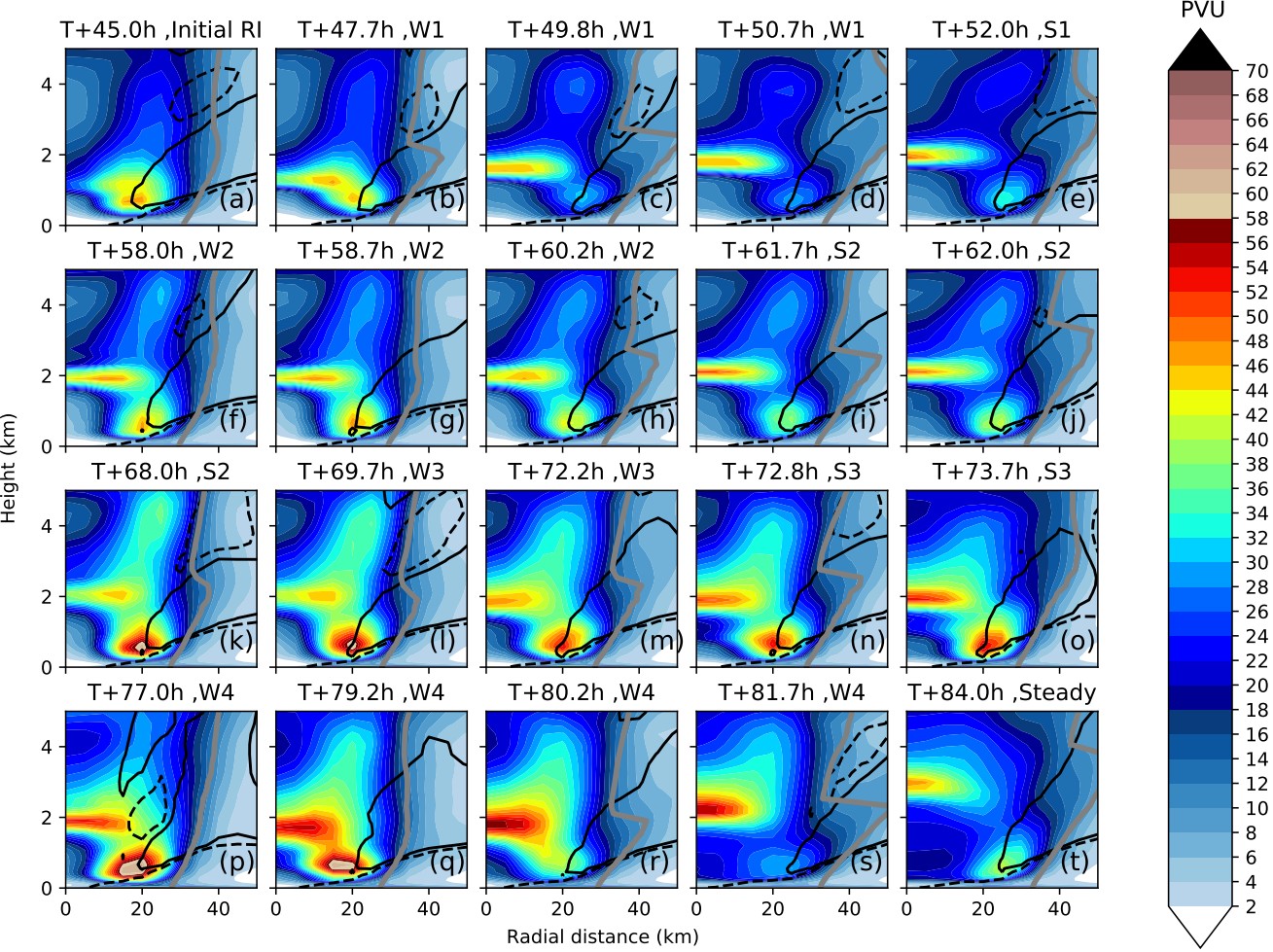

**Figure 6.** Azimuthally averaged PV (PVU, shaded) as a function of radial distance and height for selected times. The height–dependent RMW is indicated by the grey line. Also shown are the 1 $\mathrm{m\,s^{-1}}$ (black line) and -1 $\mathrm{m\,s^{-1}}$ (dashed black line) radial wind contours.

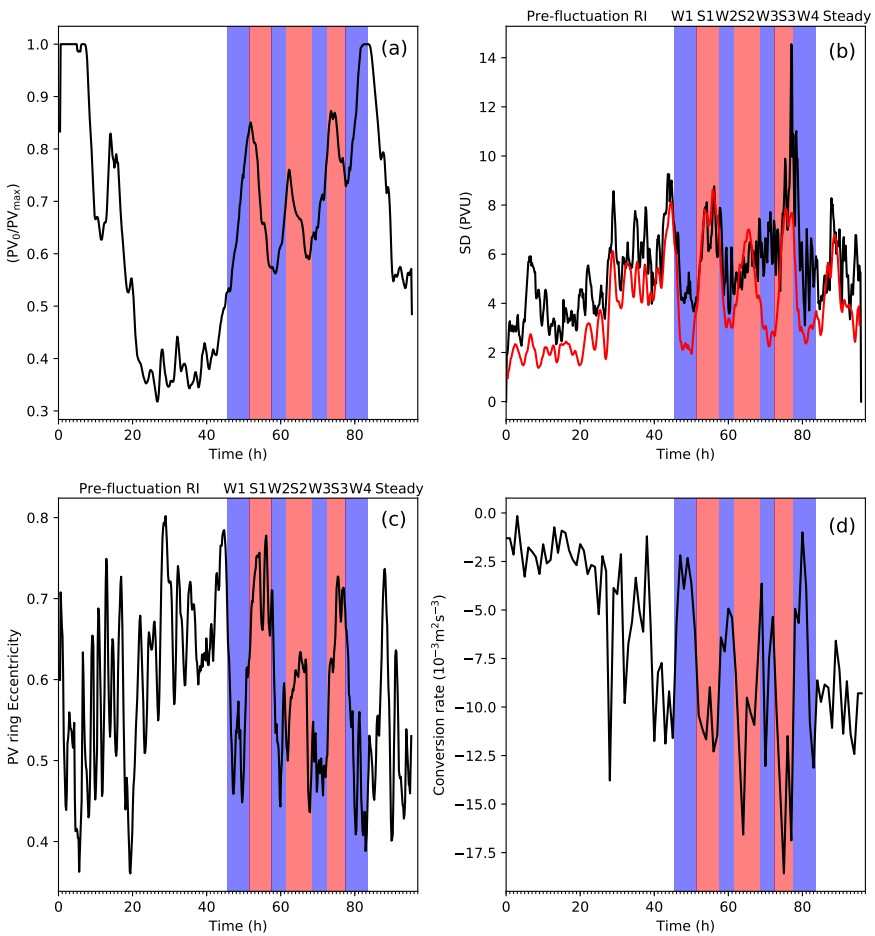

**Figure 7.** (a) Ratio of the low–level PV (depth averaged between 1052 m and 4062 m) at the centre of the TC to the maximum azimuthally averaged low-0-level PV. (b) Maximum standard deviation of PV at 1532 m (black) and standard deviation of PV at the 1532m RMW (red). (c) Eccentricity of the ring fitted to the PV distribution at 1532 m. (d) Average barotropic conversion rate from the surface to 4062 m averaged between 5 km and 70 km as a function of time. To smooth out high frequency noise a 1–h running mean is applied to the 10–minute data. Weakening (blue) and strengthening (red) phases are also shown.

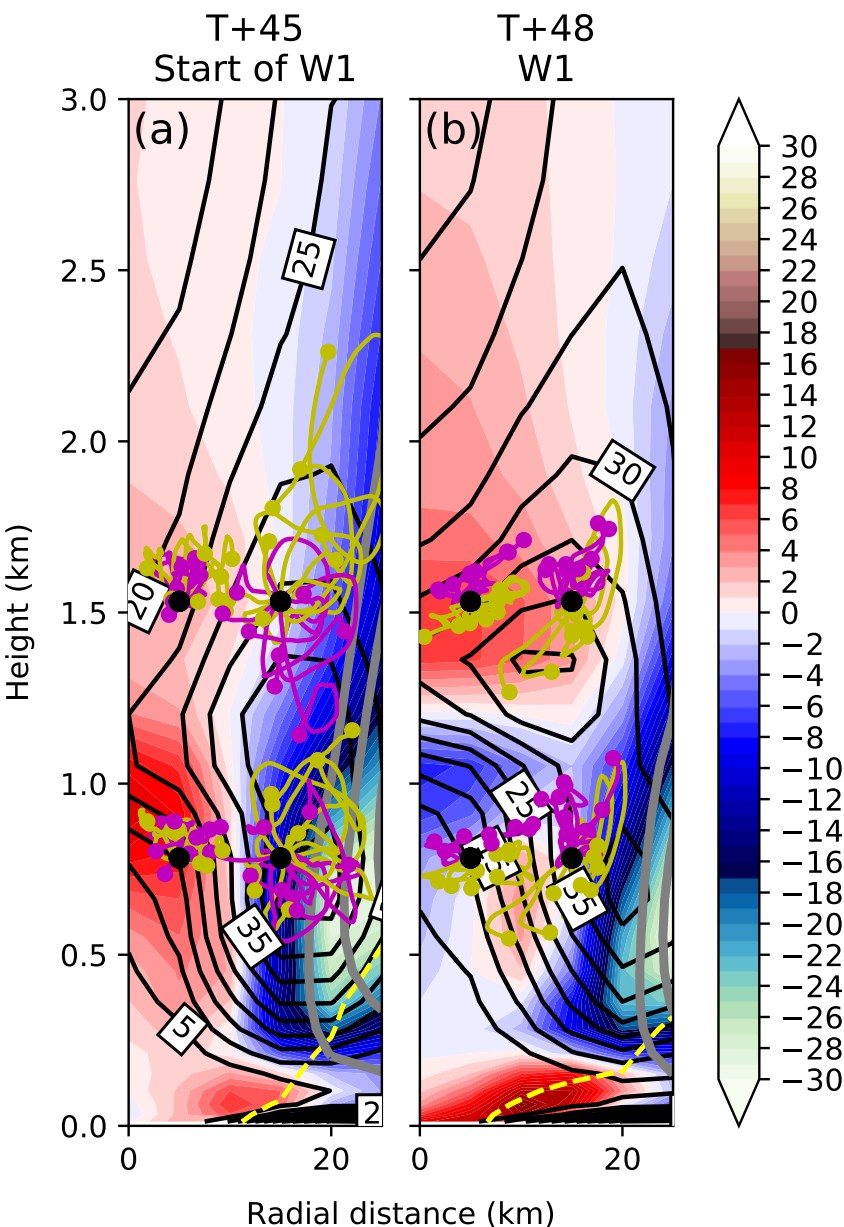

**Figure 8.** Change in PV over the past hour due to advection only (shaded, $\mathrm{PVU\,h^{-1}}$). Black line contours show the PV field in intervals of 5 PVU. Additionally, four sets of trajectories are shown for the following (r,z) points (black scatter points): (5 km, 1532 m), (15 km, 1532 m), (5 km, 782 m), and (15 km, 782 m). Purple lines and scatter points represent the forward trajectory over the next hour while mustard lines and scatter points represent the backward trajectory over the previous hour. Each set of trajectories contains 8 points going back or forward with the same radial distance from the storm centre but with different azimuthal angles around the storm centre: to the east, northeast, north, northwest, west, southwest, south and southeast of the storm centre. The grey contours show vertical velocity (ascent) in $0.25\ \mathrm{m\,s^{-1}}$ intervals indicating the location of the inner eyewall. Yellow dashed line shows the $-1\ \mathrm{m\,s^{-1}}$ inflow contour.

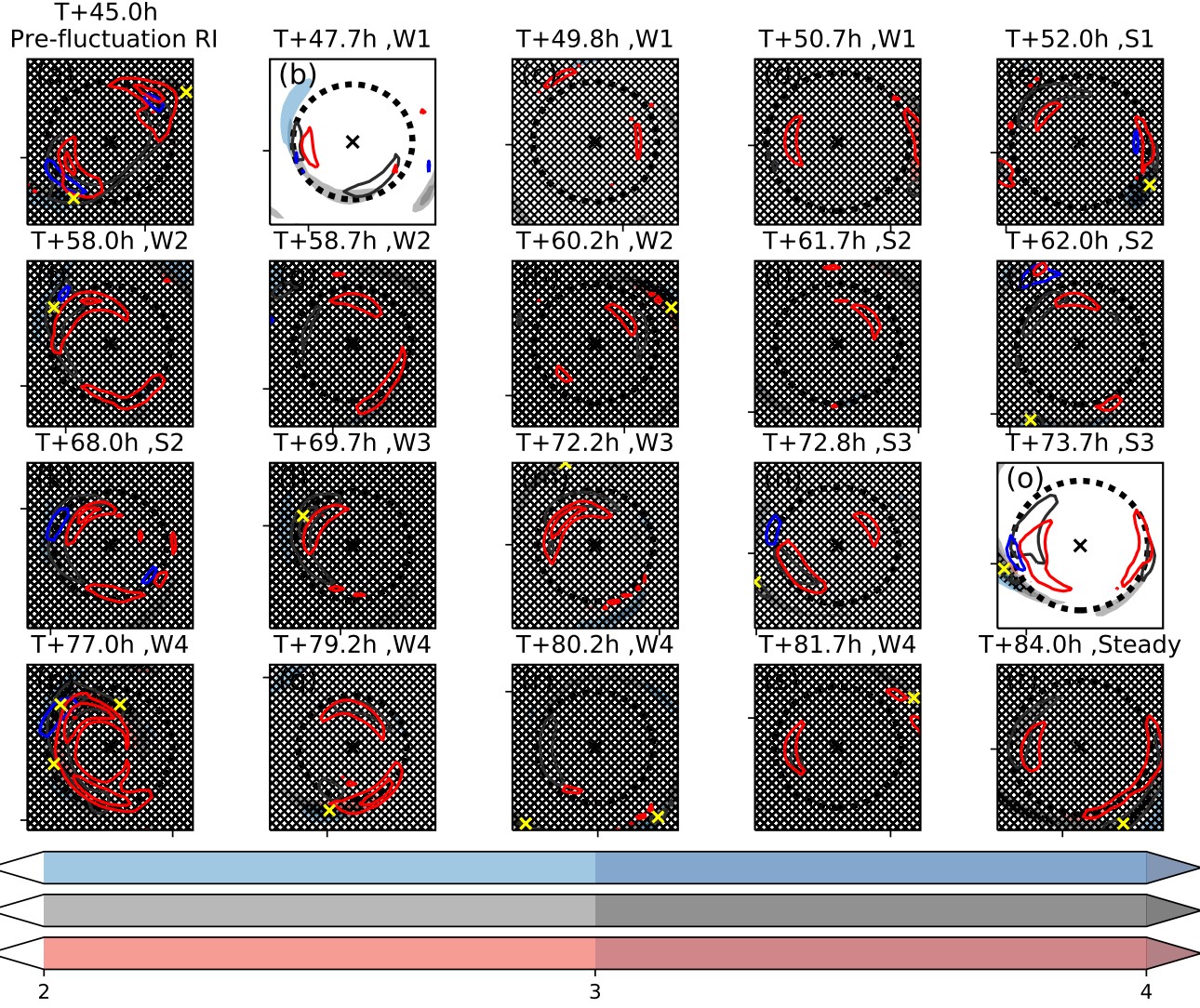

**Figure 9.** Pertubation vertical velocity (m s$^{-1}$, shaded relative to the azimuthal mean), perturbation relative vorticity ($10^{-3}$ s$^{-1}$, coloured line contours) shown at the same times as in Fig. 5. Heights shown are 2532 m for the red shades/lines, 4963 m for the grey shades/lines, and 9934 m for the blue shades/lines. The centre of the TC is denoted by the cross and the RMW at 4963 m is indicated by the black dashed line. Black hatches represent regions where the maximum perturbation vertical velocity at any level exceeds 5 m s$^{-1}$. Yellow crosses show the locations of locally high pertubation relative vorticity at 4963 m to indicate the location of isolated regions of deep rotating convection.

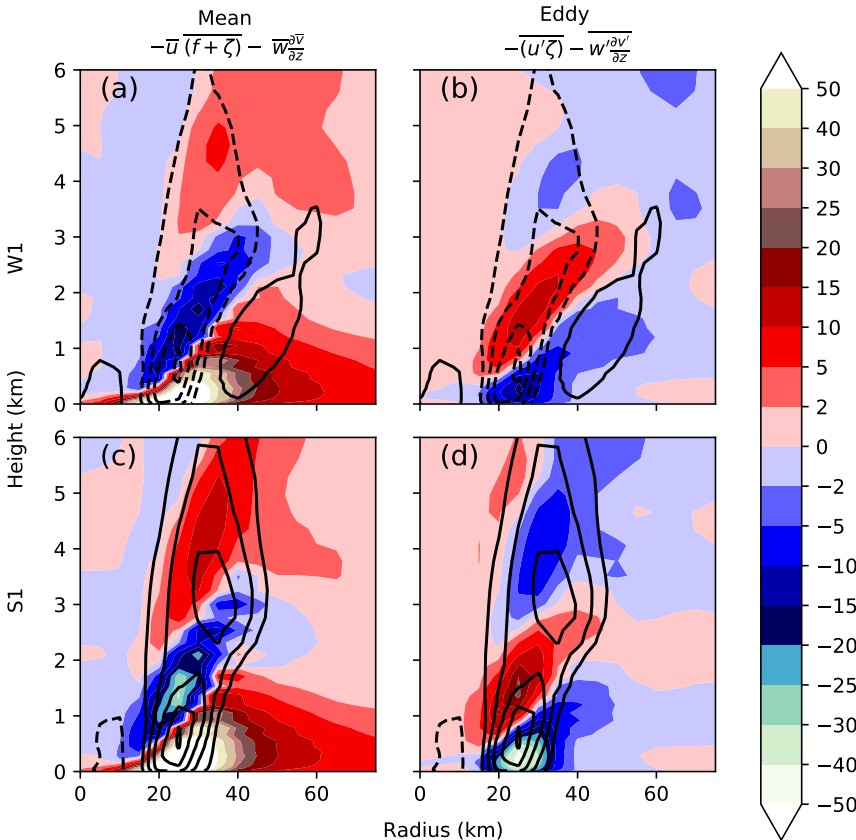

**Figure 10.** Colour shading shows the (a,c) mean and (b,d) eddy contributions to the tangential wind budget (see equation 4) in $\mathrm{m\,s^{-1}\,h^{-1}}$. Line contours show the average tangential wind tendency in $2\ \mathrm{m\,s^{-1}\,h^{-1}}$ intervals with dashed contours indicating negative tendencies. The top row shows the composite for W1 (every 10 minute output in the W1 phase averaged over) while the bottom row shows the composite for S1 (every 10 minute output in the S1 phase averaged over). The frictional term (not shown) also contributes a large negative tangential tendency in the boundary layer.

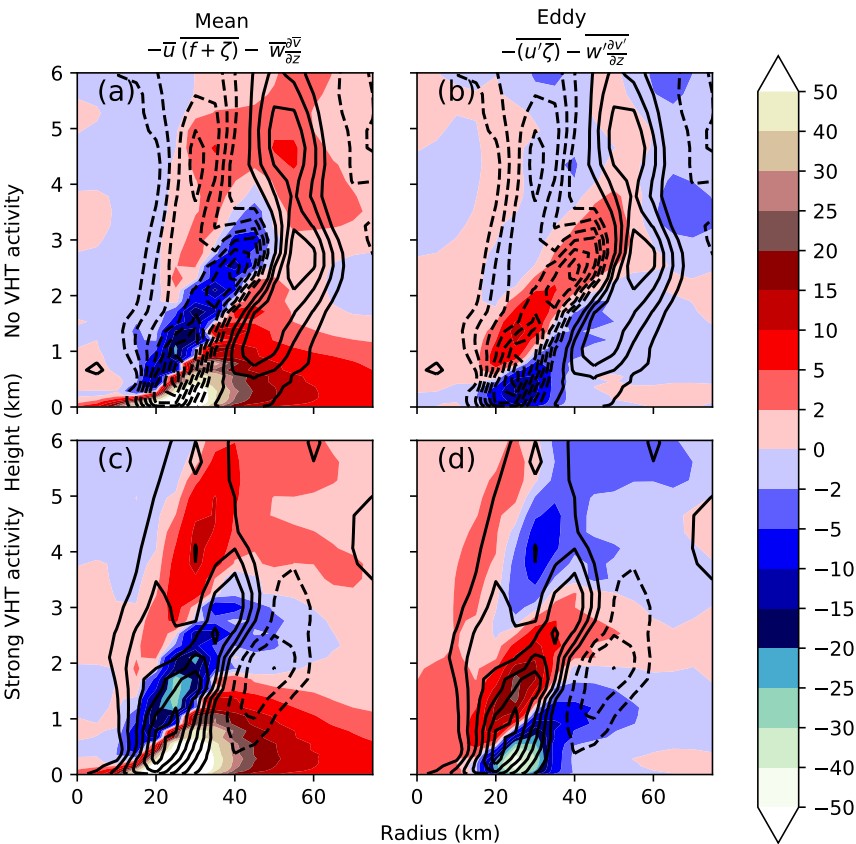

**Figure 11.** As with Fig. 10 but this time composites of times with few isolated regions of deep rotating convection (top row) and many strong isolated regions of deep rotating convection (at least one isolated region of deep rotating convection with perturbation vertical velocity above $2\,\mathrm{m\,s^{-1}}$ and perturbation relative vorticity above $10^3\,\mathrm{s^{-1}}$ at all three levels as in Fig. 9; bottom row). Composites are created by averaging any times in the W1 and S1 combined period with no distinction between weakening and strengthening periods (45.5 hours to 57.5 hours) that either have no isolated regions of deep rotating convection (top row) or many strong isolated regions of deep rotating convection (bottom row).

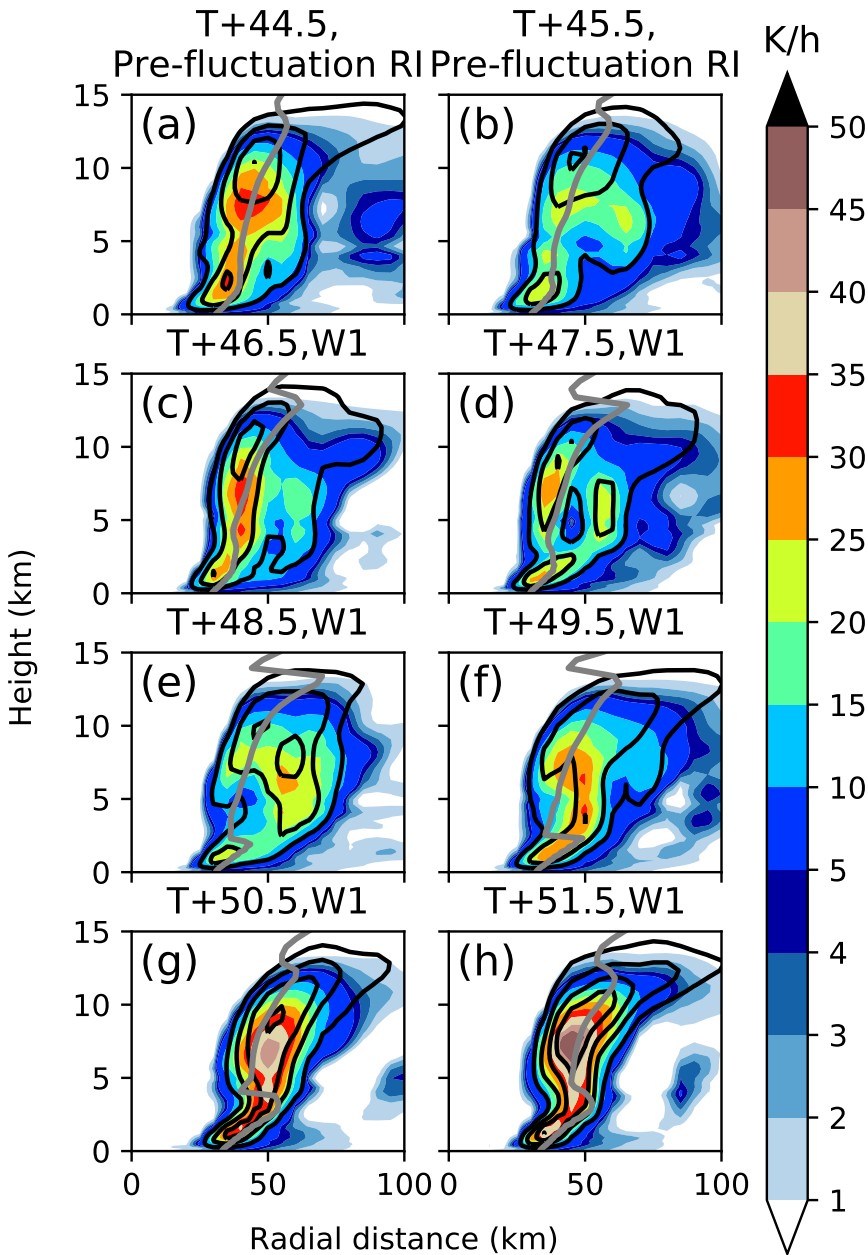

**Figure 12.** Diabatic heating (shading, $\mathrm{K\,h^{-1}}$), vertical velocity (line contours) in intervals of $0.5 \mathrm{\ m\,s^{-1}}$ before and during the first weakening phase W1. Also shown as a grey line is the height dependent RMW.

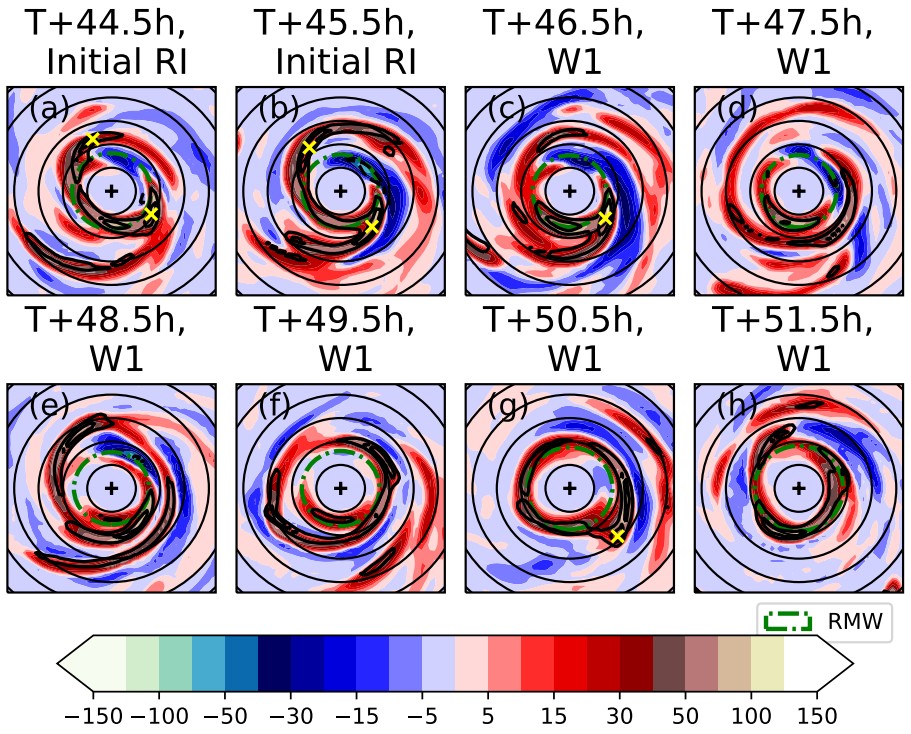

**Figure 13.** Diabatic heating ($\mathrm{K\,h^{-1}}$ shading) for height 4963 m before and during the first weakening phase W1. Vertical velocity contours in intervals of $2\,\mathrm{m\,s^{-1}}$. Yellow crosses indicate the location of the maximun local pertubation vertical velocity at the same level for any isolated regions of deep rotating convection as determined by criteria adapted fromn Smith and Eastin (2010).

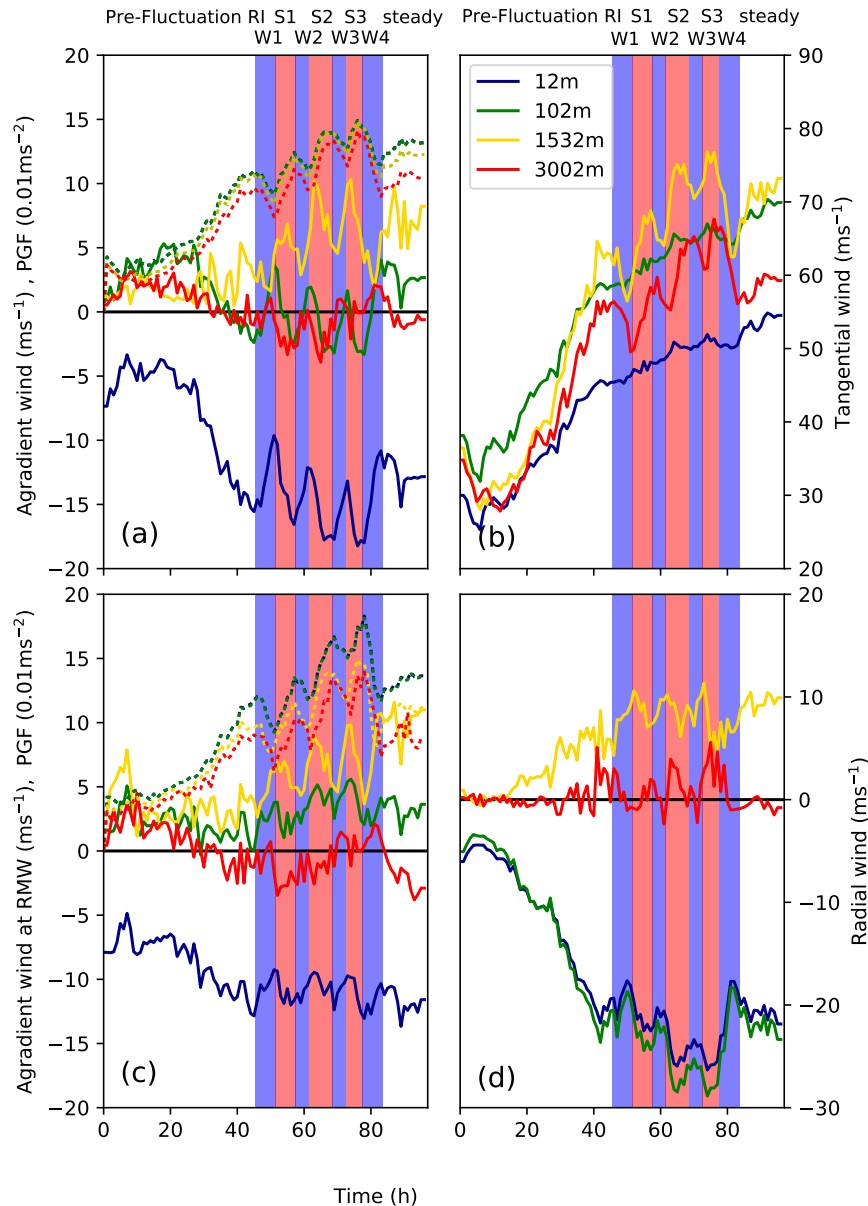

**Figure 14.** Left column shows azimuthally averaged agradient wind as a function of time (m s$^{-1}$) for (a) a radius of 35 km and (c) at the height dependent RMW. The right column shows, for the 35 km radius, the azimuthally averaged (b) tangential and (d) radial winds (m s$^{-1}$). The height of the lines are 12 m (blue), 102 m (green), 1902 m (orange) and 3002 m (red). Panels (a) and (c) also show the pressure gradient force (0.01 m s$^{-2}$, dashed lines) at selected levels. The RMW refers to the radius of maximum azimuthally averaged tangential wind at each specified height.

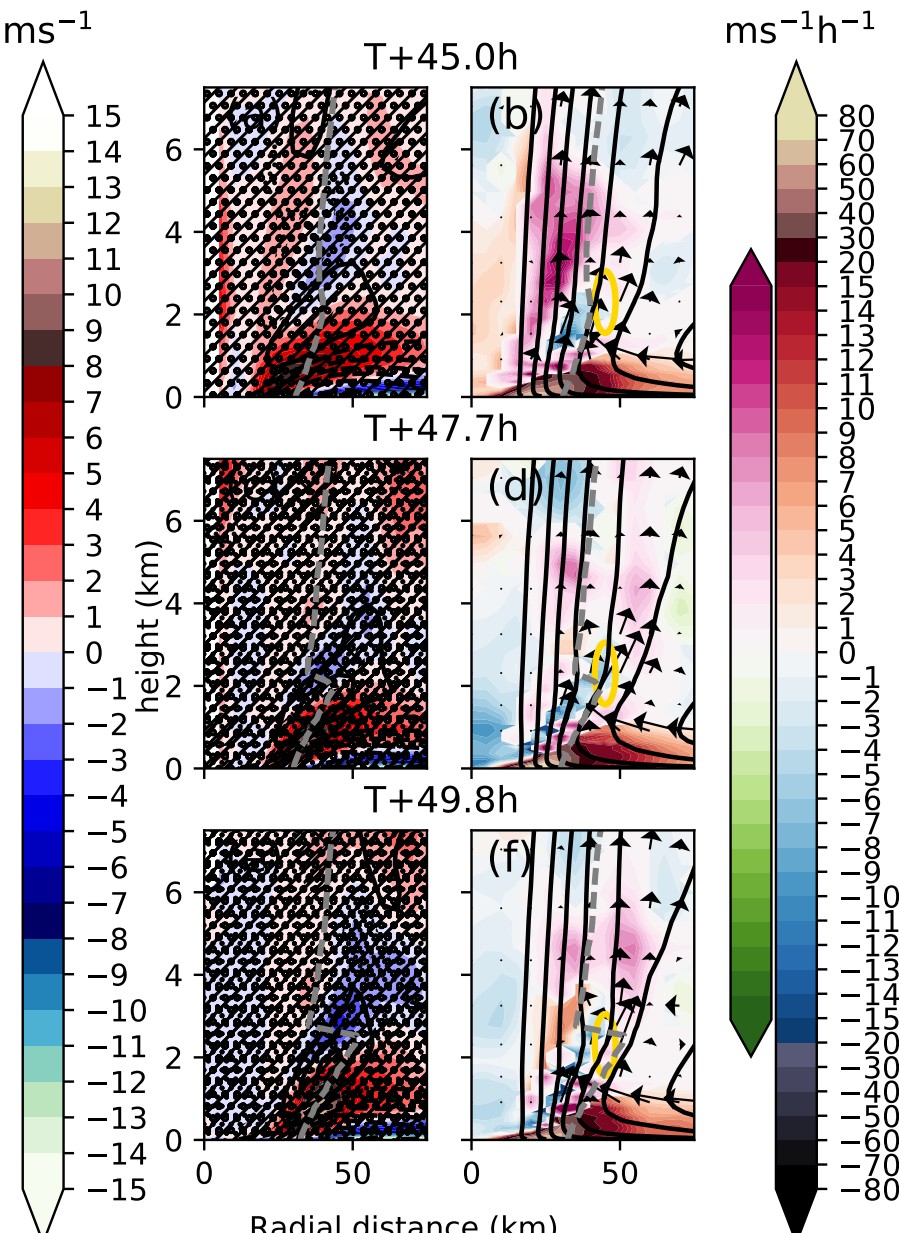

**Figure 15.** Left column shows, as a function of height and radius: the agradient wind (shading, ms$^{-1}$ left colourbar), the radial wind in intervals of $4\,\mathrm{m\,s^{-1}}$ with dashed lines indicating negative values, the tendency in tangential wind as small dots showing $+2\,\mathrm{m\,s^{-1}\,h^{-1}}$, large dots showing $+4\,\mathrm{m\,s^{-1}\,h^{-1}}$, line hatches showing $-2\,\mathrm{m\,s^{-1}\,h^{-1}}$ and cross hatches showing $-4\,\mathrm{m\,s^{-1}\,h^{-1}}$. Right column shows angular momentum (lines in units of $5\times10^{-5}\,\mathrm{m^2\,s^{-1}}$) and the secondary circulation as arrows in the plane of the cross section (with the boundary layer strong inflow omitted for clarity). The shading shows the contribution of the sum of the radial and vertical advection of angular momentum to the tangential wind budget. The colour scale used indicates which is the dominant term. If radial advection dominates over vertical advection then the blue/red shading is used and if vertical advection is dominant over radial advection then the green/purple scheme is used. For example green shading implies that the radial and vertical advection of angular momentum causes a negative tangential wind tendency and that the vertical term dominates. Also shown is RMW as the dashed grey line. The times shown in (a,b) are T+45 h, (c,d) T+47.4 h, and (e,f) T+49.8 h (the first three panels in Fig. 9). A region of interest is denoted by the yellow ellipse.

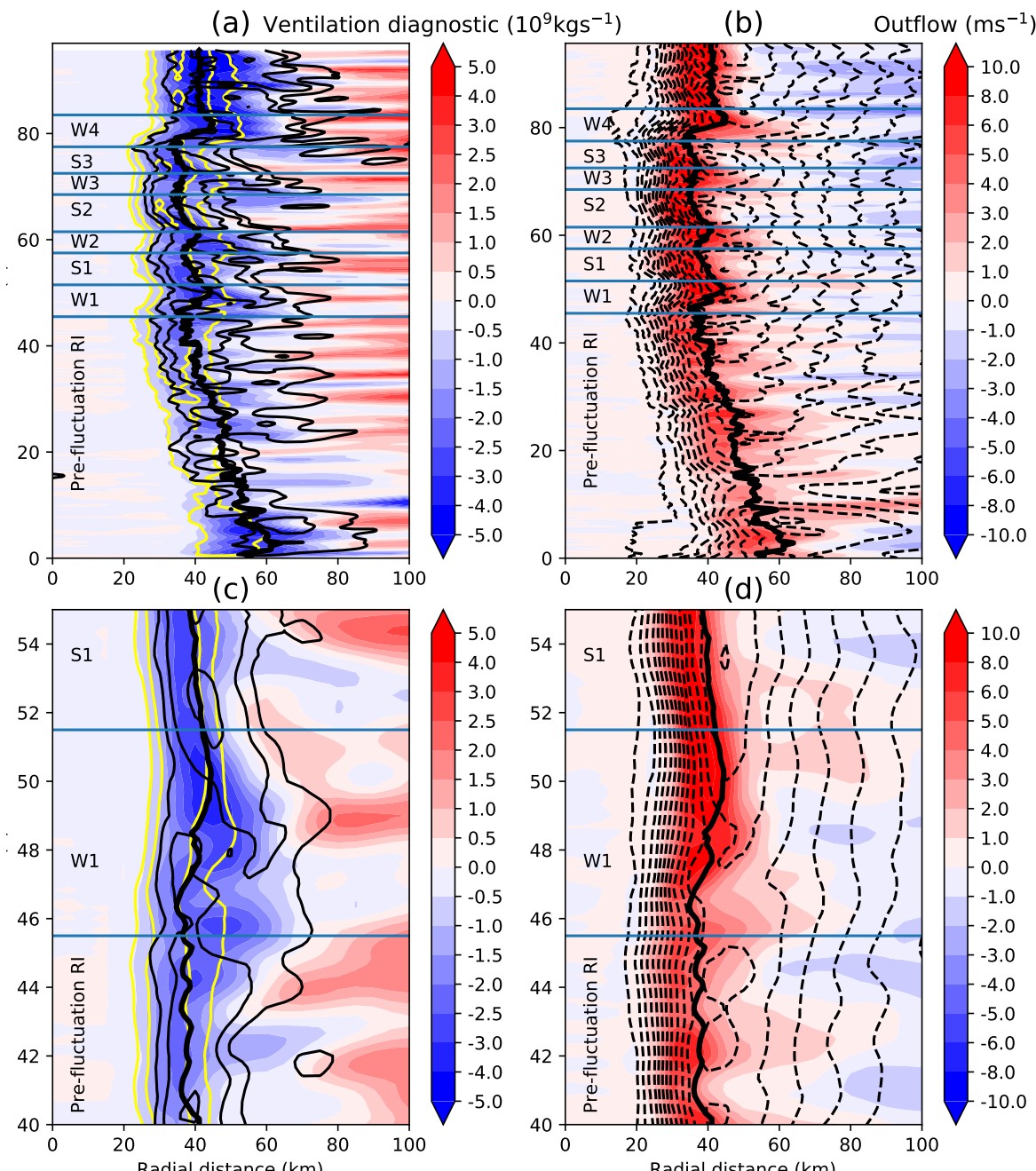

**Figure 16.** (a) Coloured contours show Ventilation diagnostic index ($10^9\mathrm{kg\,s^{-1}}$) with the azimuthally averaged radially integrated mass flux taken between a height of 6 and 1 km as a function of integration radius. Black contours show vertical velocity in $1\mathrm{m\,s^{-1}}$ intervals and the $0.5\mathrm{m\,s^{-1}}$ contour for 6 km height while the yellow contours show the same for 1 km height. (b) Coloured contours show the azimuthally averaged radial wind at 1532 m height (just above the boundary layer, $\mathrm{m\,s^{-1}}$) while black contours show the azimuthally averaged surface radial wind in $2\mathrm{m\,s^{-1}}$ intervals (dashed contours indicate negative radial wind or inflow). (c) and (d) show zoomed in versions of (a) and (b) respectively highlighting the times around W1. The RMW for the maximum azimuthally averaged tangential wind at 1532 m height is indicated by the thick black line in all subplots.

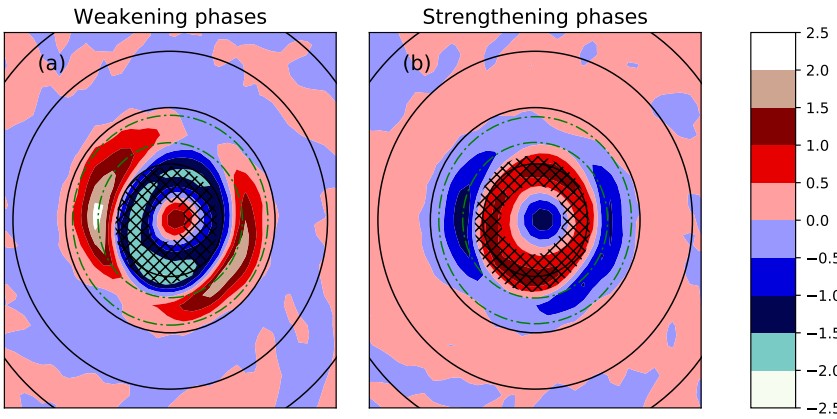

**Figure 17.** Composite PV tendencies ($\mathrm{PVU\,h^{-1}}$ shading) at $1532\,\mathrm{m}$ across all weakening and strengthening phases in the five ensembles with distinct intensity fluctuations. Green dashed lines show the full range of RMWs at the same level. Hatching indicates regions where the average PV exceeds 30 PVU. Black circles show $25\,\mathrm{km}$ radial intervals.

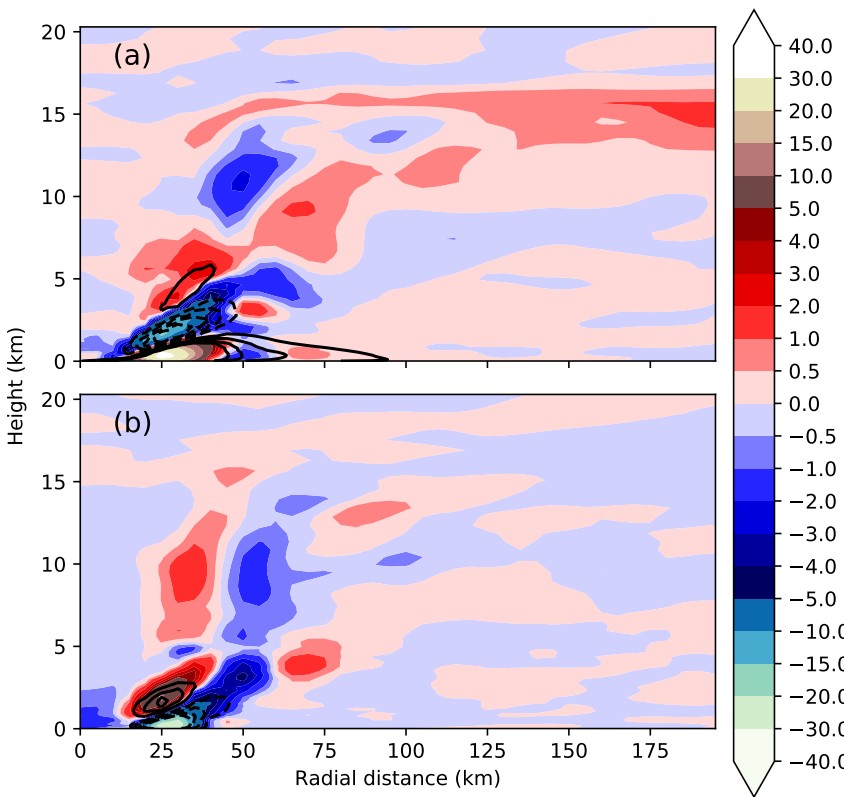

**Figure 18.** Absolute angular momentum budget composites showing: (a) mean advection of angular momentum, and (b) eddy advection of angular momentum . Colour shading shows the difference in tangential wind tendency between the strengthening phase composite and the weakening phase composite in $\mathrm{m\,s^{-1}\,h^{-1}}$. Line contours (5 $\mathrm{m\,s^{-1}\,h^{-1}}$ intervals, dashed lines imply negative values) show a composite of the contribution to tangential wind budget during all the strengthening phases (for example in subplot a at around 50 km there is a strongly positive tangential wind tendency from the mean term over all the strengthening phases).

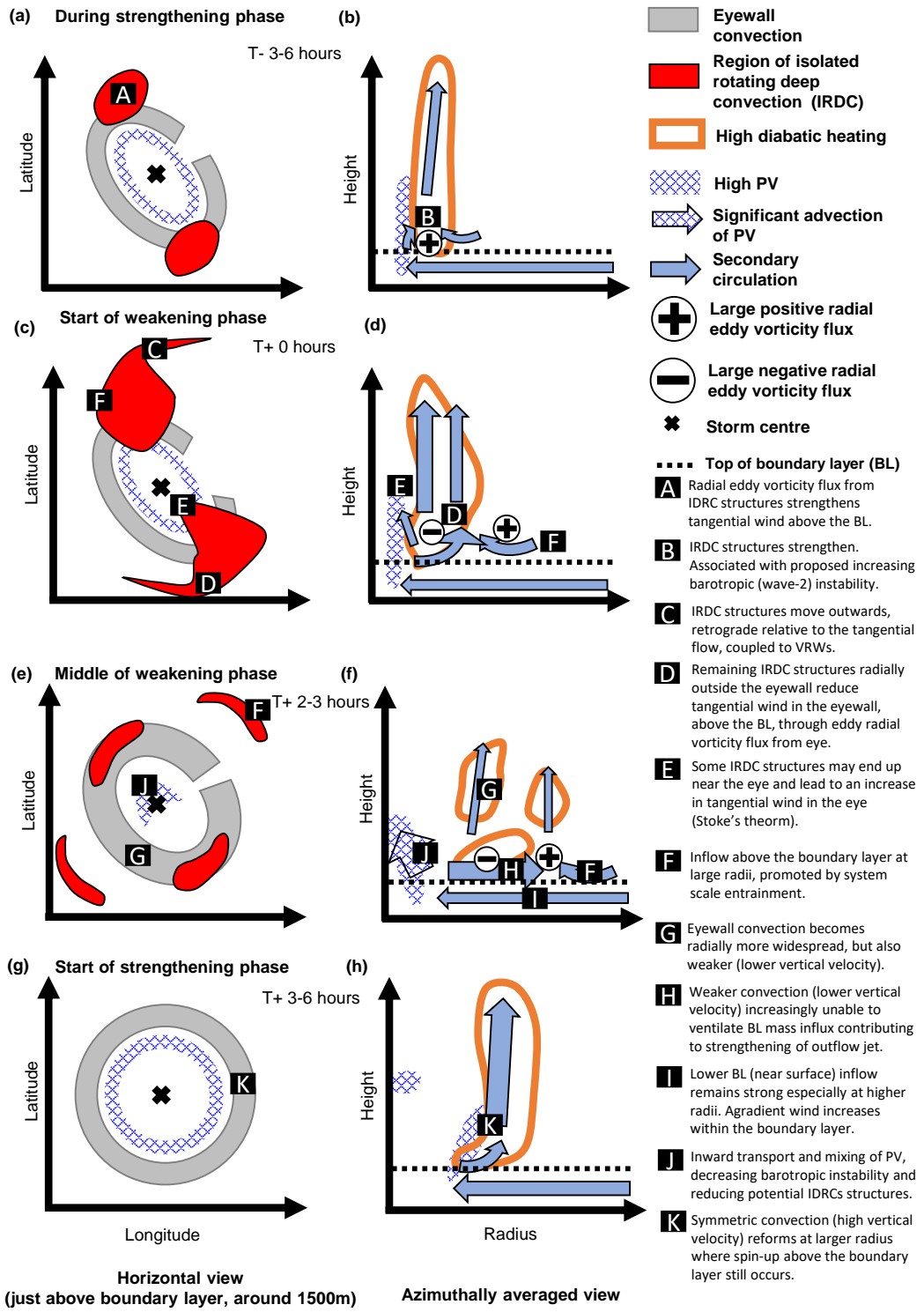

**Figure 19.** Schematic outlining the proposed mechanism for the fluctuations modelled during the rapid intensification of Hurricane Irma during: (a,b) the end of a strengthening phase, (c,d) the middle of a weakening phase and (e,f) the start of the strengthening phase. Left column shows the horizontal structure. Right column shows the azimuthally averaged structure of the storm at each stage with arrows indicating the direction of the secondary circulation (larger arrows imply stronger flow).