# Peer review of "Intensity fluctuations in Hurricane Irma (2017) during a period of rapid intensification"

_Weather and Climate Dynamics, 2021_

## Referee Comment (RC1)

Review of the manuscript entitled: "Intensity fluctuations in Hurricane Irma (2017) during a period of rapid intensification", by William Torgerson, Juliane Schwendike, Andrew Ross and Chris J. Short, submitted to *Weather and Climate Dynamics Discussions*.

**Recommendation**: Major revision

**Summary**

This paper presents an analysis of intensity fluctuations observed during a period of rapid intensification of Hurricane Irma (2017) using an ensemble of Met Office Unified Model convection-permitting forecasts. It is shown that intensity fluctuations consist of alternating weakening and strengthening phases and that during the weakening phases, the tropical cyclone temporarily paused its intensification. Reasons for the intensity fluctuations are explored.

While this study is a commendable attempt to provide dynamical interpretations of the storm behaviour, in our view it falls short of providing a clear understanding of the phenomena described. Moreover, the summary cartoon devised to underpin the explanations raises a number of questions as highlighted below.

The authors have tried to identify pieces of the "intensity-fluctuation puzzle", but have not yet provided a convincing link between the various pieces. We have some suggestions below for a way forward and would encourage the authors to consider these suggestions.

**Major comments**

1. The cartoon presented in Fig. 18 raises a number of questions:

   (a) The text in panel (b) reads: "Balanced effect of VHT in inner rainband creates convergence just above the boundary layer lowering the pressure gradient force and increasing the agradient wind." It is difficult to follow this reasoning. What is the "balanced effect of VHT"? And how does convergence lower the pressure gradient force? Are you talking about the radial pressure gradient?

   (b) In panel (d), how does the "lack of diabatic heating" *cause* "PV to be mixed inside the eye"?

   (c) In panel (e), why does the eyewall reform at a larger radius? This step would seem to be an important one in the whole process.

   The cartoon is rather suggestive of a phenomenon recently articulated by Smith et al. (2021) that the authors may be unaware of.

2. The key findings of the study are enumerated in the summary and conclusions section, but the take home message from these findings are somewhat thin. Let me discuss these in turn:

   (a) "In Hurricane Irma, during the second period of rapid intensification, intensity fluctuations occurred, which caused short term intensification and weakening periods, although overall the storm continued to intensify."

   This would seem to be a tautology: "intensity fluctuations" causing "short term intensification and weakening periods". This doesn't say very much of substance.

(b) "During strengthening phases the PV distribution was an elongated ring which became more azimuthally symmetric and monopole-like during weakening phases. Note that the azimuthal symmetry is independent of the radial PV distribution and the ring-like PV states (strengthening phases) were associated with less azimuthally symmetric distributions."

The first sentence is merely a description of the simulation. It is not clear to us why is this information is being provided and in what way it provides new understanding. The second sentence is simply mysterious.

(c) "During strengthening phases, the diabatic heating distribution had a smaller radial extent and a stronger heating maximum which is located within the RMW. During weakening phases the heating was outside the RMW and had a greater radial extent than the diabatic heating during the strengthening phases."

The same remark can be made as that in (b). Why is this finding thought to be worthy of mention? In particular, what is the explanation for these changes in diabatic heating rate. In fact, what is meant by "smaller radial extent". Do you mean radial thickness? And smaller than what?

(d) "VHT-like structures were stronger and more common during strengthening phases than weakening phases and contributed positively to intensification through eddy advection of angular momentum."

Are you talking about radial advection or vertical advection of angular momentum? Wouldn't one be surprised if this were not the case? See e.g. Nguyen et al. (2008). However, note that the localized VHT structures project also on to the mean fields as well as the eddy fields (see e.g. Persing et al., 2013).

(e) "Unbalanced dynamics were shown to play a role in the intensity fluctuations. During the weakening phases an unbalanced supergradient tangential flow produced an outflow jet which acted to spin-down the flow above the boundary layer by transferring low angular momentum from the eye outwards."

This may be the case, but how do the authors account for the unbalanced supergradient tangential flow above the boundary layer in the first place?

It would be interesting to know how the diabatic heating distribution relates to the ventilation diagnostic introduced in the paper by Smith et al. (2021), which is seen as a measure of the ability of deep convection to evacuate mass at the rate it is converging in the boundary layer. In fact, this ventilation diagnostic may provide a useful link to relate the various quantities investigated in this paper.

Specific comments

1. L4-7: It is unclear why potential vorticity and relative vorticity are invoked in the same sentence. Why not stick with one or the other? Is there any special reason to prefer potential vorticity over relative vorticity?

2. The acronym RMW is defined at line 25, but no height is specified. Is it the absolute maximum? It is relevant to know this height in Fig. 14, for example.

3. Figure 4: Must be max tangential wind speed. What is the RMW of vmax?

4. L257-326: Why are the PV fields being shown? Since PV is not conserved and you are not inverting it, its use needs to be explained. What is the significance of the structural changes of PV? In fact, how is it defined?

5. L327-335: The definition of the barotropic conversion rate should be spelled out in the text. Even so, the energy transfer is from the mean flow to the eddies, which would suggest that the maximum tangential wind speed is decreasing with time, which is apparently not the case. How important are the energy fluxes through the boundaries of the integration domain?

6. L343-369. The section on VHTs is purely descriptive and contains a lot of detail. The problem for us is, it is unclear where this section is leading. It would seem desirable to include a few sentences to guide the reader into where it is leading. For example, why are the VHTs important?

7. L405-407: This sentence seems to put the cart before the horse as it invokes the VHT structures to be the primary cause of the stirring in the eye. However, the stirring is presumably a result of barotropic instability, with the understanding that the convection is enhancing the growth rates of the unstable modes (see Nguyen et al. 2011).

8. L409-410: The authors should explain here why they are showing the distribution of diabatic heating rate. Are they invoking balance dynamics (following Shapiro and Willoughby, JAS, 1982) to infer changes in the mean secondary circulation that might ensue?

9. L412-415: What is the significance of the information provided here? Why is the lowest heating rate maximum at such a low level?

10. L416-417: Why is this information provided? Why is the greater radial extent of the heating rate worthy of note? What are the implications?

11. Section 4.3: the questions in (10) apply to much of this section. Shapiro and Willoughby op. cit. show that the balanced secondary circulation depends on the radial and vertical gradients of diabatic heating rate and not on the diabatic heating rate, itself. But what does the heating rate by itself tell us?

12. L457-459: This sentence is indigestible.

13. L463-464: This sentence is misleading since, from an axisymmetric perspective, converging air parcels in the vortex boundary layer are always losing AAM. The word "initially" is misleading as the arguments refer to radial displacements of air parcels.

14. L471ff: Figure 14 is intriguing and potentially important, but raises a number of questions. First, which RMW is being referred to here? And how does it compare with the radius of 35 km? Why would "a decrease in the agradient force per unit mass" *cause* "an increase in the agradient wind"? How does "the appearance of a convergence zone above the boundary layer" cause "a decrease in PGF"? How is the "balanced inflow" calculated, or is it speculation that the inflow is balanced? How does "rainband convection" enhance "the balanced inflow"?

15. L485ff: The arguments here seem to be pure speculation and I am totally confused.

16. First of all, why not explain the trends in this figure before trying to explain the wiggles? Second, in explaining both the trends and the wiggles, it would seem to be necessary to show the vertical advection of the supergradient winds from the boundary layer into the eyewall (Schmidt and Smith 2016, Montgomery and Smith 2017, see also Smith et al. 2020). It would appear that the results you show are affirmation of the eyewall spin up articulated in the latter papers. This would seem worthy of mentioning.

Section 4.4.2: To consider the tangential wind budget alone is not sufficient "to understand how the boundary layer and outflow jet change and lead to a spin–down above the boundary layer".

This budget cannot explain "changes in the secondary circulation and what drives these changes" as this requires consideration of the radial and vertical components of the momentum equation as well (Smith and Montgomery 2015). The primary reason that air flows outwards above the boundary layer is that inner-core deep convection is collectively too weak to ventilate mass at the rate that mass is being funnelled to the base of the eyewall by the boundary layer (Kilroy et al., 2016, Smith and Wang 2018, Montgomery et al., 2020, Smith et al., 2021).

L515: For reasons discussed above, "the intensity fluctuations in Hurricane Irma" cannot "be understood in terms of unbalanced boundary layer dynamics" without considering the changes in deep convective mass flux. One solution to this issue might be to investigate the ventilation diagnostic used by Smith et al. (2021).

L588: The inability of the boundary layer updraft to properly couple with a potential secondary updraft above is precisely what the last two points are trying to convey.

**Signed:** Roger Smith and Michael Montgomery

**References**

Kilroy G., R. K. Smith and M. T. Montgomery, 2016: Why do model tropical cyclones grow progressively in size and decay in intensity after reaching maturity? J. Atmos. Sci., 73, 487-503.

Montgomery M. T. and R. K. Smith, 2017: Recent developments in the fluid dynamics of tropical cyclones. Annu. Rev. Fluid Mech., 49, 541-574, doi: 10.1146/annurev-fluid-010816-060022.

Montgomery M. T., G. Kilroy, R. K. Smith, and N. Črnivec, 2020: Contribution of mean and eddy momentum processes to tropical cyclone intensification. Quart. J. Roy. Meteor. Soc., 146, 3101-3117.

Nguyen S. V., R. K. Smith and M. T. Montgomery, 2008: Tropical-cyclone intensification and predictability in three dimensions. Quart. J. Roy Met. Soc. 134, 563-582.

Nguyen C. M., M. J. Reeder, N. E. Davidson, R. K. SMITH, and M. T. Montgomery, 2011: Inner-core vacillation cycles during the rapid intensification of Hurricane Katrina. Quart. J. Roy Met. Soc., 137, 829-844.

Persing, J., M. T. Montgomery, J. McWilliams, and R. K. Smith, 2013: Asymmetric and axisymmetric dynamics of tropical cyclones . Atmos. Chem. Phys., 13, 12249-12341.

Schmidt C. W. and R. K. Smith, 2016: Tropical cyclone evolution in a minimal axisymmetric model revisited. Quart. J. Roy. Meteor. Soc., 142, 1505-1516.

Smith R. K. and M. T. Montgomery, 2015: Towards clarity on understanding tropical cyclone intensification. J. Atmos. Sci., 72, 3020-3031.

Smith R. K. and S. Wang 2018: Axisymmetric balance dynamics of tropical cyclone intensification: Diabatic heating versus surface friction. Quart. J. Roy. Meteor. Soc., 144, 2350-2357.

Smith R. K., G. Kilroy, and M. T. Montgomery, 2020: Comments on: How much does the upward advection of supergradient component of boundary-layer wind contribute to tropical cyclone intensification and maximum intensity? by Y Li, Y. Wang and Y. Lin. J. Atmos. Sci., 77, 4377-4378.

Smith R. K., G. Kilroy, and M. T. Montgomery, 2021: Tropical cyclone life cycle in a three-dimensional numerical simulation. Quart. J. Roy. Meteor. Soc., 147, 3373-3393.

---

## Author Response (AR1)

**Reply to the reviewers' comments**

We would like to thank Roger Smith, Mike Montgomery, and Reviewer 2 for carefully reading the manuscript and their suggestions to improve it. We have addressed all points below.

**Reviewer 1**

**Recommendation: Major revision**

**Summary**
**This paper presents an analysis of intensity fluctuations observed during a period of rapid intensification of Hurricane Irma (2017) using an ensemble of Met Office Unified Model convection-permitting forecasts. It is shown that intensity fluctuations consist of alternating weakening and strengthening phases and that during the weakening phases, the tropical cyclone temporarily paused its intensification. Reasons for the intensity fluctuations are explored.**

**While this study is a commendable attempt to provide dynamical interpretations of the storm behaviour, in our view it falls short of providing a clear understanding of the phenomena described. Moreover, the summary cartoon devised to underpin the explanations raises a number of questions as highlighted below.**

**The authors have tried to identify pieces of the "intensity-fluctuation puzzle", but have not yet provided a convincing link between the various pieces. We have some suggestions below for a way forward and would encourage the authors to consider these suggestions.**

We thank the reviewers for their interest in the research and for their helpful suggestions to improve the work. In revising the manuscript, we have attempted to address the issues raised and feel that as a result there is a much clearer message in the paper. We will respond to the individual comments below.

**Major comments**

**1. The cartoon presented in Fig. 18 raises a number of questions:**

**(a) The text in panel (b) reads: "Balanced effect of VHT in inner rainband creates convergence just above the boundary layer lowering the pressure gradient force and increasing the agradient wind." It is difficult to follow this reasoning. What is the "balanced effect of VHT"? And how does convergence lower the pressure gradient force? Are you talking about the radial pressure gradient?**

We think this may just require some clearer explanation. By 'balanced effect' of VHT we were referring to the induced secondary circulation caused by the azimuthally averaged heating and AAM of these structures.

We have been able to show that the weakening of the pressure gradient force within the eyewall is due to a rise in the pressure which lowers the radial pressure gradient (Fig. R1a,b). An increase in pressure may be due to mass convergence or a weakening in the ascent, both explanations are plausible, but we haven't been able to determine which is the main cause as the pressure budget does not balance well (Fig. R1c,d). The decrease in the pressure gradient

force is accompanied by a decrease in the tangential wind and therefore a decrease in the centrifugal and Coriolis force. As a result, approximate gradient wind balance, above the boundary layer, is maintained. This change in the pressure gradient force is instantaneously transferred into the boundary layer, however, the tangential winds do not decrease and so the centrifugal and Coriolis force remain high, hence the agradient wind within the boundary layer increases. We have added this explanation to the first section in the unbalanced dynamics part of the manuscript (Section 4.4).

**(b) In panel (d), how does the "lack of diabatic heating" *cause* "PV to be mixed inside the eye"?**

This expression tries to convey that diabatic heating is required to maintain the ring-like PV structure (since the ring is barotropically unstable), and that in the absence of diabatic heating, the most stable state is a PV monopole. So, a weakening of the diabatic heating seems to cause a tendency towards a more barotropically stable monopole-like PV state. We have revised the text in the discussion describing this process to make our meaning clear.

**(c) In panel (e), why does the eyewall reform at a larger radius? This step would seem to be an important one in the whole process.**

The BL outflow jet is causing weakening at the radius of the eyewall while outside the eyewall spin up is occurring due to the unbalanced spin up mechanism. Re-formation of the eyewall at a larger radius can be explained by spin-up occurring outside of the eyewall and weakening within the eyewall. The caption in the cartoon in Fig. 19 in the manuscript has been changed accordingly.

[Figure]

[Figure]

**Figure R1:** (a) Pressure gradient force tendency (m s$^{-2}$ h$^{-1}$, shading) and black contours show the radial wind in 1 m s$^{-1}$ intervals. (b) Coloured contours show pressure tendency (hPa h$^{-1}$) and black contours show radial wind in 1 m s$^{-1}$ intervals. (c) The right-hand side of the pressure 'budget' (integrated horizontal divergence + vertical velocity + integrated mass advection in units of hPa h$^{-1}$.)

**The cartoon is rather suggestive of a phenomenon recently articulated by Smith et al. (2021) that the authors may be unaware of.**

Thank you for pointing out the Smith et al. (2021) paper, which we had not seen before. There are indeed some striking similarities here. Our work may help tie together the boundary layer processes described in Smith et al. (2021) and previous work on vacillation cycles and 'fake SEF' in failed eyewall replacement cycles which also describes a situation where a rainband has a disruptive effect on the eyewall, but a secondary eyewall replacement does not occur. A future paper will look at some key differences between the proposed mechanism and eyewall replacement cycles. In the manuscript we added a reference to Smith et al. (2021) and included discussion of how their results fit with our findings.

**The key findings of the study are enumerated in the summary and conclusions section, but the take home message from these findings are somewhat thin. Let me discuss these in turn:**

**(a) "In Hurricane Irma, during the second period of rapid intensification, intensity fluctuations occurred, which caused short term intensification and weakening periods, although overall the storm continued to intensify."**
**This would seem to be a tautology: "intensity fluctuations" causing "short term intensification and weakening periods". This doesn't say very much of substance.**

We agree that the sentence could have been formulated better and have changed the wording. You are correct, intensity fluctuations don't cause weakening and intensification but are defined as periods of weakening. In other words, in intensity fluctuations occurring during RI, the "weakening periods" were characterised by an increase or the stagnation of the pressure and a decrease in the tangential wind. The point we want to make here is that the fluctuations do not stop the overall intensifying (RI) trajectory of the storm. In other words, the reduction of the tangential wind speed during a weakening period is always smaller than the increase in the tangential wind speed during the next strengthening period.

**(b) "During strengthening phases the PV distribution was an elongated ring which became more azimuthally symmetric and monopole-like during weakening phases. Note that the azimuthal symmetry is independent of the radial PV distribution and the ring-like PV states (strengthening phases) were associated with less azimuthally symmetric distributions."**
**The first sentence is merely a description of the simulation. It is not clear to us why is this information is being provided and in what way it provides new understanding. The second sentence is simply mysterious.**

The main point here is that azimuthal symmetry is anti-correlated with how ring-like the PV distribution is, which is the *opposite* of what previous papers found. We have made that clearer in the text.

**(c) "During strengthening phases, the diabatic heating distribution had a smaller radial extent and a stronger heating maximum which is located within the RMW. During weakening phases the heating was outside the RMW and had a greater radial extent than the diabatic heating during the strengthening phases."**
**The same remark can be made as that in (b). Why is this finding thought to be worthy of mention? In particular, what is the explanation for these changes in diabatic heating rate. In fact, what is meant by "smaller radial extent". Do you mean radial thickness? And smaller than what?**

When calculating the potential temperature budget, the term that contributed the most was the 'cloud rebalancing' parametrization (latent heat release from cloud formation), so the weakening of the eyewall, in terms of decrease in tangential wind, can be attributed to less latent heat release in the weakening phases since a decrease in the strength of the convection prevents the eyewall from being able to ventilate as much of the mass flux from the boundary layer. By smaller radial extent we do mean the thickness. If you take an arbitrary medium heating contour like 20 K h$^{-1}$, in the weakening phases it would have a greater radial extent, but a higher contour such as 50 K h$^{-1}$ might be completely absent unlike in the strengthening phases where the heating column is stronger and narrower. We have reworded this point to make it clearer what we are trying to say.

**(d) "VHT-like structures were stronger and more common during strengthening phases than weakening phases and contributed positively to intensification through eddy advection of angular momentum."**
**Are you talking about radial advection or vertical advection of angular momentum? Wouldn't one be surprised if this were not the case? See e.g. Nguyen et al. (2008). However, note that the localized VHT structures project also on to the mean fields as well as the eddy fields (see e.g. Persing et al., 2013).**

We grouped the radial advection and vertical advection terms together. However, above the boundary layer radial advection of eddy angular momentum is largely responsible for the spin up. We agree that the VHT-like structures do project onto the main field, but we viewed the effect of the VHT-like structures on the eddy fields to be the most relevant to the mechanism of the intensity fluctuations. We did also plot the mean fields which show that mean advection of absolute angular momentum is responsible for spin up within the boundary layer. We have reworded this point in the paper.

**(e) "Unbalanced dynamics were shown to play a role in the intensity fluctuations. During the weakening phases an unbalanced supergradient tangential flow produced an outflow jet which acted to spin-down the flow above the boundary layer by transferring low angular momentum from the eye outwards."**
**This may be the case, but how do the authors account for the unbalanced supergradient tangential flow above the boundary layer in the first place?**

We did not mention a supergradient wind above the boundary layer, instead we refer to a supergradient wind *within* the boundary layer that provokes an outflow response at the top of the boundary layer in order to return to gradient balance in the free vortex above. We have reworded this point in the paper.

**It would be interesting to know how the diabatic heating distribution relates to the ventilation diagnostic introduced in the paper by Smith et al. (2021), which is seen as a measure of the ability of deep convection to evacuate mass at the rate it is converging in the boundary layer. In fact, this ventilation diagnostic may provide a useful link to relate the various quantities investigated in this paper.**

Adding the ventilation diagnostic from Smith et al. (2021) is a very good suggestion as it provides a useful means of understanding the dynamics at a deeper level. We have added a figure for ventilation diagnostic to the manuscript (Fig. 16) along with a relevant subsubsection explaining its significance and compared the results to Smith et al. (2021). The figure is also included in this reply as Fig. R2 and shows how the ventilation index and the related outflow jet vary throughout the simulation. We see that during the weakening phases the ventilation index becomes more negative which is associated with the reduced convection in the eyewall region increasingly more unable to ventilate the boundary layer inflow. The decreasing ventilation index is also associated with a concomitant enhancement of the boundary layer

outflow jet which leads to further weakening of the eyewall convection, and in turn a further decrease in the ventilation diagnostic.

[Figure]

**Figure R2:** (a) Coloured contours show ventilation diagnostic index $10^9$ kg $s^{-1}$ with the azimuthally averaged radially integrated mass flux taken between a height of 6 and 1 km as a function of integration radius. Black contours show vertical velocity in 1 m $s^{-1}$ intervals and the 0.5 m $s^{-1}$ contour for 6 km height while the yellow contours show the same for 1 km height. (b) Coloured contours show the azimuthally averaged radial wind at 1532 m height (just above the boundary layer, m $s^{-1}$) while black contours show the azimuthally averaged surface radial wind in 2 m $s^{-1}$ intervals (dashed contours indicate negative radial wind or inflow). (c) and (d) show zoomed in versions of (a) and (b) respectively highlighting the times around W1. The RMW for the maximum azimuthally averaged tangential wind at 1532 m height is indicated by the thick black line in all subplots.

**Specific comments.**

**1. L4-7: It is unclear why potential vorticity and relative vorticity are invoked in the same sentence. Why not stick with one or the other? Is there any special reason to prefer potential vorticity over relative vorticity?**

We prefer PV as a diagnostic tool due to its link to barotropic stability, but relative vorticity seems to be most commonly used in literature on VHTs. That is why we use relative vorticity when talking about VHT-like structures.

**2. The acronym RMW is defined at line 25, but no height is specified. Is it the absolute maximum? It is relevant to know this height in Fig. 14, for example.**

In line 25 in the literature review we are referring to a very simplistic hypothetical case. We made this clearer in the revised version of the manuscript. Where we plot height dependent RMW in the results section we have made clear at which height this is calculated.

**3. Figure 4: Must be max tangential wind speed. What is the RMW of vmax?**

The RMW is the radius at which the azimuthally-averaged tangential wind at the specified height is maximum. Where 'surface (10m) maximum total wind speed' is used, it refers to the maximum total wind speed with no azimuthal averaging. We have updated the text and figure caption accordingly.

**4. L257-326: Why are the PV fields being shown? Since PV is not conserved and you are not inverting it, its use needs to be explained. What is the significance of the structural changes of PV? In fact, how is it defined?**

PV gives insight into how the barotropic structure and instability changes between the weakening and strengthening phases. In addition, in previous research on vacillation cycles PV was used as a key metric to distinguish between 'asymmetric' and 'symmetric' states which have some similarities to our weakening and strengthening phases. Looking at how the PV distribution changes will allow for a better comparison between our fluctuations and the described vacillation cycles and help determine whether they are equivalent phenomena.

We added a sentence at the start of the PV paragraph which reads as: "Previous studies on vacillation cycles have used PV as a metric to show the structural changes of the vortex during the weakening and strengthening phases. Van Sang et al. (2008) described how, a barotropically unstable ring–like PV state would break down into isolated inward moving PV anomalies. To determine whether the intensity fluctuations are similar to these vacillation cycles it is helpful to examine this PV structure."

**5. L327-335: The definition of the barotropic conversion rate should be spelled out in the text. Even so, the energy transfer is from the mean flow to the eddies, which would suggest that the maximum tangential wind speed is decreasing with time, which is apparently not the case. How important are the energy fluxes through the boundaries of the integration domain?**

We have added the definition of the barotropic conversion rate along with some references to its use in vacillation cycle papers. We agree it does seem counter intuitive that tangential windspeed decreases, although this is only one term of the eddy kinetic energy budget (Hankinson et al., 2014):

$$\frac{\partial K'}{\partial t} = ERF + EVF + CONV - BARO + Fric_{K'}$$

We have explicitly added the equation into the paper and explained how the barotropic conversion index can be negative throughout the simulation while the TC is still able to intensify. The variable functions as a useful proxy for the barotropic instability which is important in determining the structure of the tropical cyclone.

**6. L343-369. The section on VHTs is purely descriptive and contains a lot of detail. The problem for us is, it is unclear where this section is leading. It would seem desirable to include a few sentences to guide the reader into where it is leading. For example, why are the VHTs important?**

Adding a motivation for the VHT analysis is a good suggestion. VHTs are important because they are likely to play an important role in the transition from a strengthening phase to a weakening phase. In addition to the fact that VHT-like structures appear more often and are stronger during the strengthening phase, they are also hypothesised to be an artefact of high convective and barotropic instability and the balanced response of these structures in terms of the inflow they induce at higher radii leads to convergence which we think plays a role in the transition to the weakening phase.

**7. L405-407: This sentence seems to put the cart before the horse as it invokes the VHT structures to be the primary cause of the stirring in the eye. However, the stirring is presumably a result of barotropic instability, with the understanding that the convection is enhancing the growth rates of the unstable modes (see Nguyen et al. 2011).**

The PV-ring state is always barotropically unstable due to the presence of a sign change in the PV gradient. The key processes are that diabatic heating weakens and therefore the TC cannot remain in the barotropically unstable state. Consequently, PV is transported inwards towards the eye at the start of the weakening phase and the PV structure becomes more monopole-like. In addition, the VHTs are at their strongest just prior to the start of the weakening phase. We have added more details at the end of the tangential wind budget discussion (section 4.2.3 of the revised manuscript) with reference to the VHT-like structures emerging as a result of the barotropic instability.

**8. L409-410: The authors should explain here why they are showing the distribution of diabatic heating rate. Are they invoking balance dynamics (following Shapiro and Willoughby, JAS, 1982) to infer changes in the mean secondary circulation that might ensue?**

You are right, balanced dynamics are indeed being invoked. Additional work (not shown in the paper) was undertaken to provide robust evidence for this balanced induced circulation above the boundary layer including the running of the idealized balanced model as used in Smith et al. (2015) to replicate some of the features in the secondary circulation. We have added more explanations in lines 434-436 of the revised manuscript.

**9. L412-415: What is the significance of the information provided here? Why is the lowest heating rate maximum at such a low level?**

The diabatic heating distributions (and their radial and vertical gradients) are important to understand the balanced circulations above the boundary layer. In addition, weakening diabatic heating in the eyewall is relevant in terms of the TC's inability to maintain the unstable

barotropic structure and inability to ventilate mass from the boundary layer. The important information is the differences in the heating distributions between the weakening and strengthening phases. We have added information about potential temperature budgets in line 437 - 439

**10. L416-417: Why is this information provided? Why is the greater radial extent of the heating rate worthy of note? What are the implications?**

The larger radial extent of the diabatic heating distribution is the most substantive difference between weakening and strengthening phases in terms of diabatic heating distributions. The larger radial extent in the weakening phase is important at developing a secondary circulation at the higher RMW, sowing the seeds for the RMW increase. We have added some points in the manuscript in lines 444-445 to make this clear.

**11. Section 4.3: The questions in (10) apply to much of this section. Shapiro and Willoughby op. cit. show that the balanced secondary circulation depends on the radial and vertical gradients of diabatic heating rate and not on the diabatic heating rate, itself. But what does the heating rate by itself tell us?**

This is a good point. Instances of 'diabatic heating' should be replaced with $r$ or $z$ gradients of diabatic heating. Fortunately, the argument and conclusion would remain unchanged. We have added some sentences to emphasise that it is the radial and vertical gradients of the diabatic heating that are important.

**12. L457-459: This sentence is indigestible.**

You are right and we have reworded the sentence. We were trying to say that: (i) VHT structures outside the original eyewall spread out azimuthally to form more azimuthally symmetric convection and (ii) this convection becomes dominant at the slightly larger RMW.

**13. L463-464: This sentence is misleading since, from an axisymmetric perspective, converging air parcels in the vortex boundary layer are always losing AAM. The word "initially" is mis-leading as the arguments refer to radial displacements of air parcels.**

Yes, this is true because the AAM is lower the closer to the centre you get. It isn't true that the gain through rapid reduction in $r$ needs to completely offset the AAM loss, it just needs to offset it enough such that the AAM decreases less rapidly than the AAM contours. This inaccuracy has been corrected.

**14. L471ff: Figure 14 is intriguing and potentially important, but raises a number of questions. First, which RMW is being referred to here? And how does it compare with the radius of 35 km? Why would "a decrease in the agradient force per unit mass" *cause* "an increase in the agradient wind"? How does "the appearance of a convergence zone above the boundary layer" cause "a decrease in PGF"? How is the "balanced inflow" calculated, or is it speculation that the inflow is balanced? How does "rainband convection" enhance "the balanced inflow"?**

The RMW defined as the radial location of the maximum in the azimuthally-averaged tangential wind varies with height and time unlike the fixed radius at 35 km also shown in panel a of Figure 14. This fixed radius was chosen as it approximately corresponds to where the eyewall is, although the eyewall's radius is not constant (and nor is the RMW) hence why the panels b-d of the same figure show the values at the RMW. We chose to show plots of both a fixed radius and an RMW dependent radius to separate out effects in the supergradient wind that can be explained solely by the expansion of the RMW. We have changed our argument

in the manuscript and tried to make our points clearer. The argument goes as follows: (1) The tangential wind weakens above the boundary layer in the eyewall region due to an increase in inner rainband convection enhancing the outflow jet and advecting low absolute angular momentum air outwards. (2) The reduction in the tangential wind above the boundary layer is accompanied by a reduction in the pressure gradient force such that gradient wind balance is maintained. (3) The reduction in PGF above the boundary layer is instantaneously transferred into the boundary layer leading to an increase in the agradient wind, i.e the wind becomes more supergradient within the boundary layer and ascent is therefore encouraged at a higher radius.

**15. L485ff: The arguments here seem to be pure speculation and I am totally confused.**

We have added some clarification to the paper. The positive radial gradient of the diabatic heating associated with rainband convection is thought to accelerate the outflow, above the boundary layer and radially inside the rainband. That is to say, the rainband induces a balanced outward secondary circulation in the same region as the outflow jet above the boundary layer, and therefore enhances this outflow jet.

**16. First of all, why not explain the trends in this figure before trying to explain the wiggles? Second, in explaining both the trends and the wiggles, it would seem to be necessary to show the vertical advection of the supergradient winds from the boundary layer into the eyewall (Schmidt and Smith 2016, Montgomery and Smith 2017, see also Smith et al. 2020). It would appear that the results you show are affirmation of the eyewall spin up articulated in the latter papers. This would seem worthy of mentioning.**

This is a good suggestion. We have added a figure below (Fig R4b) showing the vertical advection of the tangential wind in the supergradient boundary layer and have discussed the results in relation to Schmidt and Smith (2016) in the 3rd paragraph of the Unbalanced dynamics section of the revised manuscript. In the eyewall region the supergradient winds which are strongest in the middle of the boundary layer (at around 700m) are advected upwards. Above the boundary layer the radial advection of lower absolute angular momentum air approximately cancels out the vertical advection from the boundary layer and is close to gradient wind balance. During the weakening phases the wind becomes more supergradient in the boundary layer (see also Figure 14 in the revised manuscript). However, the vertical transfer of this supergradient wind only happens effectively outside the RMW at the top of the boundary layer. Hence, during the weakening phase the tangential wind is able to spin up outside the RMW but weakening still occurs inside.

[Figure]

**Figure R3:** Coloured contours show the strength of the agradient wind (m s⁻¹) at various levels throughout the boundary layer. Line contours show the vertical velocity. The RMW is indicated by the thick black line.

[Figure]

Figure R4: (a) Coloured contours show the difference in the tangential wind (m s⁻¹) between the top of the boundary layer (1532 m) and the middle of the boundary layer (667 m). Line contours show the vertical velocity (m s⁻¹). (b) Coloured contours show the magnitude of the contribution to the tangential wind of the vertical advection of absolute angular momentum (m s⁻¹ h⁻¹) at the 1532 m level. The RMW is indicated by the thick black line.

In addition, to the overall trend of the agradient wind throughout the simulation we can also see that the tangential wind at the top of the boundary layer is typically at least 5 m s⁻¹ weaker than at the middle of the boundary layer in the eyewall. At the start of, and a couple of hours prior to, the weakening phase outside the RMW the tangential wind at the top of the boundary layer becomes even weaker while inside the RMW the tangential wind may become stronger.

**Section 4.4.2: To consider the tangential wind budget alone is not sufficient "to understand how the boundary layer and outflow jet change and lead to a spin–down above the boundary layer".**

**This budget cannot explain "changes in the secondary circulation and what drives these changes" as this requires consideration of the radial and vertical components of the momentum equation as well (Smith and Montgomery 2015). The primary reason that air flows outwards above the boundary layer is that inner-core deep convection is collectively too weak to ventilate mass at the rate that mass is being funnelled to the base of the eyewall by the boundary layer (Kilroy et al., 2016, Smith and Wang 2018, Montgomery et al., 2020, Smith et al., 2021).**

We have included a figure in the redraft showing the ventilation diagnostic to provide a more complete picture as in Smith et al. 2021. We do believe the findings in our paper have some similarities to that of Smith et al., 2021 but with some key differences. Unlike Smith et al., 2021 the eyewall is always unable to completely ventilate the mass inflowing from the boundary layer and, as such, always has an outflow jet above the boundary layer. We do find that at the radius where the strongest inner rainband convection (and VHT-like structures) occur the convection is still insufficient to ventilate the incoming mass just prior to the weakening phases which leads to an outflow occurring above the boundary layer at radii greater than this radius of strongest convection, outside the eyewall. Also, we find that during the course of the weakening phase the ventilation diagnostic index, in the eyewall, becomes more negative as the convection within the eyewall weakens, which further increases the boundary layer outflow and provides an increasingly favourable environment for convection at a greater radius.

**L515: For reasons discussed above, "the intensity fluctuations in Hurricane Irma" cannot "be understood in terms of unbalanced boundary layer dynamics" without considering the changes in deep convective mass flux. One solution to this issue might be to investigate the ventilation diagnostic used by Smith et al. (2021).**

We have calculated the ventilation index as suggested and included a discussion of this index in ll. 542-566 of the revised manuscript. Please also see the reply to the previous point.

**L588: The inability of the boundary layer updraft to properly couple with a potential secondary updraft above is precisely what the last two points are trying to convey.**

We addressed this issue when editing the manuscript in regard to the previous two points.

**Reviewer 2**

This paper examines the relationship between changes in the intensity of Hurricane Irma, particularly during a 2-day period of rapid intensification, and a range of inner-core processes. The calculations and deductions are based on an ensemble of runs with the UK Met Office Unified Model, although most of the analysis is focused on the most realistic member of the ensemble.

**General Comments:**
My main criticism is that there is a great deal of detail presented, and that this level of detail is difficult at times to follow, making the paper hard work. It's not always easy to see the point of some of the details and the relevance to the narrative. Of course, the scientific story is complicated and, in my view, no paper on the topic has really nailed it yet. Nonetheless, the paper makes a strong contribution to the general topic of rapid intensification and documents some of the inner-core processes that lead to fluctuations in the rate of intensification. I recommend that it be published after major revision.

Thank you for your overall positive view on the paper. We have tried to "declutter' the paper to make it easier to read and not to distract the reader from the main message.

**Specific Comments:**

**L 53 - 56. Some of the references are a bit misleading. For example, the results attributed to Hankinson et al. (2014) and Reif et al. (2014) should be attributed to Nguyen et al. (2011) as the results appeared first in the original paper. The main contribution from Hankinson et al. was to extend the results of Nguyen et al. to an ensemble, and the main contribution from Reif et al. was to examine the robustness of the results to using a different non hydrostatic model (WRF).**

That is a good point. We have fixed that.

**L 95. "… and did not intensify due to less favourable environmental conditions." Be more explicit. What was it about the environment that prevented intensification?**

During this period the SSTs were marginal (around 26-27°C) and dry air was present to the northwest of the storm centre (Saharan air mass). We have added that information to the manuscript.

**L 98. " … with sufficient mid-level tropospheric moisture for intensification …". How much is sufficient? "… high sea surface temperatures …". Be explicit: what was the SST?**

During this period 500-700 hPa layer relative humidity has increased to around 55% around the TC as the TC had moved to the south of the dry airmass. SSTs had increased more markedly to 28-28.5°C and are at this point continuing to increase. We have updated the manuscript to include these details.

**L 104. "Despite favourable conditions …". What exactly was it about the environment that made the conditions favourable?**

Low vertical wind shear, high SSTs and adequate mid-level moisture (>50% 500-700 hPa relative humidity). We have updated the manuscript to make this clear.

**L 185. How well does the PV budget close?**

The Lagrangian form of the PV budget closes almost perfectly. The initial advected PV tracer plus the total PV tracer plus the dynamics tracers (first two terms in equation 1) almost exactly equal the LHS with the value of epsilon being extremely small. This budget is shown in Fig. R5 with the residual term being an order of magnitude smaller than the physics term. We have not included the figure in the manuscript but do now mention the small size of the residual term.

[Figure]

**Figure R5:** Lagrangian PV budget (PVU units) showing (a) the initial advection tracer subtracted from the diagnosed PV (Left hand side of Equation (1) in the revised manuscript), (b) the total physics tracer (first term in the right hand side of equation (1) in the revised manuscript), (c) the dynamic correction term (second term in the right hand side of equation (1) in the revised manuscript) and (d) the residual which is equivalent to $\varepsilon$ in equation (1) in the revised paper.

**L 291-307. The advective part of the PV change is discussed and plotted in Fig. 8. What about the physics part? How large is it? What's its structure and evolution? What part does it play in the story?**

The physics part is large (of comparable magnitude and opposite sign within the eyewall) and is dominated by the cloud rebalancing component (latent heat from cloud formation) which is shown in Figure R6 with the cloud rebalancing tracer and microphysics tracer responsible for most of the PV increases in the eyewall region. The main role of the physics part is maintenance of the PV ring within the eyewall to counterbalance loss through vertical advection. We have added a footnote to explain this in the paper.

[Figure]

**Figure R6:** Lagrangian PV budget (PVU units) showing (a) the advection calculated by taking the initial advection tracer subtracted from the diagnosed PV from the previous hour, (b) the cloud rebalancing tracer, (c) boundary layer tracer, (d) the microphysics tracer, and (e) the gravity wave tracer.

**L 331. I don't really follow this argument. The barotropic conversion rate becomes more positive (less negative) at the onset of the weakening phase, which means that barotropic processes are increasing the mean state. Wouldn't we expect that to correspond with an intensification of the vortex?**

The total kinetic energy in the mean state is also determined by other terms such as the mean and radial advection which contribute more to the total kinetic energy budget as in Hankinson et al. (2014). The main point that should be taken away from this discussion is that when the weakening phase starts, the barotropic instability has started to decline (become less negative). The emphasis should be on the increase in the magnitude of the barotropic instability towards the end of the strengthening phases, i.e. during the course of a strengthening phase the barotropic instability increases (barotropic conversion rate becomes more negative) until reaching a maximum prior to the weakening phase where it suddenly decreases and (barotropic conversion rate becomes more positive) less energy is now transferred from the mean to eddy state. An explanation has been added to the previous paragraph of the revised paper.

**L 339 - 340. ".. . with significant vertical depth albeit with lower values in these quantities." Be more explicit. How deep are the clouds and how strong are the updrafts compared to Smith and Eastin's definition?**

In Smith and Eastin the definition of a vortical hot tower requires perturbation vertical velocities greater than 5 m s$^{-1}$ and cloud depths greater than 6 km along with perturbation relative vorticity above 10$^{-3}$ s$^{-1}$. It is common to see perturbation vertical velocities in the 3-5 m s$^{-1}$ range and sometimes exceeding the 5 m s$^{-1}$ requirement and perturbation relative vorticities above 10$^{-3}$ s$^{-1}$. However, strict implementation of the algorithm requires that this be maintained over a depth of at least 6 km for the perturbation vertical velocity requirement and 3 km for the relative vorticity requirement, which never occurs in our simulations. In addition, these requirements must be met for the same latitude and longitude point, so a VHT-like structure that is sheared will also not meet the requirement. The algorithm was likely designed for a weaker storm (less than hurricane strength) prior to rapid intensification where VHTs are much stronger and deeper relative to the storm scale circulation. The strict definition has been added to the manuscript and compared to the less strict usage adopted here.

**L 394. How is weak and strong VHT activity defined?**

Where strong VHT exists, there will be perturbation relative vorticities above 10$^{-3}$ s$^{-1}$ and perturbation vertical velocities above 2 m s$^{-1}$ and either the perturbation relative vorticity or vertical velocity is maintained over a large depth (visible at 2532 m, 4963 m and 9934 m) for the same VHT. We have added this information to the manuscript.

**Figure 10. Consider W1 (row 1). The tangential wind in the eye wall is decreasing everywhere. In the boundary layer, the mean contribution is strongly positive and the contribution from friction is large and positive (from the figure caption). The contribution from the eddy terms is negative, but smaller than the mean term, and even smaller than the sum of the mean and friction terms. How then is the tangential wind tendency negative in the boundary layer? The same goes for row 1 of Fig. 11.**

This is a typo in the caption. The friction term is always large and negative, not positive. We have corrected that in the manuscript.

**L 402-407. As noted, the position of the VHT relative to the position of the eye wall is important as it effects whether the VHT spins up or spins down the vortex. The authors have done some work on this but not shown it in the paper. The relationship between the positions of the VHT and eye wall seems to me to be important and I think that the authors should expand on this point. In fact, it's central to the schematic later in the paper.**

Thank you for the suggestion. We have looked into it, and a key point we wish to emphasise which can be seen in Figures 10, 11 and 18 of the revised manuscript is that the VHT-like structures contribute to the increase in the strength of the tangential wind, in the eyewall above the boundary layer. The contribution to the strength of the tangential wind, above the boundary layer, is especially strong when the radial location of the VHT-like structures are near the RMW (see for example Fig. R8c,h where there is significant eddy contribution to the tangential wind just inside the RMW when the VHT-like structures are near the RMW). Prior to the weakening phases some differences are apparent, firstly the lifespan of the VHT-like structures are longer (compare rows 3, 4 with rows 1, 2 in Fig. XX) and the radial location of the structures are not limited to near the RMW. In the case of Fig. R8m,r there is distinct ascent (associated with these structures) both outside and inside the RMW. It is apparent that strong VHT-like activity, especially where these structures are present away from the RMW, are not beneficial to the TC in spinning up the tangential wind above the boundary layer just inside the RMW.

[Figure]

**Figure. R7:** Coloured contours show the eddy contribution to the tangential wind in ms$^{-1}$. Line contours show the azimuthally averaged vertical velocity in 0.2 ms$^{-1}$ intervals. The tendency in tangential wind is also shown as small dots showing +2 ms$^{-1}$ h$^{-1}$ large dots showing +4 ms$^{-1}$h$^{-1}$ line hatches showing -2 ms$^{-1}$ h$^{-1}$ and cross hatches showing -4 ms$^{-1}$h$^{-1}$ Each row represents a period where strong VHTs occurred with row a-j showing examples of where strong VHT activity did not subsequently lead to a weakening phase. Panels k-t show VHT activity prior to and during the start of W1 and W4 respectively. Columns show selected times with (a,f,k,p) representing the start of the period with strong VHT activity, (c,h,m,r) when the strongest activity occurred and (e,j,o,t) after the VHT activity. Other panels represent intermediate periods.

**L 448. "… inner rainbands which de-localized …". I don't know what this means.**

Initially the strongest convection associated with the inner rainbands is confined, largely, to the intersection between the rainbands and the eyewall. During the weakening phase this convection spreads out azimuthally covering an increasingly greater proportion of the eyewall until the localized structures have completely lost their individuality and the eyewall is essentially homogenous azimuthally. We have rephrased and clarified this sentence.

**L 476. ")". There's only a closing bracket.**

Changed.

**L 536. The introduction of the schematic is a a bit abrupt. There's no statement telling the reader that you're synthesising the results in a schematic. What's more Fig. 18 is reference before Figs. 16 and 17.**

This is a good point. We reference the schematic later in the revised manuscript and introduce it more to make it easier for the reader to follow.

**L 536. Should be "(Fig. 18 a, \*\*b\*\*)"?**

Changed.

**L 538. "... rainbands are not associated with the convective generation of PV outside the eyewall ...". Do you really know this? You've only told us about the advective changes. Have you calculated the diabatic change in the PV from Eq. 1? What does it look like?**

The other terms in the PV budget were calculated, and there was no change in the dominance of any particular budget term (cloud rebalancing was always the dominant physics term). During the weakening phase there was less PV generation within the eyewall but there was no indication of any increase in PV generation outside of the eyewall associated with convection and no secondary PV rings as in the case of an eyewall replacement cycle. We have added an explanation.

**L 544 and Fig. 18b. You can't really say that the convergence lowers the pressure gradient force, can you? Isn't the wind field responding to changes in the gradient of the pressure field?**

This sentence was unclear and we have reworded it to avoid confusion. The reduction in PGF is accompanied by a weakening of the tangential wind above the boundary layer such that gradient wind balance is maintained. In terms of the initial cause of the reduction in the tangential wind above the boundary layer, this is likely related to the inner rainband convection which has two effects; firstly by directly depriving the eyewall of moisture diabatic heating in the eyewall may be weakened (leading to reduced convection, a less well ventilated eyewall, an enhanced boundary layer outflow and spin down), but also by creating a region of convergence near this inner rainband from the induced balanced circulation which promotes convection near the rainband rather that at the eyewall. We have clarified this explanation in lines 618 – 624 within the discussion section.

**Figure 18d. How exactly does the lack of diabatic heating cause PV mixing? Nguyen et al. (2011) that the instability on the PV ring is a combination of barotropic and convective instability (as pointed out in L 567-568). In other words, the instability depends (in part) on the diabatic heating.**

The diabatic heating is necessary to maintain the barotropically unstable ring structure. Without the diabatic heating, the storm reverts to the more barotropically stable monopolar state and PV is mixed into the eye. We have made this clearer in the text.

**L 552. Should be "(Fig. 18 \*\*c\*\*)"?**

Changed.

**L 552. Why should the symmetric structure be maintained initially?**

Initially at the start of any strengthening phase inner rainband activity is weaker, and there is no or weak VHT-like activity. Throughout the strengthening phase these structures become more common and stronger and contribute to the less azimuthally symmetric structure. Initially though there has been no time for them to form, and with a greater amount of time passed the probability of a particularly strong VHT-like structure having developed increases. We have added a clarifying sentence.

**L 553. Why do the conditions for VHT-like structures become increasingly better? This is an important point that hasn't really been addressed.**

The eyewall convection becomes stronger throughout the strengthening phases, and the water vapour mixing ratio increases. Although convection is random and stochastic, as time passes and the region of the eyewall moistens any burst of convection is likely to be stronger,

and hence also has a greater probability of being disruptive to the eyewall. We have added a discussion of this point to the manuscript.

**L 568. "… VHT-like structures … seem to be a cause of the instability …". I don't see how you can conclude this if the instability is a combined barotropic-convective instability.**

We have changed the language so it sounds less speculative and added some additional clarification.

**4.5 Discussion. Kosin and Eastin (2001), perhaps the most important paper on the topic, has been left out of the discussion. I think it has to be included. In the terminology of Kosin and Eastin: regime 1 = ring structure —> relative intensification; and regime 2 = monopole —> relative weakening. Kosin and Eastin discuss how the moisture and equivalent potential temperature changes with the fluctuations. This has implications for the formation of convection and VHTs. How do their observations fit with the schematic?**

This is a good point. We have referenced the study by Kosin and Eastin (2001) now and have expanded the discussion of the results to compare with their results. In particular, we think their regimes 1 and 2 are similar to our strengthening phases and weakening phases, respectively. Fig. R8 shows the azimuthally-averaged equivalent potential temperature, is increasing in the eye, consistent with their hypothesis of eyewall to eye mixing and Figure 8 in the revised manuscript. which shows the gain in PV within the eye during the W1 weakening phase is due to the large scale transport of PV. Trajectories also show that during the course of W1 a weak inward and upward airflow develops within the eye as PV is transported towards the centre. We have not included Fig.R8 in the revised manuscript as it does not add further insight to the figures already presented.

[Figure]

**Figure. R8**: Azimuthally averaged equivalent potential temperature (K, shaded) and vertical velocity (0.1,0.2, 0.5, 2, 5, 10 ms$^{-1}$ positive (solid) and negative (dashed) black contours) for various times prior, during, and after the W1 weakening phase.

---

## Referee Report (RR1)

Review of the revised manuscript entitled: "Intensity fluctuations in Hurricane Irma (2017) during a period of rapid intensification", by William Torgerson, Juliane Schwendike, Andrew Ross and Chris J. Short, submitted to *Weather and Climate Dynamics Discussions*.

**Recommendation**: Major revision

**Summary**

Our summary of the first review of this manuscript read: "While this study is a commendable attempt to provide dynamical interpretations of the storm behaviour, in our view it falls short of providing a clear understanding of the phenomena described. Moreover, the summary cartoon devised to underpin the explanations raises a number of questions as highlighted below. The authors have tried to identify pieces of the intensity-fluctuation puzzle, but have not yet provided a convincing link between the various pieces." Unfortunately, this summary remains valid.

In our first review, we made a few suggestions for a way forward and the authors have made a commendable attempt to explore these suggestions. The authors have set themselves a tough problem to research, but reading the discussion section 4.5 and the summary and conclusions in section 6, it is not clear to us that they have made much headway. In fact, we really struggled to understand what they think is going on and to make matters worse, they seem to have changed their own view in going from the abstract to the conclusions. We have tried to articulate our concerns in detail in the hope that it will help the authors to rethink and revise what they have written to make the arguments clearer.

Review signed: Roger Smith and Michael Montgomery

**Comments and questions**

1. In the abstract, it is stated that "The boundary layer was found to play an important role in the cause of the intensity fluctuations with an increase in the agradient wind within the boundary layer *causing* (our emphasis) a spin-down just above the boundary layer during the weakening phases whereas during the strengthening phases the agradient wind reduces." For a start, doesn't this depend on whether the agradient wind is positive or negative? And what metric is being used to charaterize the "intensity fluctuations"? Are you talking about the maximum wind speed anywhere or the maximum 10 m wind that forecasters use? In the summary (section 6), you say that your study " ... emphasises the role of the inner rainbands in *causing* (our emphasis) weakening periods." Finally, the last part of the quoted sentence would be improved by using "weakens" instead of "reduces".

2. Incidentally, at line 6, what is the difference between an "isolated local" region and an "isolated" region?

Before reading the main body of the manuscript, we studied the Summary and Conclusions, hoping to gain an overview of the new insights emerging from the manuscript, but we were most disappointed. Almost every sentence raised scientific questions and by the end of the section we were no wiser. We strongly recommend that the authors go through this section in detail and revise accordingly to address our questions enumerated below.

3. At lines 716-718 we are told that "Key and novel results include the finding that intensity fluctuations are related to convective and barotropic structural changes with the asymmetric convection playing a key role in the fluctuations. So what are these key novel results?

4.   In the next sentence we are told that "Both unbalanced and balanced intensification processes were important with the balanced effect of inner rainband convection *leading to* (our emphasis) an unbalanced boundary layer response which, in turn, caused a spin-down during weakening phases." But where did the inner rain band convection come from in the first place?

5. At lines 721-722 we are told that: "In Hurricane Irma, during the second period of rapid intensification, intensity fluctuations occurred, defined as short term intensification and weakening periods". But how are these intensity fluctuations characterized? By the maximum tangential wind speed or the maximum total wind speed at 10 m (the forecaster definition)?

6. In the next sentence we are told that: "The tangential wind, *at all levels (our emphasis)*, increased more during strengthening phase than it decreased during weakening phase so the fluctuations do not prevent the storm from rapidly intensifying." Does at all levels mean everywhere in the flow? See last point.

7. In the next sentence we are told that: "During the weakening phase the mean sea level pressure rose nearly concurrently with the weakening of the tangential wind which was the opposite of e.g. Nguyen et al. (2011) where the weakening of the tangential wind was accompanied by a mean sea level pressure drop." Why is this information provided at this point and why is it a key finding? Key findings should consist of explanations.

8. The next key finding, at lines 727-729 is: "During strengthening phases the PV distribution was an elongated ring which became more azimuthally symmetric and monopole-like during weakening phases. This contradicts previous studies (e.g. Nguyen et al., 2011) which show an association between azimuthal symmetry and a ring-like radial state and use the terms interchangeably." It seems reasonable to ask what is meant by an elongated ring becomong more monopole-like means? Is it just that that the hole in the ring became smaller? The second sentence about the contradiction with previous studies is unclear, but if it qualifies as a key finding, it requires an explanation. And how many other studies does the finding contradict?

9. The next key finding, at lines 730-733 is: "The change in PV structure is thought to be linked to a build up of barotropic and convective instability during the strengthening phases. During the start of the next weakening phase a breakdown and reorganisation of the eyewall occurs as the diabatic heating is no longer strong enough to maintain the barotropically unstable state. This leads to PV being transported towards the eye and to a rapid increase in barotropic stability." First of all, "is thought to be linked to" is not a very strong statement. What is the basis for this thought? Second, what change in PV structure are you talking about here? In what way does diabatic heating maintain the barotropically unstable state and what is the relevance of the barotropically unstable state? Is there lateral PV mixing going on here as in Schubert et al (1999)? Note that mixing and instability are not synonymous. You say that "This leads to PV being transported … ", but to what, precisely, does "This" refer?

10.   The next key finding, at lines 734-735 is: "The increase in barotropic stability during the weakening phases makes the formation of the VHT-like structures less likely. As a result the eyewall becomes more azimuthally symmetric." The reader might ask, why these statements are true. What does the liklihood of "VHT-like structures" have to do with barotropic stability? Why would barotropic instability be favourable to VHTs? Wouldn't VHTs be more related to convective instability? And why would the reduced likelihood result in a more symmetric eyewall?

11.     The next key finding, at lines 736-739 is: "During strengthening phases, the diabatic heating distribution had a smaller radial spread and a stronger heating maximum which is located within the RMW. During weakening phases the heating was outside the RMW and had a greater radial spread than the diabatic heating during the strengthening phases. The change in diabatic heating during the weakening phase was linked to convection becoming weaker and the eyewall thicker." Was *all* the heating outside the RMW during weakening phases? Also, why did the convection become become weaker and what are the consequences of having a thicker eyewall?

12.     The next key finding, at lines 740-743 is: "The change in heating structure at the start of the weakening phase, associated with VHT–like structures forming just outside the eyewall near the inner rainbands caused the strengthening of the outflow jet above the boundary layer both directly through the induced balanced circulation and by depriving the eyewall of heat and moisture, weakening the eyewall convection and further reducing the ability of the eyewall to ventilate the mass inflow from the boundary layer." This whole statement is somewhat indigestible, but it raises some questions. First, why are the VHT structures forming just outside the eyewall? And how do you know that these "caused the strengthening of the outflow jet above the boundary layer"? Are you arguing that this is an enhanced "suction effect" associated with the VHTs? Also, how do you know that the VHTs deprived the eyewall of heat and moisture? Doesn't the weakening of the eyewall convection depend in part on the degree of convective instability?

13.     The penultimate key finding, at lines 744-749 is: "VHT–like structures were stronger and more common during strengthening phases than weakening phases and contributed positively to intensification through eddy advection of angular momentum. During the weakening phase as the VHT–like structures became less common, this lack of contribution to the tangential wind above the boundary layer likely led to further weakening. Vertical advection of absolute angular momentum contributes positively to intensification above the boundary layer. In the boundary layer the radial advection of mean absolute angular momentum contributes positively towards intensification." The first question is, in the first sentence, are you talking about the *vertical* eddy advection of angular momentum? To what "further weakening" does the lack of contribution to the tangential wind above the boundary layer lead to? What is weakening? In the penultimate sentence, what, precisely, does intensification refer to? The tangential wind speed? Regarding the last sentence, wouldn't it be most surprising if the radial advection of mean absolute angular momentum did not contribute positively towards intensification in the boundary layer, assuming of course that intensification refers to a spin up of the tangential wind speed?

14.     The final key finding, at lines 744-749 is: "Unbalanced dynamics were shown to play a role in the intensity fluctuations. During the weakening phases an unbalanced supergradient tangential flow within the boundary layer, which could not be adequately ventilated by the eyewall convection, produced an outflow jet, above the boundary layer, which acted to spin-down the flow above the boundary layer by transferring low angular momentum from the eye outwards." We do not understand the second sentence. First it is unclear to what "which" refers to in each case. You seem to be talking about the ventilation of a tangential component of the flow, but the immediate idea of ventilation refers to the radial inflow of mass. Why would the vertical advection of a supergradient tangential flow lead to spin down aloft?

**Other issues**

Having been unable to find much to latch on to from the Summary and Conclusions section, we were getting rather burned out and it is possible that other readers would have a similar problem. However, we did make an effort to understand the cartoon in Fig. 19 that the authors developed to summarize the processes responsible for the intensity fluctuations in their study. As it did in our first review, the cartoon still raises a number of questions:

15.     One very basic question is how the authors envisage VHTs differ from ordinary eyewall convection? Panel (a) of the cartoon highlights such a difference. In panel (b) it is indicated that these VHTs help strengthen winds above the boundary layer through radial eddy advection of absolute angular momentum, but so would any convection beyond the eyewall. Why are the two VHTs so special?

16.     In panel (c), why are the VHTs extending radially outwards in an upstream direction compared with panel (a)?

17.     In panel (d), in what way does convergence enhance convection? What aspect of convection is enhanced and why? Does convergence enhance the buoyancy within convection? Why are there reduced tangential winds above the boundary layer when, as indicated, the radial inflow above the boundary layer is enhanced? Wouldn't this inflow lead to enhanced spin up and therefore a larger inward pressure gradient force (assuming approximate gradient wind balance)?

18.     Panel (e) suggests a broadening of the eyewall. If that is the case, wouldn't this thickening decrease rather than increase the potential for barotropic instability?

19.     Panel (f): according to balance theory, the strength of the overturning circulation depends on the spatial gradients of diabatic heating rate and not on the heating rate, itself. Therefore invoking the diabatic heating as "insufficient to maintain a ring-like PV structure" is obscure. For the same reason, one cannot argue that "reduced diabatic heating means eyewall is less able to ventilate BL mass influx."

20.     A more general comment on the cartoon is that it does not appear to connect with the metric you use to characterize strengthening or weakening? See point 5 above. Further, is the cartoon consistent with the description in the Abstract?

---

## Author Response (AR2)

**Reply to the reviewers' comments**

We would like to thank Roger Smith and Mike Montgomery for carefully reading the manuscript and their suggestions to improve it. We have addressed all points below.

The manuscript has undergone substantial corrections particularly in the discussion and interpretation of the results. In particular we have clarified our results in a way that should be more intuitive to the reader and have been more careful to not over ascribe cause and effect between separate elements of the narrative.

1. **In the abstract, it is stated that "The boundary layer was found to play an important role in the cause of the intensity fluctuations with an increase in the agradient wind within the boundary layer *causing* (our emphasis) a spin-down just above the boundary layer during the weakening phases whereas during the strengthening phases the agradient wind reduces." For a start, doesn't this depend on whether the agradient wind is positive or negative? And what metric is being used to charaterize the "intensity fluctuations"? Are you talking about the maximum wind speed anywhere or the maximum 10 m wind that forecasters use? In the summary (section 6), you say that your study " ... emphasises the role of the inner rainbands in *causing* (our emphasis) weakening periods." Finally, the last part of the quoted sentence would be improved by using "weakens" instead of "reduces".**

   We found that the increase/decrease in the agradient wind during the weakening/strengthening phases does not depend on the sign of the agradient wind. This is shown in Figure 14a where the sharp increases or decreases are well aligned in the whole boundary layer regardless of whether the wind is subgradient (blue line) or supergradient (yellow line). The intensity fluctuations are defined with respect to the 10-m total wind speed, minimum sea level pressure and radius of 10-m total wind speed. These three parameters (and others) are shown in Fig. 4. We also no longer attribute cause and effect in the manner described.

2. **Incidentally, at line 6, what is the difference between an "isolated local" region and an "isolated" region?**

   We intended to mean there is a separation between these two regions. We have made this clearer in the text and describe these structures as 'isolated regions of high relative vorticity and vertical velocity'.

3. **At lines 716-718 we are told that "Key and novel results include the finding that intensity fluctuations are related to convective and barotropic structural changes with the symmetric convection playing a key role in the fluctuations. So what are these key novel results?**

   The intensity fluctuations are caused by the growth of wave-2 convective modes (during the strengthening phases) as a result of barotropic instability, that eventually destabilize the eyewall through balanced and unbalanced processes. We have made changes in the text to more explicitly highlight the role of barotropic instability in the intensity fluctuations. In

particular, details of the relationship between barotropic instability and the fluctuations are summarised in lines 783 to 809.

4. **In the next sentence we are told that "Both unbalanced and balanced intensification processes were important with the balanced effect of inner rainband convection leading to (our emphasis) an unbalanced boundary layer response which, in turn, caused a spin-down during weakening phases." But where did the inner rain band convection come from in the first place?**

The convection is generated stochastically but is increasingly more likely towards the end of a strengthening phase due to the increase in wave-2 barotropic instability. We have made this clearer in the text, in particular in lines 783 to 809.

5. **At lines 721-722 we are told that: "In Hurricane Irma, during the second period of rapid intensification, intensity fluctuations occurred, defined as short term intensification and weakening periods". But how are these intensity fluctuations characterized? By the maximum tangential wind speed or the maximum total wind speed at 10 m (the forecaster definition)?**

We define the weakening periods to be when the surface 10-m wind speed is decreasing or stable and the mean sea level pressure is increasing. This definition is merely used to define the start and end of the weakening and strengthening periods. The azimuthally averaged tangential wind above the boundary layer at 1500 m shows even bigger fluctuations which can be seen in Fig. 4. This is also explained in the summary section in lines 779 to 782.

6. **In the next sentence we are told that: "The tangential wind, at all levels (our emphasis), increased more during strengthening phase than it decreased during weakening phase so the fluctuations do not prevent the storm from rapidly intensifying." Does at all levels mean everywhere in the flow? See last point.**

We are referring to the height dependent maximum azimuthally averaged tangential wind. Throughout the text it has been made clearer when we are referring to maximum azimuthally averaged tangential wind or the maximum radius of azimuthal tangential wind. In addition, we have been clearer about the definition of the fluctuations (which is based on changes in 10m wind speed) and large changes that occur during these fluctuations, in particular just above the boundary layer in the azimuthally averaged 1500m tangential wind speed.

7. **In the next sentence we are told that: "During the weakening phase the mean sea level pressure rose nearly concurrently with the weakening of the tangential wind which was the opposite of e.g. Nguyen et al. (2011) where the weakening of the tangential wind was accompanied by a mean sea level pressure drop." Why is this information provided at this point and why is it a key finding? Key findings should consist of explanations.**

The main point we want to make here is that the mechanism proposed in Nguyen et al. (2011) where PV is mixed into the eye during the symmetric to asymmetric transition and causes a subsequent drop in pressure during the asymmetric phase is not happening in our study. We have restructured the text slightly to explain this point in more detail. In particular, we discuss the relevance of pressure drop in the context of Nguyen et al. (2011) in lines 651 to 664.

8. **The next key finding, at lines 727-729 is: "During strengthening phases the PV distribution was an elongated ring which became more azimuthally symmetric and monopole-like during weakening phases. This contradicts previous studies (e.g. Nguyen et al., 2011) which show an association between azimuthal symmetry and a ring-like radial state and use the terms interchangeably." It seems reasonable to ask what is meant by an elongated ring becomong more monopole-like means? Is it just that that the hole in the ring became smaller? The second sentence about the contradiction with previous studies is unclear, but if it qualifies as a key finding, it requires an explanation. And how many other studies does the finding contradict?**

"More monopolar" refers to the radial distribution of PV as measured by the metric in Fig. 7a. Specifically, "more monopolar" means a higher PV in the centre of the storm relative to the PV at the RMW at that height (in this case 1500m). However we no longer use this terminology given that a true monopole does not form, instead we refer to the distribution being less 'ring-like'. Elongated refers to the shape of the PV ring prior which is more elliptical in the strengthening phases (Fig. 7c). The key point being made here is that there is a change in both the azimuthal *and* radial structure of the PV. In previous studies the reduction in azimuthal symmetry was correlated to a reduction in radial symmetry; this is shown not to be the case in our study. No other study on vacillation cycles shows this anticorrelation, they either show a correlation or do not measure both radial and azimuthal distributions. Within the text we have also made the points clearer by referencing to appropriate figures when describing aspects of the storm structure such as PV ring eccentricity.

9. **The next key finding, at lines 730-733 is: "The change in PV structure is thought to be linked to a build up of barotropic and convective instability during the strengthening phases. During the start of the next weakening phase a breakdown and reorganisation of the eyewall occurs as the diabatic heating is no longer strong enough to maintain the barotropically unstable state. This leads to PV being transported towards the eye and to a rapid increase in barotropic stability." First of all, "is thought to be linked to" is not a very strong statement. What is the basis for this thought? Second, what change in PV structure are you talking about here? In what way does diabatic heating maintain the barotropically unstable state and what is the relevance of the barotropically unstable state? Is there lateral PV mixing going on here as in Schubert et al (1999)? Note that mixing and instability are not synonymous. You say that "This leads to PV being transported … ", but to what, precisely, does "This" refer?**

The barotropic instability may be implied from the horizontal distribution of the PV. A change in sign of the radial PV gradient implies a barotropically unstable state. By comparing, for example Fig. 6a to Fig. 6c, we can see that the initially barotropically unstable state, above the boundary layer becomes barotropically stable with no longer any change in sign in the radial PV gradient. So, to answer your 2nd question, this change in barotropic stability is linked to the radial change in the PV structure. In the absence of diabatic heating, since the PV distribution satisfies the Charney-stern criteria for barotropic instability, the ring structure will be mixed out over time. A breakdown, of sorts, does occur in the weakening phase, so the point being made is that the diabatic heating is no longer sufficient to maintain the barotropically unstable state which causes a reversion to a more monopolar PV distribution. We know from trajectory calculations (see Fig. 8) there is transport of PV inwards which is then mixed out in the eye, which happens at the end of the strengthening phase when instability is at its highest. We have made this clearer in the text the relevance of the changing PV structure is explained more clearly in the context of the ongoing weakening of the TC during the middle of the weakening phase

10.      **The next key finding, at lines 734-735 is: "The increase in barotropic stability during the weakening phases makes the formation of the VHT-like structures less likely. As a result the eyewall becomes more azimuthally symmetric." The reader might ask, why these statements are true. What does the liklihood of "VHT-like structures" have to do with barotropic stability? Why would barotropic instability be favourable to VHTs? Wouldn't VHTs be more related to convective instability? And why would the reduced likelihood result in a more symmetric eyewall?**

As in Nguyen et al. (2011) a combined convective, barotropic instability is being proposed and in a similar way the increase in the barotropic instability promotes the growth of these structures which, for clarity, we are now calling 'isolated regions of rotating deep convection'. As the isolated regions of rotating deep convection' grow, convection being preferentially stronger in these regions is what leads to a more asymmetric looking eyewall. In the absence of the isolated regions of rotating deep convection the convection is more uniform. We have made this clearer in the text.

11.      **The next key finding, at lines 736-739 is: "During strengthening phases, the diabatic heating distribution had a smaller radial spread and a stronger heating maximum which is located within the RMW. During weakening phases the heating was outside the RMW and had a greater radial spread than the diabatic heating during the strengthening phases. The change in diabatic heating during the weakening phase was linked to convection becoming weaker and the eyewall thicker." Was *all* the heating outside the RMW during weakening phases? Also, why did the convection become become weaker and what are the consequences of having a thicker eyewall?**

Not all the heating, but during the weakening phases the majority of the heating is, particularly during the middle of the weakening phase. For example, compare Fig. 12a with Fig. 12e. The weaker convection, from an azimuthally averaged perspective is linked to the convection from the VHTs that formed during the strengthening phase merging with the eyewall convection. One consequence of a thicker eyewall is a reduced radial gradient of heating, which has an effect on the balanced response. We have made this clearer in the text.

12.      **The next key finding, at lines 740-743 is: "The change in heating structure at the start of the weakening phase, associated with VHT–like structures forming just outside the eyewall near the inner rainbands caused the strengthening of the outflow jet above the boundary layer both directly through the induced balanced circulation and by depriving the eyewall of heat and moisture, weakening the eyewall convection and further reducing the ability of the eyewall to ventilate the mass inflow from the boundary layer." This whole statement is somewhat indigestible, but it raises some questions. First, why are the VHT structures forming just outside the eyewall? And how do you know that these "caused the strengthening of the outflow jet above the boundary layer"? Are you arguing that this is an enhanced "suction effect" associated with the VHTs? Also, how do you know that the VHTs deprived the eyewall of heat and moisture? Doesn't the weakening of the eyewall convection depend in part on the degree of convective instability?**

The isolated regions of rotating deep convection' do form in the eyewall, but on the major axis of the ellipse so from an azimuthally average perspective they are further out (see Fig. 13a for example which shows the location of two isolated regions of rotating deep convection' at either end of the eyewall along the major axis). We have experimented with the balanced model and have been able to show that this distribution of diabatic heating does lead to enhanced outflow

and, yes, we do posit a suction effect associated with the isolated regions of rotating deep convection'. We have been able to plot equivalent potential temperature to show that, during the weakening phase the region outside of the eyewall is a moister, warmer environment. We have made this clearer in the text.

13. **The penultimate key finding, at lines 744-749 is: "VHT–like structures were stronger and more common during strengthening phases than weakening phases and contributed positively to intensification through eddy advection of angular momentum. During the weakening phase as the VHT–like structures became less common, this lack of contribution to the tangential wind above the boundary layer likely led to further weakening. Vertical advection of absolute angular momentum contributes positively to intensification above the boundary layer. In the boundary layer the radial advection of mean absolute angular momentum contributes positively towards intensification." The first question is, in the first sentence, are you talking about the *vertical* eddy advection of angular momentum? To what "further weakening" does the lack of contribution to the tangential wind above the boundary layer lead to? What is weakening? In the penultimate sentence, what, precisely, does intensification refer to? The tangential wind speed? Regarding the last sentence, wouldn't it be most surprising if the radial advection of mean absolute angular momentum did not contribute positively towards intensification in the boundary layer, assuming of course that intensification refers to a spin up of the tangential wind speed?**

When we refer to 'eddy advection of angular momentum' we have changed this to eddy radial vorticity flux which is the third term in our equation 4. The eddy radial vorticity flux contributes more positively to the overall eddy advection term, and overall, the eddy advection is also a positive contributor to the tangential wind especially in the strengthening phases. In the weakening phases the vertical advection terms of equation 4, above the boundary layer, reduces which also contributes to the weakening, so the further weakening is describing the less intuitive eddy radial vorticity flux. 'Weakening and intensification' in this context refers to decreases and increases of the azimuthally averaged tangential wind around the RMW respectively. We agree the last sentence is confirming our intuition, the main novel point being made here is really that the eddies are contributing to the continued intensification of the TC above the boundary layer in the strengthening phases through eddy radial vorticity flux. We have made this clearer in the text.

14. **The final key finding, at lines 744-749 is: "Unbalanced dynamics were shown to play a role in the intensity fluctuations. During the weakening phases an unbalanced supergradient tangential flow within the boundary layer, which could not be adequately ventilated by the eyewall convection, produced an outflow jet, above the boundary layer, which acted to spindown the flow above the boundary layer by transferring low angular momentum from the eye outwards." We do not understand the second sentence. First it is unclear to what "which" refers to in each case. You seem to be talking about the ventilation of a tangential component of the flow, but the immediate idea of ventilation refers to the radial inflow of mass. Why would the vertical advection of a supergradient tangential flow lead to spin down aloft?**

We agree this is unclear, in terms of adequate ventilation we are referring to the strong mass influx in the boundary layer not the primary circulation as this sentence appears to read. This is a grammatical mistake rather than a misunderstanding of the concepts on our part. There are three reasons for the increased outflow (i) balanced effects above the BL which we described in a previous response, (ii) outward acting force as a result of the tangential flow becoming supergradient, and (ii) inability for convection to ventilate the frictionally induced mass convergence within the boundary layer. We have made this clearer in the text.

**Other issues**

Having been unable to find much to latch on to from the Summary and Conclusions section, we were getting rather burned out and it is possible that other readers would have a similar problem. However, we did make an effort to understand the cartoon in Fig. 19 that the authors developed to summarize the processes responsible for the intensity fluctuations in their study. As it did in our first review, the cartoon still raises a number of questions:

15.    **One very basic question is how the authors envisage VHTs differ from ordinary eyewall convection? Panel (a) of the cartoon highlights such a difference. In panel (b) it is indicated that these VHTs help strengthen winds above the boundary layer through radial eddy advection of absolute angular momentum, but so would any convection beyond the eyewall. Why are the two VHTs so special?**

Symmetrical convection beyond the eyewall might yield similar results though contributions to the mean advection of AAM but there is a mechanism for how this asymmetric convection might develop and then its impact on the TC eyewall. We updated the schematic and the text to make both clearer.

16.    **In panel (c), why are the VHTs extending radially outwards in an upstream direction compared with panel (a)?**

It is hypothesised that the isolated regions of rotating deep convection'that form within the eyewall are convectively coupled to outward propagating vortex Rossby waves that move upwind relative to the tangential flow. We updated the schematic and the text to make both clearer.

17.    **In panel (d), in what way does convergence enhance convection? What aspect of convection is enhanced and why? Does convergence enhance the buoyancy within convection? Why are there reduced tangential winds above the boundary layer when, as indicated, the radial inflow above the boundary layer is enhanced? Wouldn't this inflow lead to enhanced spin up and therefore a larger inward pressure gradient force (assuming approximate gradient wind balance)?**

Convergence increases outside of the eyewall during the start of the weakening phases. In cases where there is a conditionally unstable environment, convergence provides a lifting mechanism to initiate convection. Regarding reduced tangential winds, we acknowledge that this is unclear, tangential winds can increase at radii where the inflow is present. The mention of 'reduction of tangential winds' refers to the radial location around the eyewall where tangential wind above the boundary layer starts to drop at the start of the weakening phase. We updated the schematic and the text to make both clearer.

18.    **Panel (e) suggests a broadening of the eyewall. If that is the case, wouldn't this thickening decrease rather than increase the potential for barotropic instability?**

Yes it does, we have perhaps been unclear by referring to barotropic *stability* instead of instability which we have said increases. We updated the schematic and the text to make both clearer.

19.     **Panel (f): according to balance theory, the strength of the overturning circulation depends on the spatial gradients of diabatic heating rate and not on the heating rate, itself. Therefore invoking the diabatic heating as "insufficient to maintain a ring-like PV structure" is obscure. For the same reason, one cannot argue that "reduced diabatic heating means eyewall is less able to ventilate BL mass influx."**

Apologies, this is clumsy terminology on our part. In the PV tendency equation, the term we are referring to is the spatial gradient of diabatic heating projected onto the vertical component of absolute vorticity. Many studies crudely refer to this as simply 'diabatic heating' (e.g. Wang et al 2009). Further barotropic experiments in the absence of diabatic heating (and where the spatial gradient of DH will also be zero) show the ring structure is unstable and degrades into a monopole. It would be more rigorous for us to say, which is also true, that a high spatial gradient of diabatic heating is necessary to generate the PV necessary to maintain the hollow structure. This is what we have found, with the spatial gradient of diabatic heating being high prior to the start of the weakening phase.

Similarly, when we refer to greater values of diabatic heating in the context of ventilating mass flux in the boundary layer what we should say is greater radial gradients of diabatic heating.

We updated the schematic and the text to make both clearer.

**A more general comment on the cartoon is that it does not appear to connect with the metric you use to characterize strengthening or weakening? See point 5 above. Further, is the cartoon consistent with the description in the Abstract?**

Our definitions of 'weakening' and 'strengthening' phases revolve around surface changes in MSLP, RMW and total wind speed partly because these are more easily compared with the observations. However, we only use these quantities to define the start and end of weakening and strengthening phases. In fact, there are more dramatic structural changes that occur at the top of the boundary layer than at the surface which is why we concentrate, throughout, on the 1500-m level.